# P3D: Highly Scalable 3D Neural Surrogates for Physics Simulations with Global Context

**Benjamin Holzschuh**[*]     **Georg Kohl**     **Florian Redinger**     **Nils Thuerey**

School of Computation, Information and Technology
Technical University of Munich

## Abstract

We present a scalable framework for learning deterministic and probabilistic neural surrogates for high-resolution 3D physics simulations. We introduce *P3D*, a hybrid CNN-Transformer backbone architecture targeted for 3D physics simulations, which significantly outperforms existing architectures in terms of speed and accuracy. Our proposed network can be pretrained on small patches of the simulation domain, which can be fused to obtain a global solution, optionally guided via a scalable sequence-to-sequence model to include long-range dependencies. This setup allows for training large-scale models with reduced memory and compute requirements for high-resolution datasets. We evaluate our backbone architecture against a large set of baseline methods with the objective to simultaneously learn 14 different types of PDE dynamics in 3D. We demonstrate how to scale our model to high-resolution isotropic turbulence with spatial resolutions of up to $512^3$. Finally, we show the versatility of our architecture by training it as a diffusion model to produce probabilistic samples of highly turbulent 3D channel flows across varying Reynolds numbers, accurately capturing the underlying flow statistics. Our source code is available at `https://github.com/tum-pbs/P3D`.

## 1 Introduction

Training neural networks on high-resolution data substantially increases the required GPU memory and compute costs. Scaling models and their input dimensions typically requires substantial engineering effort, posing a major barrier to the widespread and cost-effective adoption of machine learning across application domains. Scientific machine learning and engineering are especially affected due to the multi-scale nature of relevant phenomena whose modeling often requires specialized and highly computationally demanding numerical solutions. In this paper, we focus on learning surrogate models for simulations focusing on fluid dynamics that have downstream applications in fields such as aerospace (Arranz et al., 2024), climate science (Bodnar et al., 2024), energy systems (Degrave et al., 2022), and biomedical engineering (Morris et al., 2016). Machine learning models inherently compete with existing solvers, which are often employed to create the reference targets for learning. As such, they need to either significantly outperform the corresponding solvers while maintaining an acceptable level of accuracy (Kochkov et al., 2021; Pestourie et al., 2023), or yield solutions where traditional solvers fall short, for example working with noisy (Franz et al., 2023) or only partial input data (Shu et al., 2023), or by providing uncertainty estimates (Jacobsen et al., 2023). A large fraction of papers in this area address learning problems for either low-dimensional or comparatively smooth data in 2D. In this paper, we deliberately focus on high-resolution 3D phenomena, covering a wide variety of different types of PDEs.

We propose *P3D*: a hybrid CNN-Transformer backbone built on PDE-Transformer (Holzschuh et al., 2025) that combines fast processing of local features via convolutions and windowed attention mechanisms for learning generalizable token representations. P3D makes important extensions for scaling to very high resolutions in 3D: *crop-based pre-training* combined with a designated module for processing *global dependencies*. This design achieves superior scalability and accuracy compared to existing 3D baselines. We evaluate the P3D backbone architecture through an extensive comparison

---

[*]Correspondence to `benjamin.holzschuh@tum.de`

with existing architectures for modeling physics simulations in 3D, training on 14 different types of dynamics simultaneously. We then employ P3D as a surrogate model for isotropic turbulence at resolution $512^3$ and demonstrate how the P3D architecture pretrained on crops of $128^3$ can be scaled to the entire domain, while achieving high accuracy with temporally stable autoregressive rollouts.

A significant difficulty when modeling large systems is how to aggregate and distribute information globally across the network. We propose to link the bottleneck layers of P3D with a sequence-to-sequence model, called *context model*, for an efficient global processing and information aggregation utilizing highly optimized self-attention mechanisms in LLM layers, which is combined with different scalable and memory-aware training and inference strategies. Additionally, we propose a direct mechanism to let aggregated global information flow back to the decoder modules via adaptive instance normalization layers. In our final experiment, we train P3D as a diffusion model to learn the distribution of velocity and pressure fields of a turbulent channel flow on a non-equidistant grid. This setup requires access to global information like the relative position to the walls, and cannot be addressed by learning local representations alone. We verify that velocity profiles from the generated samples of P3D closely match the ground truth, demonstrating that high distributional accuracy can be obtained even when the solution fields are constructed from smaller regions, which are only coordinated through the information flow from the context model. To summarize our contributions:

- We introduce *P3D*, a hybrid CNN-Transformer architecture for autoregressive prediction of *high-resolution physics simulations in 3D*, combining convolutions for fast learning of local features and windowed self-attention for deep representation learning.
- We demonstrate the efficiency and versatility of P3D in three experiments: (1) jointly learning multiple types of simulations (2) scaling P3D pretrained on crops of $128^3$ to a high-resolution simulation of isotropic turbulence at $512^3$, and, (3) generating probabilistic samples from P3D trained via flow matching for the velocity and pressure fields of a turbulent channel flow, closely matching the ground truth flow statistics.
- We propose mechanisms for efficient global information processing, including linking bottleneck layers with a sequence-to-sequence *global context model* and injecting global information into adaptive instance normalization layers via region tokens.
- We evaluate *different setups for finetuning*, which enable a more fine-grained control of precomputation and gradient backpropagation through encoder and decoder blocks to reduce VRAM and compute requirements.

## 2 RELATED WORK

**PDEs and machine learning** Machine learning has sparked much resonance in accelerating and improving numerical PDE solvers as well as fully replacing them (Kim et al., 2019; Thuerey et al., 2020). Approaches that are combined with existing PDE solvers can replace components of the solver (Bar-Sinai et al., 2019), learn closure models (Duraisamy et al., 2019; Sirignano & MacArt, 2023) or learn corrections to a fast solver on a coarse grid (Um et al., 2020; Kochkov et al., 2021; Dresdner et al., 2023). Other directions target problems that are more difficult to address with numerical solvers, such as inverse problems (Raissi et al., 2019; Bruna et al., 2022; Holzschuh et al., 2023), or uncertainty quantification (Xiao & Cinnella, 2019; Liu & Thuerey, 2024). A variety of tailored learned representations was likewise proposed for PDEs Hagnberger et al. (2024); Koupaï et al. (2025). Leveraging diffusion models for autoregressive prediction and inverse problems for PDEs has been explored by several works in the past (Lippe et al., 2023; Kohl et al., 2024; Shu et al., 2023; Shehata et al., 2025), albeit limited to data in 2D.

**Pretraining and 3D neural surrogates** Subramanian et al. (2023); Yang et al. (2023); McCabe et al. (2023) pretrain foundation models for PDEs on multiple PDE dynamics simultaneously. Such models can be used either for zero-shot-predictions or for finetuning when adapting to new dynamics, which allows for improved results with fewer training samples. Previous works have demonstrated learning PDEs in 3D, e.g., learning large-eddy simulations (Peng et al., 2023; Li et al., 2023a; 2024a; Jiang et al., 2025), and elastic wave propagation (Lehmann et al., 2024). Smoke buoyancy in 3D was targeted as a test case by Li et al. (2023b) Notably, most previous work targets resolutions of up to $64^3$, an exception being probabilistic experiments at a resolution of $128^3$ (Molinaro et al., 2024), and surrogate training with up to $128 \times 128 \times 256$ (Ohana et al., 2024). This motivates our contributions for scalable architectures, as surrogate models for truly high-resolution 3D physics simulations are of paramount interest in different scientific areas.

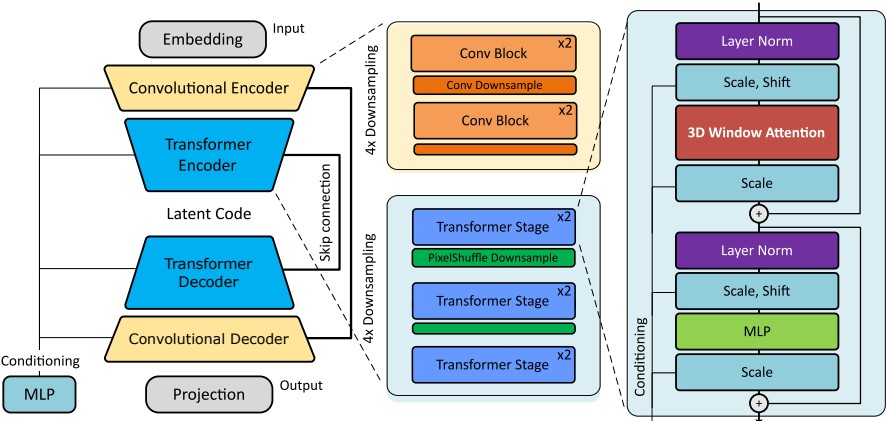

Figure 1: Overview of P3D. Convolutional blocks for local feature processing are combined with transformers for deep representation learning, yielding a U-shaped multi-scale architecture. The transformer backbone combines windowed attention and conditioning via adaptive instance normalization, which are modified and optimized for 3D.

**Scalable transformer architectures** Transformers have become one of the dominant backbone architectures in deep learning due to their high computational efficiency and their ability to model long-range causal relationships (Vaswani et al., 2017; Devlin et al., 2018). Transformers have also become a popular competitor to CNNs in vision and understanding tasks (Dosovitskiy et al., 2020; Rodrigo et al., 2024), scale to large images (Gupta et al., 2024), and have recently been adopted for learning surrogate models for physics simulations (McCabe et al., 2023; Wu et al., 2024; Alkin et al., 2024). A major computational difficulty is the quadratic complexity of the global self-attention mechanism. Liu et al. (2021) address this limitation by restricting the computation of the attention operation to a local window and Ho et al. (2019) by computing the attention only across the data axes.

## 3 METHOD

**Problem formulation** Let $\Omega$ denote a spatial domain with $n$ physical quantities $u(x,t)$ : $\Omega \times [0,T] \to \mathbb{R}^n$ that are discretized in time and space and described by the temporal sequence $[\mathbf{u}_0, \mathbf{u}_{\Delta t}, ..., \mathbf{u}_T]$. We consider all additional information about the sequence such as the type of PDE or hyperparameters of the simulator to be encoded in an $m$-dimensional conditioning vector $\mathbf{c} \in \mathbb{R}^m$. We assume the availability of many such sequences as training data, representing the temporal evolution of different types of PDEs with varying initial conditions or coefficients. We denote our proposed network architecture by $\mathcal{M}_\Theta$ with weights $\Theta$.

We address two main tasks in this paper. The first is *autoregressive prediction*: For a given sequence of $P$ preceding states $[\mathbf{u}_{t-P\Delta t}, ..., \mathbf{u}_{t-\Delta t}]$, denoted by $\mathbf{u}_{\text{in}}$, our target is to predict the next state $\mathbf{u}_t := \mathbf{u}_{\text{out}}$. The second task is to train a *probabilistic sampler* to draw samples from a distribution of states representing solutions for a PDE as specified by the parameter vector $\mathbf{c}$. In this case $\mathbf{u}_{\text{in}} = \emptyset$.

### 3.1 BACKBONE ARCHITECTURE: P3D TRANSFORMER

The key components of our proposed hybrid CNN-Transformer architecture P3D are the hierarchical U-shape structure with the hybrid encoder and decoder-pair based on convolutional and transformer blocks. A visual overview is given in Figure 1. In the following, we highlight the main components of the architecture and explain how they support its central goal to enable the efficient training and inference for high-resolution 3D simulations.

**Hybrid encoder/decoder** We utilize convolutional en- and decoders with skip connections. Fully transformer-based architectures that work in the pixel space for 2D data and images like ViTs rely on a patchification operation to transform patches of size $p \times p$ into tokens. A corresponding approach in 3D would transform patches of size $p^3$ into a single token, significantly increasing the amount of information encoded in each token. To balance both the number of tokens for the transformer as well as the information density of each token, we learn local features via the convolutional encoder to obtain an optimized compressed representation.

**Attention and positional encoding**  The self-attention operation used by transformers has quadratic complexity in the number of tokens. For 3D data, this becomes a major computational issue as the number of tokens grows cubically with the spatial discretization, leading to computational blow-up as the domain size increases. The central building block of our transformer encoder is the windowed multi-head self-attention  (Liu et al., 2021), which only computes self-attention within a local windows. For computing the attention scores between tokens, we use the log-spaced relative positions of tokens inside the same window.

The architecture of the P3D combines Swin transformers (Liu et al., 2021), diffusion transformers (Peebles & Xie, 2023, DiT) and UNets (Ronneberger et al., 2015) into a 3D variant, which can be seen as an extension of PDE-Transformer (Holzschuh et al., 2025) in 3D. The main alteration is the replacement of the patchification with large convolutional en- and decoders. We also removed the shifting of windows during the computation of self-attention. The convolutional en- and decoder follow the design of modern UNet blocks, using adaptive instance normalization and group normalization. We give a detailed description of the architecture in Section A.4.

## 3.2  CONTEXT NETWORK

The P3D backbone architecture does not include any absolute positional embeddings as well as no operations aggregating and distributing information globally. Thus, it has to rely on learning local features and dynamics within its perceptual field. This promotes translation-equivariance, which is an important inductive bias for surrogate modeling of PDEs. Our design choice to not include other physical inductive biases is intentional to keep the P3D backbone architecture flexible. For learning large-scale simulations, we pretrain P3D on smaller crops of the simulation domain and then scale the pretrained network to larger inputs. However, global information and long-range dependencies often play a crucial role to obtain correct solutions, which is also a major limitation of PDE-Transformer. To address this shortcoming, we introduce the context network.

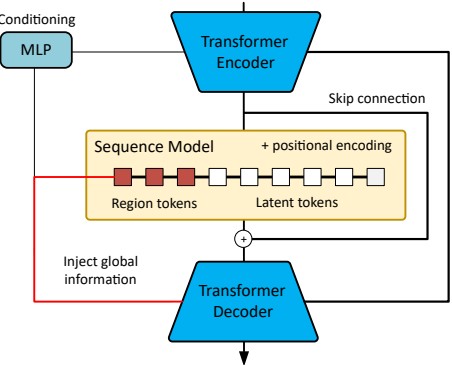

Figure 2:  Global context via a sequence model. The bottleneck layers are connected to the sequence model, which embeds the bottleneck representation as latent tokens. Region tokens are used to inject global information directly into the decoder.

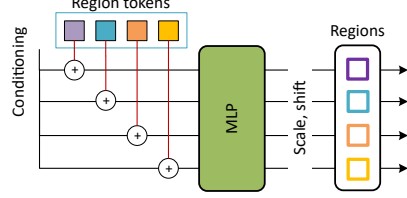

Figure 3:  Conditioning via the region tokens. The input domain is partitioned into regions/crops, each of which has a corresponding messenger token. The regions are modulated individually via learnable scale and shifts based on the region tokens.

**Token embeddings**  The bottleneck layer of the transformer encoder consist of tokens, which are embedded into latent tokens via a linear layer. The P3D encoder compresses a crop of size $16^3$ into a single latent token. Then, a frequency-based positional embedding vector is added to each latent token, similar to Dosovitskiy et al. (2020), which constitutes an absolute positional encoding of latent tokens. In addition, we partition the domain into *regions* and we match the size of regions with the size of the domain crops P3D was pretrained on. For each region, we include a corresponding so-called *region token* in the sequence of latent tokens, similar to the classification token in ViTs. Each region token is initialized via a learnable embedding layer and we add a frequency-based positional embedding vector. The purpose of region tokens is to serve as a more direct feedback mechanism to the decoder, which we describe in the next paragraph. Our implementation processes the token sequence via $n = 6$ transformer layers. Figure 2 provides an overview of this setup. In principle, any efficient sequence model can be used. After the sequence of region and latent tokens is processed, the latent tokens are added to the input of the decoder via a skip connection. See Section A.5 for a full description.

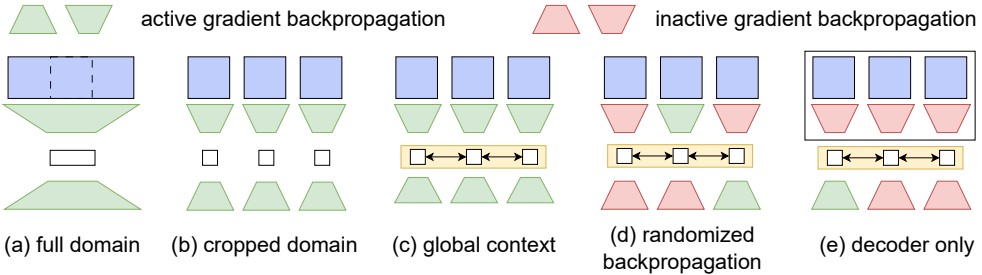

Figure 4: Different training and inference setups. (a) shows training on the full domain and (b) on domain crops. (c) includes the context network for global information processing, which can also be trained by randomly disabling gradient backpropagation for a percentage of the encoders and decoders, see (d). In (e) the latent codes from a pretrained encoder can be precomputed and only the context network and decoder are trained.

**Region tokens** The region tokens are retained and used as a more direct mechanism to let information flow through the decoder network. Region tokens are initialized as a learnable embedding vector with frequency-based positional encoding and are processed in the sequence model together with the latent tokens. Each region token corresponds to a crop that is processed independently of other crops by the encoder and decoder blocks. We use scale and shift operations to condition the decoder block of each region on the region token. Within each decoder block, for each adaptive instance normalization layer with scale or shift operations, we transform the region token via a linear layer to a region embedding vector which is added to the embedding vector of the conditioning **c**. Each region gets modulated differently based on the region token. This is visualized in Figure 3.

## 3.3 SCALING OUTPUT DOMAINS

We consider different setups for training and inference, see Figure 4, which include training on the full domain (a), on crops (b) and different training variants in combination with the context network (c) to (e). While it is preferable to train on small crops due to compute requirements, for inference, we generally want to process the full domain. We consider two strategies: (1) we scale to the full domain via the translation equivariance of the P3D architecture, i.e., we combine the domain crops and process them as a single input, and, (2) we encode and decode each crop of the full domain independently and combine the network outputs. We tag a model that is trained on crops of resolution $x^3$ and which internally runs inference on resolution $y^3$ by <$x|y$>. For example, a network trained on crops of size $64^3$ that is scaled via strategy (1) to resolution $128^3$ is tagged <64|128>, while the same network scaled via strategy (2) is tagged <64|64>. For strategy (2), if we use the context network for communication between the latent codes, we use the tag <X$x$|X$y$>.

## 4 EXPERIMENTS

We evaluate P3D as well as different scaling and finetuning setups in three experiments. P3D has 3 different configurations: $S$, $B$ and $L$ that determine the embedding dimension $d$ (32, 64 and 128 respectively) of the first layer. The corresponding models have 11M to 180M parameters. We denote the configurations with P3D-$S$ for our model with the $S$ config, changed accordingly for $B$ and $L$.

## 4.1 JOINTLY LEARNING MULTIPLE PDES

Our dataset for this task comprises 14 different types of PDE dynamics including Burger's equation, Kuramoto-Shivashinsky, Gray-Scott, Swift-Hohenberg and many others. The dataset is based on APEBench (Koehler et al., 2024), and a full description of each PDE with visualizations can be found in Section B.2 in the appendix. For all PDEs except Gray-Scott, we consider 60 different simulations with varying initial conditions and PDE-specific parameters such as viscosity, domain extent or diffusivity. For Gray-Scott, we include 10 simulations for each of its hyperparameters. Each spatially periodic simulation contains 20 snapshots discretized at resolution $320^3$ for Gray-Scott and $384^3$ for all other PDEs. We evaluate and benchmark models on random crops of the simulation domain of size $128^3$, $64^3$ and $32^3$ using supervised training, see Section A.1. Even with full information about simulation hyperparameters and the type of PDE, the behavior is not fully deterministic as quantities

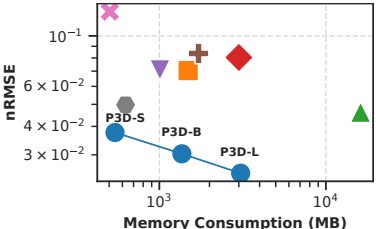 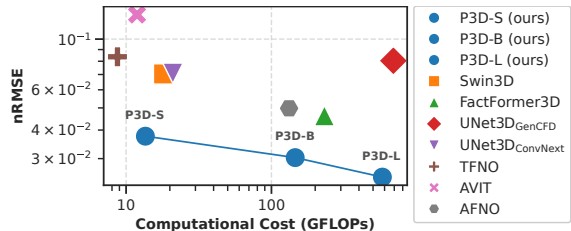

Figure 5: Comparison of model accuracy vs. (left) memory usage during backpropagation and (right) computational costs for inference for jointly learning different types of PDEs with crops of size $64^3$ for P3D and baselines.

beyond the cropped regions influence the solution inside it. Simulations have different numbers of channels and we zero-pad data with fewer channels than the number of maximum channels $N_C = 3$.

**Training on cropped data** Cropped data has an implicit time-dependent boundary condition, which is not known by the model. This can be seen as an extension and more difficult variant of the multi-physics training (Subramanian et al., 2023; Yang et al., 2023; McCabe et al., 2023), where in addition to not knowing the PDE or simulation hyperparameters, the model has to estimate the boundary conditions in a data-driven manner. Mathematically, the model $\mathcal{M}_\Theta$ is trained to regress

$$\underset{\Theta}{\arg\min} \ \mathbb{E}_{(s_t, s_{t+\Delta t}, \mathbf{c}) \in \mathcal{D}_{\text{train}}}[\mathbb{E}_{(s_t^{\text{crop}}, s_{t+\Delta t}^{\text{crop}})}[||\mathcal{M}_\Theta(s_t^{\text{crop}}, \mathbf{c}) - s_{t+\Delta t}^{\text{crop}}||_2^2]], \tag{1}$$

where we sample $(s_t, s_{t+\Delta t}, \mathbf{c}) \in \mathcal{D}_{\text{train}}$ from the training dataset and apply a random cropping to obtain $(s_t^{\text{crop}}, s_{t+\Delta t}^{\text{crop}})$. The input $\mathbf{u}_{\text{in}}$ corresponds to $s_t^{\text{crop}}$ and $\mathbf{u}_{\text{out}}$ to $s_{t+\Delta t}^{\text{crop}}$. The mapping $s_t^{\text{crop}} \mapsto s_{t+\Delta t}^{\text{crop}}$ is not deterministic since the boundary conditions are not prescribed. The model $\mathcal{M}_\Theta$ has to learn a prediction that minimizes the prediction error w.r.t. all possible simulation states that are outside the cropped domain, i.e., the optimal prediction $s^*$ for $s_{t+\Delta t}^{\text{crop}}$ minimizes

$$s^* = \underset{s}{\arg\min} \ \mathbb{E}_{(\hat{s}_t, \hat{s}_{t+\Delta t}) \sim \mathcal{D}_{\text{train}}} \left[ ||\text{crop}(\hat{s}_{t+\Delta t}) - s||_2^2 \mid \text{crop}(\hat{s}_t) = s_t^{\text{crop}} \right], \tag{2}$$

where $\text{crop}(\cdot)$ is the crop operation used for $s_t^{\text{crop}}$. The performance depends on how well the model is able to extrapolate the dynamics outside the cropped input for a short prediction horizon.

**nRMSE evaluation** We evaluate the nRMSE, see Section A.3, and consider a wide range of sota baseline architectures. Specifically, we include Swin3D, our own implementation of the SwinV2 architecture (Liu et al., 2021) extended to 3D, AViT, an axial vision transformer (McCabe et al., 2023), Adaptive FNOs (Guibas et al., 2021, AFNO), Tucker-Factorized FNOs (Kossaifi et al., 2023, TFNO), and FactFormer (Li et al., 2024b), Additionally, we consider two different convolutional UNet architectures, UNet$_{\text{ConvNeXt}}$ as used in Ohana et al. (2024) and UNet$_{\text{GenCFD}}$ (Molinaro et al., 2024). We train all models for 1000 epochs on four H100 GPUs. We use a fixed learning rate of $2.0 \cdot 10^{-4}$ for all models with the AdamW optimizer with weight decay $10^{-15}$ and batch size 256 in bf16-mixed precision. See Table 1 for the results and Tables B3 to B5 for detailed evaluations of each PDE type. As we train on bigger crop

Table 1: Comparison of the nRMSE ($\times 10^{-2}$) on the test dataset (averaged over all PDEs).

| Method | Crop size | | |
|---|---|---|---|
| epochs = 1000 | $32^3$ | $64^3$ | $128^3$ |
| TFNO | 8.46 | 8.37 | - |
| FactFormer | 6.24 | 4.62 | - |
| UNet$_{\text{ConvNext}}$ | 8.59 | 7.09 | - |
| UNet$_{\text{GenCFD}}$ | 7.61 | 8.04 | 8.27 |
| AViT | 20.9 | 25.0 | 15.1 |
| Swin3D | 7.92 | 7.04 | 5.03 |
| AFNO | 9.95 | 4.98 | 4.79 |
| P3D-$S$ | 6.27 | 3.76 | 3.33 |
| P3D-$B$ | 4.69 | 3.03 | 2.52 |
| P3D-$L$ | **4.13** | **2.49** | **2.08** |

sizes, the observed simulation domains becomes larger, increasing the amount of information that becomes available to the network. At the same time, the relative volume of the boundary becomes smaller, thus decreasing its relative weight and causing an according decrease in the nRMSE. P3D performs best across all crop sizes. Performance significantly improves when scaling the model size from $S$ to $L$.

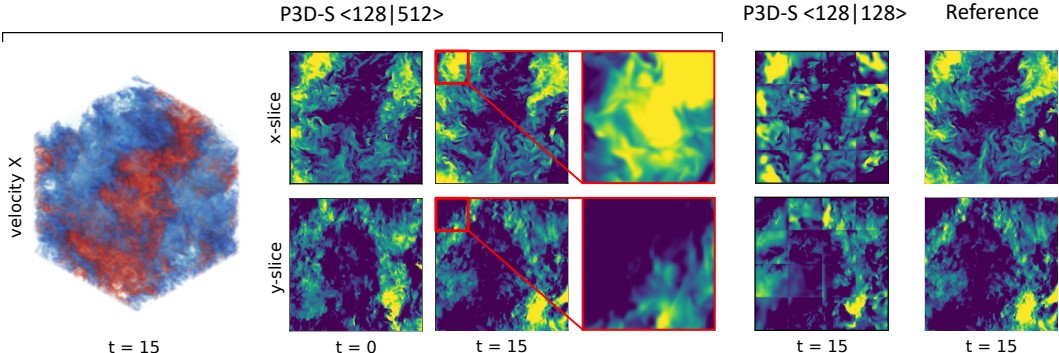

Figure 6: Forced isotropic turbulence. Prediction on the test set at resolution $512^3$ with an autoregressive rollout of 15 steps. The model is pretrained on patches of size $128^3$, without finetuning on $512^3$. P3D-$S$ <128|512> successfully scales to the full domain, whereas for P3D-$S$ <128|128> discontinuities at the borders of crop regions become apparent.

**Memory and compute** For scaling an architecture to high-resolution 3D simulations, the memory requirements as well as the inference speed are essential. Transformer architectures have been shown to achieve improved accuracy as the number of parameters and floating point operations increase, therefore comparing different architectures needs to take both factors into account. In Figure 5, we compare the nRMSE at patch size $64^3$ against (1) the computational cost for inference measured in GFLOPs, and (2) the VRAM consumption in MB for a backward pass with batch size 1. The P3D networks provide the best tradeoff between accuracy and computational cost/memory requirements. See Table A2 for additional information.

**Ablation on network design** We empirically verify our network design and consider: (1) P3D without the transformer backbone, which purely relies on the convolutional encoder and decoder, denoted *P3D-conv*, (2) a patch-based P3D with linear tokenizer for patch size $p = 4$ and no convolutional encoder and decoder, denoted *P3D-patch*. This baseline ($^*$) extends the mixed channel version of *PDE-Transformer to 3D*. (3) disabling PDE type conditioning by setting $\mathbf{c} = \mathbf{0}$, denoted *P3D-no-$\mathbf{c}$*, (4) P3D with window shifting enabled, *P3D-shift*. Additionally, we evaluate different values of the window size $w = 2$, $w = 4$ (default) and $w = 8$. P3D achieves a relative improvement of 16.0% ($32^3$), 26.7% ($64^3$) and 37.5% ($128^3$) over 3D PDE-Transformer($^*$), see Table 2. Shifting and larger window sizes give no significant improvements here.

Table 2: Validation MSE ($\times 10^{-3}$, time-weighted EMA with $\lambda = 0.99$).

| Method | Crop size | | |
|---|---|---|---|
| epochs = 400 | $32^3$ | $64^3$ | $128^3$ |
| P3D-$S$-conv | 8.33 | 5.40 | 3.25 |
| P3D-$S$-patch$^*$ | 6.48 | 3.78 | 2.16 |
| P3D-$S$-no-$\mathbf{c}$ | 5.69 | 2.94 | 1.41 |
| P3D-$S$-shift | **5.41** | 2.84 | 1.37 |
| P3D-$S$ | 5.44 | **2.77** | **1.35** |
| P3D-$S$ $w = 2$ | 5.68 | 2.96 | 1.49 |
| P3D-$S$ $w = 4$ | 5.44 | 2.77 | 1.35 |
| P3D-$S$ $w = 8$ | 5.44 | 2.90 | 1.32 |

## 4.2 ISOTROPIC TURBULENCE

The goal of the this experiment is to scale P3D to a high-resolution simulation involving complex dynamics. For this, we consider forced isotopic turbulence simulated via direct numerical simulation (DNS) at resolution $1024^3$ provided by the John Hopkins Turbulence Database (Perlman et al., 2007). The dataset is cropped from the original resolution to $512^3$ with a total of 500 snapshots, which are saved after reaching a statistical stationary state and comprises channels for the velocity X/Y/Z and pressure. The data is split into test and training sets, where the first 420 snapshots are used for training and the last 80 snapshots for testing.

**RMSE and enstrophy error for crops $128^3$** We first evaluate the performance of P3D compared to baselines from the previous experiment trained on domain crops of size $128^3$. We consider an evaluation of autoregressive rollouts from 1 to 15 steps. We evaluate the RMSE and a spectral error based on the enstrophy spectrum, which we compute based on the vorticity that is derived from the velocity fields of the data, see Section C.3. We train all models for 4000 epochs using the same setup as in Section 4.1, but reduce the batch size to 32. P3D performs best and achieves the lowest RMSE

for few autoregressive rollout steps. It consistently has by far the best spectral error, see Table 3 and Table C1 for an extended evaluation. For many autoregressive rollout steps, the RMSE becomes less informative, as the prediction starts to deviate from the reference due to the uncertainty from the boundary of the crop regions. See Figures C6 and C7 for visualizations.

**Scaling P3D to $512^3$** We scale P3D pre-trained on crops of size $128^3$ to the full domain $512^3$. Training P3D on cropped data requires less than a day on four A100 GPUs. See Section C.4 for details. We compare the performance of P3D when scaling the network to larger domain sizes using the scaling strategies introduced in Section 3.3. P3D $\langle 128|128 \rangle$ processes blocks independently using the original training resolution. Thus the domain $512^3$ is split into $64$ blocks of size $128^3$ that are processed independent of each other. P3D $\langle 128|512 \rangle$ processes the full domain, leveraging the translation-equivariance of the architecture. See Table 4 for an evaluation of the RMSE. Increasing the domain size during inference consistently gives improvements. With increasing domain size, the relative volume of the boundary shrinks, thus the uncertainty of turbulent motions is reduced and networks are able to provide more accurate predictions. Similar to previous results, larger networks improve performance as well. Note that the RMSE is different between Tables 3 and 4, since we do not consider longer rollouts in Table 4 which affects the distribution of samples in the test set. In this experiment, due to the isotropic and homogeneous nature of the simulation, we achieve highly accurate results without finetuning or global information.

Table 3: RMSE ($\times 10^{-2}$) and the enstrophy spectrum error ($\times 10^2$) at different autoregressive rollout steps on the test set with crop size $128^3$.

| Method | RMSE | | Enstrophy | |
|---|---|---|---|---|
| | 1 | 15 | 1 | 15 |
| UNet$_{\mathrm{GenCFD}}$ | 5.48 | 67.7 | 4.2 | 140 |
| Swin3D | 3.22 | 24.3 | 4.0 | 156 |
| AViT | 9.45 | 37.7 | 26.7 | 112 |
| AFNO | 3.69 | 29.8 | 7.6 | 190 |
| P3D-$S$ | 2.81 | 28.2 | 2.15 | 31.9 |
| P3D-$B$ | **2.04** | 31.5 | **0.72** | **19.2** |

Table 4: RMSE ($\times 10^{-2}$) on the test set for P3D trained on crop size $128^3$ and evaluated at $512^3$ for different scaling strategies.

| Method | Scaling strategy | | |
|---|---|---|---|
| | <128\|128> | <128\|256> | <128\|512> |
| P3D-$S$ | 1.90 | 1.68 | 1.60 |
| P3D-$B$ | 1.38 | 1.24 | - |

**Long auto-regressive rollouts** To evaluate accuracy over prolonged periods, we evaluated additional tests comparing the full-scale an individual crop models for 50 rollout steps. We performed inference with the scaled-up model P3D-$S$ $\langle 128|512 \rangle$ on the full domain, while for evaluation, we computed the spectral error and RMSE on the $128^3$ crops to ensure the metrics are directly comparable to those in Table 3.

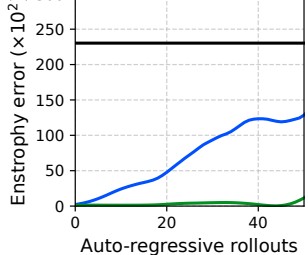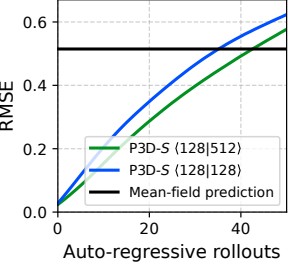

Figure 7: Enstrophy spectrum error ($\times 10^2$) and RMSE over autoregressive rollout steps at resolution $512^3$.

P3D-$S$ $\langle 128|512 \rangle$ maintains an excellent enstrophy spectrum error below $10 \times 10^2$ for the full period, whereas P3D-S $\langle 128|128 \rangle$, which processes individual regions, achieves this for only 10 steps. We note that the $512^3$ domain itself is cropped from an even larger $1024^3$ resolution, introducing boundary uncertainties that complicate auto-regressive unrolling, even for the scaled-up model. Not being able to reliably compare small scale structures, the RMSE values are still slightly lower for the scaled-up $\langle 128|512 \rangle$ model. This P3D-$S$ version outperforms the mean-field prediction for up to 40 to 50 steps. In particular, the enstrophy comparison highlights the advantages of applying P3D models pre-trained on $128^3$ crops to the full $512^3$ domain for auto-regressive rollouts.

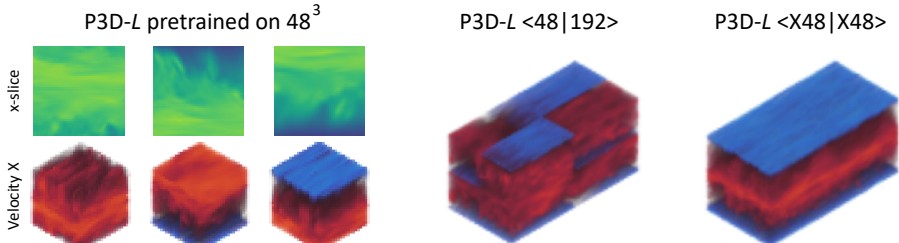

Figure 8: Turbulent channel flow. We pretrain PDE-$L$ on crops of size $48^3$. For P3D-$L$ <48|192> scaled to the full domain $96^2 \times 192$, the relative positions of crops to the wall is critical. Without absolute positional encodings and global information passing, the generated samples are incorrect. P3D-$L$ <X48|X48> finetuned with the context network resolves this issue.

### 4.3 TURBULENT CHANNEL FLOW

For the last experiment we train P3D as a generative model for learning to sample from a turbulent channel flow simulation with a periodic channel with no-slip boundaries at $\pm$y and is driven by a dynamic forcing to prevent a loss of energy. The seemingly simply geometry represents a well-studied and relevant scenario that is highly challenging as it requires fine spatial and temporal discretizations with correspondingly long simulation times to provide converged turbulence statistics (Hoyas & Jiménez, 2008).

Due to the very costly initial transient phase of these simulations, it is especially attractive to phrase the turbulent channel flow (TCF) problem as a probabilistic learning problem, where states from the relevant equilibrium phase can be sampled directly, i.e. without resolving the initial warm-up phase. We generate a dataset comprising 20 simulations with Reynolds numbers Re within the interval $[400, 800]$ spaced equidistantly, see Section D.1. The computational grid comprises $96 \times 96 \times 192$ spatially adaptive cells with a finer discretization near the wall. The data contains channels for the velocity in X/Y/Z direction as well as pressure.

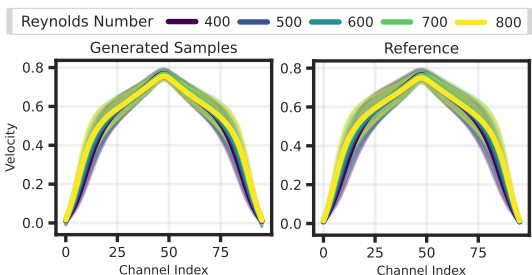

Figure 9: Flow statistics of samples produced by P3D-$L$ conditioned with varying Reynolds number (left) and the time-resolved DNS (right).

**P3D-$L$ pretrained on** $48^3$  We train P3D-$L$ as a diffusion model following Section A.2 on crops of size $48^3$. Pretraining on crops requires significantly less compute and VRAM and the network converges much faster. For P3D-$L$ with scaling strategy <48|192>, the network is not aware of the position of the wall relative to the position of the individual regions, causing the global structure of the generated flow to be incorrect, see Figure 8. For a quantitative statistical evaluation including training on the full domain and baselines, see Table 5.

Table 5: L2 error for the mean $\mu$ ($\times 10^{-5}$) and variance $\sigma^2$ ($\times 10^{-5}$) of the velocity profiles along the channel flow direction, see Section D.2 for details. For finetuning, we report the mean and std. dev. over 5 different seeds.

| Model | Mean L2 | Variance L2 | VRAM | epochs |
|---|---|---|---|---|
| UNet$_{\text{GenCFD}}$ full domain | 132.38 | 17.66 | 17.4 GB | 400 |
| AFNO full domain | 28.73 | 1849.3 | 3.4 GB | 400 |
| P3D-$L$ full domain | **3.02** | **13.20** | 14.9 GB | 400 |
| P3D-$L$ <48|192> | 5862.81 | 233.77 | 2.8 GB | 2000 |
| finetune w/o region tokens | 4541 ± 495 | 2026 ± 267 | 15.8 GB | 20 |
| finetune | 23.6 ± 21.4 | 40.4 ± 49.4 | 15.8 GB | 20 |
| finetune, decoder only | 941 ± 484 | 1170 ± 392 | 6.0 GB | 20 |
| finetune, decoder only | 97.7 ± 102.6 | 131 ± 149 | 6.0 GB | 100 |
| finetune, decoder only | **16.8 ± 5.0** | **24.1 ± 17.2** | 6.0 GB | 500 |

**Ablation on finetuning and context**  We evaluate finetuning P3D pretrained on the cropped domain $48^3$ using the context model, see Table 5. Finetuning corresponds to P3D-$L$ <X48|X48> with different training setups, see Figure 4. *finetune* and *finetune w/o region tokens* follows the setup of Figure 4c,

which backpropagates gradients through all crops and network modules. While this requires more VRAM than training on the full domain, only few epochs to achieve good results are necessary. For *finetune, decoder only*, we only backpropagate gradients through the decoder and context network, and also only backpropagate through 10% randomly selected decoder blocks, which corresponds to the setup in Figure 4e. We do not precompute the bottleneck representations from the frozen encoders, but this could be done to further reduce the VRAM and compute requirements. This setup achieves a significant reduction in VRAM, but requires more training epochs. The generated sample for P3D-*L* <X48|X48>, see Figure 8, qualitatively shows the correct global structure and content of individual regions. See also Figure 9 and Figures D6 to D9 for comparisons of the velocity profiles. While finetuning matches the statistics well, there are still visible discontinuities between generated crop regions, which leaves further room for improvement.

**Benchmarking speedups**    We compare the runtime per sample (s) and the mean L2 error of the different methods, see Figure 10. We also compare against running the simulation at half resolution $48^2 \times 96$. The warmup phase of the DNS takes on average 2 hours and 24 minutes, which cannot be skipped. After the warumup, we simulate a total time of $t = 20$. To calculate statistics, we take samples with $\Delta t = 1$ to avoid high autocorrelation. The reference simulation at full resolution has a runtime of 454s per sample. For the generative models, we consider 20 samples for each Reynolds number and 100 inference steps. Timings are obtained on a L40 GPU. P3D-*L* and AFNO represent the Pareto frontier for the generative models regarding the mean L2 error.

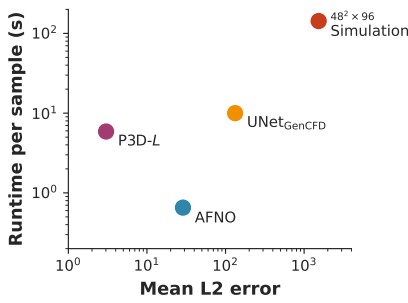

Figure 10: Runtime per sample (s) vs. Mean L2 error, see Table 5, w.r.t. to reference simulation at $96^2 \times 192$.

## 5 LIMITATIONS

P3D relies on grids to achieve high computational efficiency. Extending P3D to arbitrary geometries is not trivial, and requires careful modification of each architectural layer. Also, training P3D on datasets featuring diverse data modalities, resolutions, and dimensionalities remains an unaddressed task. The finetuning process using the context network decodes each region independently, which can cause discontinuities at the boundary of each region. Further strategies to prevent and mitigate discontinuities are an interesting direction for future work.

## 6 CONCLUSION

We have presented *P3D*: an efficient hybrid CNN-Transformer architecture for learning surrogates for high-resolution 3D physics simulations. We demonstrated the strong advantages of P3D over a comprehensive list of baselines for simultaneously learning different types of PDE dynamics, showing improved accuracy and stable training while at the same time being faster and more memory efficient than the strongest competitors. We scaled P3D to a high-resolution isotropic turbulence simulation by pretraining on smaller crops from the domain, and demonstrated its capabilities as a probabilistic generative model. The P3D model accurately predicts distributions of high-resolution velocity and pressure fields for a turbulent channel flow with varying Reynolds numbers, demonstrating how to include global information and coordinated pretrained networks via a global context model. Our architecture enables the scaling of neural surrogate models to very high resolutions, unlocking their potential to deliver real-world impact across scientific domains.

ACKNOWLEDGEMENTS

This work was supported by the ERC Consolidator Grant *SpaTe* (CoG-2019-863850). The authors gratefully acknowledge the scientific support and HPC resources provided by the Erlangen National High Performance Computing Center (NHR@FAU) of the Friedrich-Alexander-Universität Erlangen-Nürnberg (FAU) under the NHR project b278bb. NHR funding is provided by federal and Bavarian state authorities.

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

# A  TRAINING METHODOLOGY AND NETWORK ARCHITECTURES

## A.1  SUPERVISED TRAINING

For tasks that have a deterministic solution, such as training a surrogate model for a numerical solver, the P3D can be trained in a supervised manner using mean squared error (MSE) loss, enabling fast, single-step inference. In this case the network is directly optimized with the MSE

$$\mathcal{L}_S = \mathbb{E}\left[||\mathcal{M}_\Theta(\mathbf{u}_{\text{in}}, \mathbf{c}) - \mathbf{u}_{\text{out}}||_2^2\right]. \tag{3}$$

## A.2  PROBABILISTIC TRAINING

To allow for sampling from the full posterior distribution rather than approximating an averaged outcome, diffusion training is preferable for probabilistic solutions. For such cases, we employ the flow matching formulation of diffusion models (Lipman et al., 2023; Liu et al., 2023; Ho et al., 2020). Given the input $\mathbf{u}_{\text{in}}$ and a conditioning vector $\mathbf{c}$, samples $\mathbf{x}_0$ drawn from a noise distribution $p_0 = \mathcal{N}(0, I)$ are transformed into samples $\mathbf{x}_1$ from the posterior distribution $p_1$ by solving an ordinary differential equation (ODE) of the form $d\mathbf{x}_t = v(\mathbf{x_t}, t); dt$. Then the model $\mathcal{M}_\Theta$ learns the velocity field $v$ by regressing a vector field that defines a probabilistic trajectory from $p_0$ to $p_1$. Samples along this trajectory are produced through the forward process

$$\mathbf{x}_t = t\, \mathbf{u}_{\text{out}} + [1 - (1 - \sigma_{\text{min}})t]\, \epsilon \tag{4}$$

for $t \in [0, 1]$ with $\epsilon \sim \mathcal{N}(0, I)$ and a time threshold of $\sigma_{\text{min}} = 10^{-4}$. The velocity $v$ can be regressed by training via

$$\mathcal{L}_{\text{FM}} = \mathbb{E}\left[||\mathcal{M}_\Theta(\mathbf{u}_{\text{in}}, \mathbf{x}_t, \mathbf{c}, t) - \mathbf{u}_{\text{out}} + (1 - \sigma_{\text{min}})\epsilon||_2^2\right]. \tag{5}$$

After training, samples can be drawn from the posterior conditioned on $\mathbf{u}_{\text{in}}$ and $\mathbf{c}$, by sampling $\mathbf{x}_0 \sim \mathcal{N}(0, I)$ and integrating the corresponding ODE $d\mathbf{x}_t = \mathcal{M}(\mathbf{u}_{\text{in}}, \mathbf{x}_t, \mathbf{c}, t)\, dt$ over the time interval $t = 0$ to $t = 1$. We typically employ explicit Euler steps with a suitable, chosen step size $\Delta t$.

## A.3  NORMALIZED RMSE

The normalized RMSE (nRMSE) is defined as

$$\text{nRMSE} = \frac{1}{M} \sum_{i=1}^{M} \sqrt{\frac{\text{MSE}(\hat{\mathbf{u}}_{\text{out}}, \mathbf{u}_{\text{out}})}{\text{MSE}(\mathbf{0}, \mathbf{u}_{\text{out}})}}, \tag{6}$$

where $\hat{\mathbf{u}}_{\text{out}}$ is the network prediction and $M$ corresponds to the number of trajectories for a specific PDE in the test dataset. In our evaluation, we calculate the nRMSE for each PDE dataset and report the nRMSE averaged over all PDE datasets.

We can also define the nRMSE for autoregressive rollouts over the entire simulation trajectory. We define the $\text{nRMSE}_t$ at time $t$ by comparing the predicted state $\hat{\mathbf{u}}_t^S$ for a given system $S$ at time $t$ with the reference $\mathbf{u}_t^S$. We average over all systems $S$ for each PDE dataset.

## A.4  P3D ARCHITECTURE

We provide additional details on the backbone architecture of P3D below.

**Embedding of time, class labels and physical parameters**  We combine all three types of conditionings within a combined embedding layer. Time for flow matching/diffusion and physical parameters are implemented via timestep embeddings. Class labels are implemented via label embeddings. The embedding vectors of all three types are added and used as the joint embedding. The embedding dimension for each in the convolutional encoder/decoder is 64. In our experiments, class labels are used in Section 4.1. Physical parameters and timestep embeddings are used in Section 4.3.

**Convolutional encoder**   The convolution encoder first embeds the input using a Conv3D layer (kernel size 3, padding 1) with filters that correspond to the embedding dimensions of the configuration. This is followed by downsampling layers implemented via Conv3D layers (kernel size 3, padding 1, stride 2). Intermediate states before each downsampling operations are saved for residual connections. Encoder blocks and consecutive downsampling are applied twice. For each layer, the corresponding number of filters is shown in Table A1. Encoder blocks are repeated twice. Each encoder block consists of GroupNormalization layers, followed by GELU activations, Conv3D layer (kernel size 3, padding 1), GroupNormalization, modulation via shift and scale operations depending on the conditioning, GELU and an additional Conv3D layer (kernel size 3, padding 1). The input and output of each encoder block are connected via skip connections. The shift and scale vectors are learned via linear layers from the embedding vectors of the convolutional encoder/decoder.

**Convolutional decoder**   The design of the convolutional decoder mirrors the convolutional encoder in a U-shape architecture with residual connections. Upsampling layers are implemented via a combination of Conv3D layers to increase the number of filters and PixelShuffle3D layers. For an input number of channels $C_{\text{in}}$ and a target number of channels in the upsampled output $C_{\text{out}}$, the Conv3D operation first expands the number of channels $C_{\text{in}} \times H \times W \times D \rightarrow 8\,C_{\text{out}} \times H \times W \times D$ and PixelShuffle3D spatially rearranges the pixels $8\,C_{\text{out}} \times H \times W \times D \rightarrow C_{\text{out}} \times 2H \times 2W \times 2D$.

## A.5   Context model

The context module is a transformer-based architecture for processing 3D volumetric data in a sequence-like fashion. The model's design and operation can be broken down into the following key components:

**Latent tokens**   The model accepts a 5D tensor of shape $(B, C, H, W, D)$ with batch size $B$, token embedding dimension $C$, height $H$, width $W$, and depth $D$, corresponding to the token representations of the transformer encoder block arranged on a 3D spatial grid according to their positions. This input is first reshaped into a 2D sequence of tokens with shape $(B, (H \times W \times D), C)$ by flattening the spatial dimensions $(H, W, D)$ into a single sequence length. This token sequence is then passed through an input projection layer to map it to the model's internal hidden size $d_C = 512$. The new tokens are called *latent tokens*.

**Region tokens**   The model utilizes learnable region tokens, which are a set of `nn.Parameter` tensors. The number of tokens is defined by the number of regions. The region tokens are initialized from a normal distribution with a standard deviation of $10^{-6}$. Regions tokens are appended to the input sequence and are used to obtain embeddings for each region, which are injected into the adaptive instance normalization layers of the decoder modules.

**Positional encoding**   Positional information is incorporated using 3D sine-cosine positional embeddings. The embeddings are added to the input token sequence after the initial projection and concatenation with region tokens. The positional embedding for a given position $(w, h, d)$ and dimension $i$ is calculated by concatenating three separate 1D sine-cosine embeddings for each spatial axis. The 1D positional embedding at position $p$ and dimension $i$ are calculated via:

$$\text{PE}(p, 2i) = \sin\left(\frac{p}{T^{2i/d_P}}\right) \tag{7}$$

$$\text{PE}(p, 2i+1) = \cos\left(\frac{p}{T^{2i/d_P}}\right), \tag{8}$$

where $d_P = d_C/3$ is the embedding dimension per axis, and $T = 10$ is a temperature hyperparameter. Both latent tokens and regions tokens have their own positional encoding.

**Core architectural blocks**   The sequence of tokens is processed by a stack of $n_L = 6$ transformer blocks. Each block consists of:

- **RMSNorm**: A root mean square normalization layer that normalizes the input to a unit RMS, followed by a learned scaling factor. It is a more computationally efficient alternative to traditional layer normalization.

- **Self-Attention**: A multi-head self-attention module that computes attention weights to model relationships between tokens in the input sequence. This block receives the input query tokens and calculates query, key, and value vectors. There is no causal masking. We use hyper attention (Han et al., 2023), this is a specialized attention variant designed for efficient long-sequence processing.
- **MLP**: A gated multi-layer perceptron with a GELU activation function.

**Output**  The processed latent tokens are projected back to the original token size $C$ via a linear layer and reshaped to match the input of the context network. The resulting tensor is added to the input of the transformer decoder, forming a skip connection. For each adaptive instance normalization module in the decoder, the regions tokens are embedded via a unique linear layer. The region embeddings are added to the input of the adaptive instance normalization layer of their corresponding region, resulting in a different modulation for each region.

Table A1: Different configurations $S$, $B$ and $L$ of P3D. Table shows the total number of weights, the number of filters within the convolutional encoder/decoder and the number of groups for GroupNormalization layers.

| Configuration | Number of parameters | Embedding dimensions | Number of groups |
|---|---|---|---|
| S | 11.2M | [32, 32, 64] | 16 |
| B | 46.2M | [64, 128, 128] | 32 |
| L | 181M | [128, 256, 256] | 32 |

## A.6  BASELINE ARCHITECTURES

**Performance comparison of baseline architectures**  Below, we summarize the numbers of the different architectures used in the experiments for an input of size $128^3$ with a single channel, see Table A2.

Table A2: Performance comparison of architectures.

| Model | Params | GFLOPS | Memory | Throughput |
|---|---|---|---|---|
| Swin3D | 50.3M | 289.6 | 2.9GB | 9.86it/s |
| FactFormer | 5.0M | - | 20.4GB | 0.82it/s |
| UNet$_{\text{GenCFD}}$ | 100.0M | 11039.6 | 4.8GB | 2.03it/s |
| UNet$_{\text{ConvNext}}$ | 9.2M | 335.6 | 2.4GB | 11.9it/s |
| TFNO | 75.6M | 139.4 | 5.1GB | 4.7it/s |
| AViT | 60.0M | 143.0 | 0.3GB | 188.3it/s |
| AFNO | 64.1M | 2116.4 | 0.4GB | 31.7it/s |
| P3D-$S$ | 11.2M | 238.7 | 0.9GB | 35.2it/s |
| P3D-$B$ | 46.1M | 2330.6 | 2.1GB | 10.0it/s |
| P3D-$L$ | 181.2M | 9276.2 | 4.6GB | 3.7it/s |

Regarding the number of parameters, GFLOPs, memory and throughput, the three configurations $S$, $B$ and $L$ of P3D are well positioned compared to the baselines we chose. Timings were obtained on a RTX A5000 GPU with batch size 1. Importantly, across all tasks, P3D outperforms the baselines in terms of accuracy. We fixed the training setup for all architectures, using learning rate/optimizer/batch size that are common for training large transformer and UNet models (Esser et al., 2024). Due to the number and size of the different models, we do not perform any hyperparameter tuning for individual models.

**Swin3D**  Our implementation of extends the SwinV2 (Liu et al., 2021) transformer in 3D. The code is based `Swinv2PreTrainedModel` in the Python `transformers` package, with pixel shuffling in 3D for upsampling. For the $B$ (default) configuration of Swin3D, we use a token embedding dimension of 96 with patch size $p = 2$, four stages with two layers each, window size $w = 4$ and 12 attention heads for each stage.

**FactFormer** We use the official code repository of FactFormer `https://github.com/BaratiLab/FactFormer`. We adapt the `FactFormer3D` module used for the 3D Navier-Stokes experiment (Li et al., 2023b). Hyperparameters are taken from the configuration file in the repository: `dim=128`, `depth=4`, `dim_head=64`, `heads=6`.

**UNet$_{\mathrm{GenCFD}}$** We use the official code repository for UNet$_{\mathrm{GenCFD}}$ `https://github.com/camlab-ethz/GenCFD`. We remove any modifications of the core UNet structure specific to the diffusion training setup in Molinaro et al. (2024). Hyperparameters are taken from the repository: `num_channels=[128,256,256]`, `downsample_ratio=[2,2,2]`, `num_blocks=4`, `padding_method='circular'`, `use_position_encoding=True` and `num_heads=8`.

**UNet$_{\mathrm{ConvNeXt}}$** We use the implementation of UNet$_{\mathrm{ConvNeXt}}$ with ConvNeXt blocks provided by the GitHub repository `https://github.com/PolymathicAI/the_well`. This architecture was used as a benchmark in Ohana et al. (2024). Hyperparameters are kept the same: `spatial_resolution=[128,128,128]`, `stages=4`, `blocks_per_stage=1`, `blocks_per_neck=1`, `init_features=32`.

**TFNO** We use the implementation of TFNO (Kossaifi et al., 2023) from the Python `neuralop` package `https://github.com/neuraloperator/neuraloperator`. Hyperparameters are: `spatial_resolution=[128,128,128]`, `modes_1=16`, `modes_2=16`, `modes_3=16`, `hidden_channels=64`.

**AViT** We use the implementation of axial vision transformers (Ho et al., 2019) provided by the GitHub repository `https://github.com/PolymathicAI/the_well`. This architecture was used as a benchmark in Ohana et al. (2024). Hyperparameters for the $B$ configuration are kept the same: `spatial_resolution=[128,128,128]`, `hidden_dim=768`, `num_heads=12`, `processor_blocks=8`.

**AFNO** We use the implementation of AFNO (Guibas et al., 2021) provided by the GitHub repository `https://github.com/PolymathicAI/the_well`. This architecture was used as a benchmark in Ohana et al. (2024). Hyperparameters for the $B$ configuration are kept the same: `spatial_resolution=[128,128,128]`, `hidden_dim=768`, `n_blocks=12`, `cmlp_diagonal_blocks=8` and `patch_size=8`.

# B    EXPERIMENT 1: JOINTLY LEARNING PDEs

The datasets used in this experiment were carefully selected for a diverse list of partial differential equations (PDEs) with a focus on high spatial resolution. This is combined with variations of the physical parameters of the PDEs and different initial conditions, creating diverse dynamics across the different simulations for each type of PDE.

The dataset encompasses linear, reaction-diffusion, and nonlinear PDEs. We utilized the *Exponax* solver, as detailed by Koehler et al. (2024) in the APEBench benchmark. The solver utilizes Exponential Time Differencing Runge-Kutta (ETDRK) methods. We intentionally opted not to use the APEBench dataset directly from the original authors. This decision was driven by our goal to create datasets with enhanced resolution and greater diversity in the underlying physical behaviors, rather than relying solely on variations in initial conditions as done in APEBench. It is a characteristic of the ETDRK methods that they operate within the Fourier domain. Consequently, their application is limited to scenarios with periodic domains and cannot accommodate complex boundary conditions.

Table B1: Summary of datasets produced for the joint PDE learning task, covering linear, reaction-diffusion, and nonlinear PDEs. The table details the dimensions of each dataset: number of simulations (s), time steps (t), fields/channels (f), and spatial dimensions (x, y, z). Beyond the explicitly varied quantities listed for each dataset, the initial conditions for every simulation (s) are also distinct.

| **Dataset** | s | t | f | x | y | z | Varied Quantities across s | Test Set |
|---|---|---|---|---|---|---|---|---|
| hyp | 60 | 30 | 1 | 384 | 384 | 384 | hyper-diffusivity | $s \in [50, 60[$ |
| fisher | 60 | 30 | 1 | 384 | 384 | 384 | diffusivity, reactivity | $s \in [50, 60[$ |
| sh | 60 | 30 | 1 | 384 | 384 | 384 | reactivity, critical number | $s \in [50, 60[$ |
| gs-alpha | 10 | 30 | 2 | 320 | 320 | 320 | initial conditions only | separate: s=3, t=100 |
| gs-beta | 10 | 30 | 2 | 320 | 320 | 320 | initial conditions only | separate: s=3, t=100 |
| gs-gamma | 10 | 30 | 2 | 320 | 320 | 320 | initial conditions only | separate: s=3, t=100 |
| gs-delta | 10 | 30 | 2 | 320 | 320 | 320 | initial conditions only | $s \in [8, 10[$ |
| gs-epsilon | 10 | 30 | 2 | 320 | 320 | 320 | initial conditions only | separate: s=3, t=100 |
| gs-theta | 10 | 30 | 2 | 320 | 320 | 320 | initial conditions only | $s \in [8, 10[$ |
| gs-iota | 10 | 30 | 2 | 320 | 320 | 320 | initial conditions only | $s \in [8, 10[$ |
| gs-kappa | 10 | 30 | 2 | 320 | 320 | 320 | initial conditions only | $s \in [8, 10[$ |
| burgers | 60 | 30 | 3 | 384 | 384 | 384 | viscosity | $s \in [50, 60[$ |
| kdv | 60 | 30 | 3 | 384 | 384 | 384 | domain extent, viscosity | $s \in [50, 60[$ |
| ks | 60 | 30 | 1 | 384 | 384 | 384 | domain extent | separate: s=5, t=200 |

## B.1    DATA GENERATION SETUP

A key aspect of the simulations, in addition to parameter variation, is the use of randomized initial conditions. The standard approach for constructing these conditions involves randomly selecting one of three initialization methods, each providing a unique spectral energy distribution. The first method, a random truncated Fourier series initializer, involves summing multiple Fourier series up to a cutoff frequency, chosen as a uniformly random integer between 2 and 10 (exclusive of 11). The second, the Gaussian random field initializer, produces a power-law spectrum in Fourier space, where energy diminishes polynomially with the wavenumber; its exponent is uniformly randomly selected from [2.3,3.6[. The third method, the diffused noise initializer, generates a tensor of values from normally distributed white noise, subsequently diffusing it. This results in a spectrum that decays exponentially quadratically, with an intensity rate uniformly random from [0.00005,0.01[. After generation, all initializers ensure the initial conditions' values are normalized to a maximum absolute value of one. For vector quantities, the randomly chosen initializer is applied independently to each component.

## B.2    PDE TYPES

We make use of Exponential Time Differencing Runge-Kutta (ETDRK) methods to efficiently simulate different PDEs via Exponax. While the chosen linear PDEs are simple and analytically

solvable, the underlying dynamics are essential for more complicated PDEs. These linear PDEs can be understood as representing a scalar attribute, such as density. Unless stated otherwise, sampling from intervals is consistently performed using a uniform random distribution. In the following, we describe a range of different three-dimensional PDE problems that are employed in our experiments ranging from linear, over reaction-diffusion, to non-linear PDEs. The class of non-linear PDEs is particularly challenging, as these cases more closely resemble real-world problems. Table B2 gives an overview of the different PDE equations and dynamics.

**Hyper-Diffusion (`hyp`)** behaves similarly to diffusion, where density dissipates inside a periodic domain due to the effects of hyper-diffusion. Unlike diffusion, hyper-diffusion does not treat all wavelengths equally, analogous to the relation between dispersion and advection. Higher frequency components are damped even more aggressively compared to normal diffusion, leading to visually stronger blur effect of the density field over time.

- Dimensionality: $s = 60$, $t = 30$, $f = 1$, $x = 384$, $y = 384$, $z = 384$
- Initial Conditions: random truncated Fourier / Gaussian random field / diffused noise
- Boundary Conditions: periodic
- Time Step of Stored Data: 0.01
- Spatial Domain Size of Simulation: $[0, 1] \times [0, 1]$
- Fields: density
- Varied Parameters: hyper-diffusivity $\zeta \in [0.00005, 0.0005[$
- Validation Set: random $15\%$ split of all sequences from $s \in [0, 50[$
- Test Set: all sequences from $s \in [50, 60[$

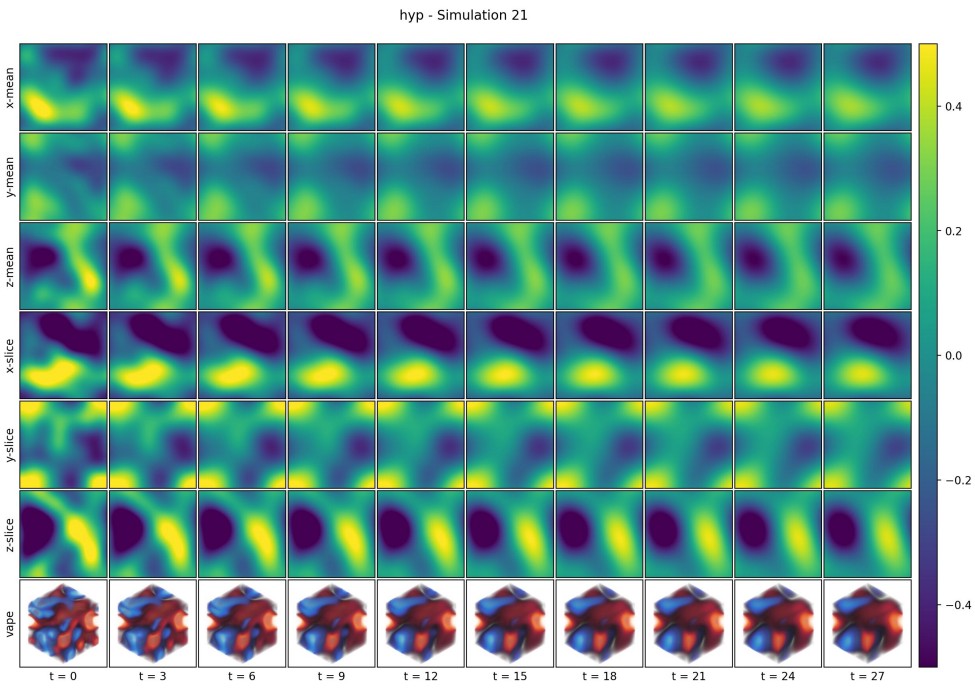

Figure B1: Random example simulation from `hyp`.

Table B2: Overview of the PDE dynamics. Table adapted from Koehler et al. (2024).

| Name | Dynamics | PDE | Coefficients |
|---|---|---|---|
| Hyper-Diffusion | `hyp` | $\frac{\partial u}{\partial t} = -\zeta((\nabla \odot \nabla) \cdot (\nabla \odot \nabla))u$ | $\zeta$ varied |
| Fisher-KPP | `fisher` | $\frac{\partial u}{\partial t} = \nu(\nabla \cdot \nabla)u + ru(1-u)$ | $\nu$ varied 
 $r$ varied |
| Swift-Hohenberg | `sh` | $\frac{\partial u}{\partial t} = ru - (k+\nabla)^2 u + u^2 - u^3$ | $r$ varied 
 $k$ varied |
| Gray-Scott | `gs-alpha gs-beta` 
 `gs-delta gs-iota` 
 `gs-theta gs-theta` 
 `gs-epsilon gs-kappa` | $\frac{u_0}{\partial t} = \nu_0 \nabla \cdot \nabla u_0 - u_0 u_1^2 + f(1-u_0)$ 
 $\frac{u_1}{\partial t} = \nu_1 \nabla \cdot \nabla u_1 + u_0 u_1^2 - f(f+k)u_1$ | $\nu_0 = 0.002$ 
 $\nu_1 = 0.001$ 
 $f$ varied 
 $k$ varied |
| Burgers | `burgers` | $\frac{\partial \mathbf{u}}{\partial t} = -\frac{1}{2}\nabla \cdot (\mathbf{u} \otimes \mathbf{u}) + \nu(\nabla \cdot \nabla)\mathbf{u}$ | $\nu$ varied |
| Korteweg-de-Vries | `kdv` | $\frac{\partial \mathbf{u}}{\partial t} = -\frac{1}{2}b\nabla \cdot (\mathbf{u} \otimes \mathbf{u}) - a\vec{1} \cdot (\nabla \odot \nabla \odot (\nabla \mathbf{u})) + \nu(\nabla \odot \nabla)\mathbf{u} - \zeta((\nabla \odot \nabla) \cdot (\nabla \odot \nabla))\mathbf{u}$ | $b = -6.0$ 
 $a = 1.0$ 
 $\nu$ varied 
 $\zeta = 0.01$ |
| Kuramoto-Sivashinsky | `ks` | $\frac{\partial u}{\partial t} = -b\frac{1}{2}\|\nabla u\|_2^2 + \nu(\nabla \odot \nabla)u - \zeta((\nabla \odot \nabla) \cdot (\nabla \odot \nabla))u$ | $b = 1.0$ 
 $\nu = 1.0$ 
 $\zeta = 1.0$ |

**Fisher-KPP (`fisher`)** is a foundational reaction-diffusion PDE. These systems are used to model biological or geological processes, often resulting in pattern formation. This equation details how the concentration of a substance varies over time and space, influenced by a reaction process governed by a reactivity parameter, and its dispersal through diffusion, which is defined by a diffusivity parameter. Its applications extend to various domains, including wave propagation, population dynamics, ecology, and plasma physics. Figure B2 shows example visualizations from `fisher`.

- Dimensionality: $s = 60$, $t = 30$, $f = 1$, $x = 384$, $y = 384$, $z = 384$
- Initial Conditions: random truncated Fourier / Gaussian random field / diffused noise (with clamping to $[0, 1]$)
- Boundary Conditions: periodic
- Time Step of Stored Data: 0.005
- Spatial Domain Size of Simulation: $[0, 1] \times [0, 1]$
- Fields: concentration
- Varied Parameters: diffusivity $\nu \in [0.0001, 0.02[$ and reactivity $r \in [5, 15[$
- Validation Set: random 15% split of all sequences from $s \in [0, 50[$
- Test Set: all sequences from $s \in [50, 60[$

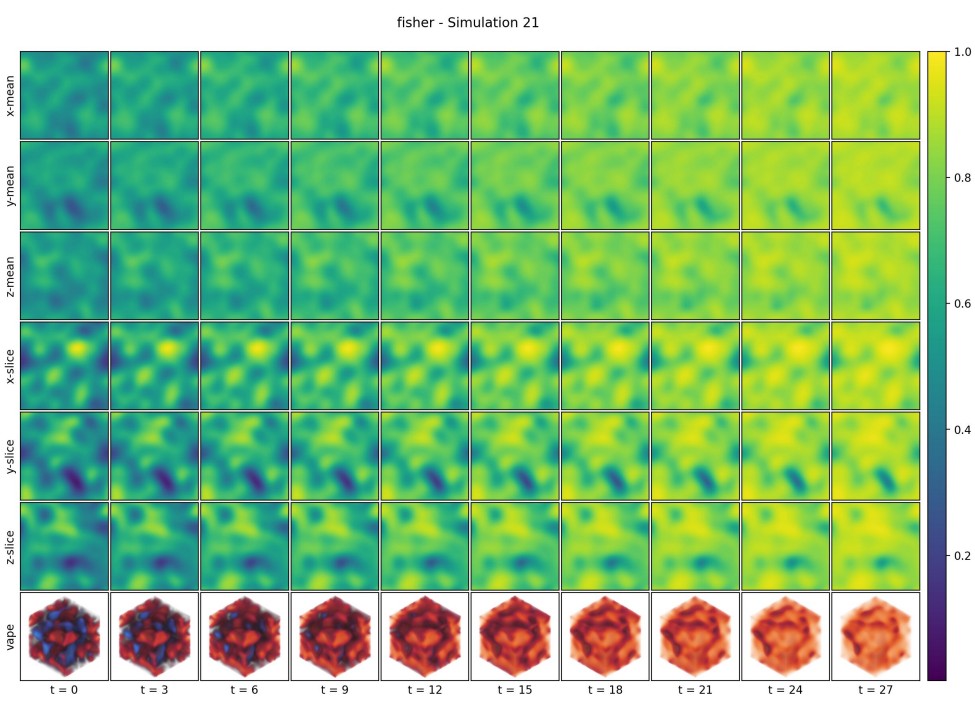

Figure B2: Random example simulation from `fisher`.

**Swift-Hohenberg (`sh`)**   is known for depicting various pattern formation processes. This equation can be applied to illustrate the structure of wrinkles in curved elastic bilayer materials. A prime example is the formation of human fingerprints, where tensions between skin layers generate their unique wrinkling. Figure B3 shows example visualizations from `sh`.

- Dimensionality: $s = 60$, $t = 30$, $f = 1$, $x = 384$, $y = 384$, $z = 384$
- Initial Conditions: random truncated Fourier / Gaussian random field / diffused noise
- Boundary Conditions: periodic
- Time Step of Stored Data: 0.5 (with 5 substeps for the simulation)
- Spatial Domain Size of Simulation: $[0, 20\pi] \times [0, 20\pi]$
- Fields: concentration
- Varied Parameters: reactivity $r \in [0.4, 1[$ and critical number $k \in [0.8, 1.2[$
- Validation Set: random $15\%$ split of all sequences from $s \in [0, 50[$
- Test Set: all sequences from $s \in [50, 60[$

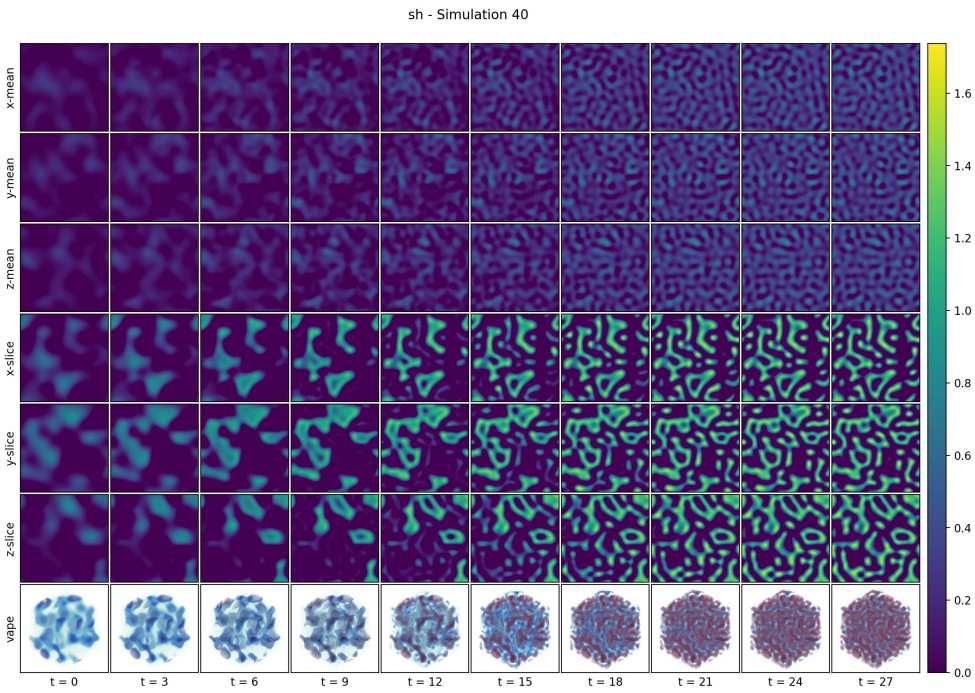

Figure B3: Random example simulation from `sh`.

**Gray-Scott (`gs`)** illustrates the dynamic interplay of two reacting and diffusing chemical substances. Substance $s_a$ with concentration $c_a$ is depleted through reaction but resupplied based on a defined feed rate. Meanwhile, substance $s_b$, the reaction's product with concentration $c_b$, is eliminated from the domain at a given kill rate. The balance between these two rates profoundly influences the simulation outcomes, leading to diverse stable or evolving patterns. We simulate several cases: four with temporally steady configurations, which result in a state that does not substantially change anymore (gs-delta, gs-theta, gs-iota, and gs-kappa), and four temporally unsteady configurations, which continuously evolve over time (gs-alpha, gs-beta, gs-gamma, and gs-epsilon). For the unsteady case, separate test sets with longer temporal rollouts are created. Figure B4 shows example visualizations from the steady configurations, and Figure B5 from the unsteady configurations and corresponding test sets. For further details, see (Pearson, 1993).

For all simulations, the diffusivity of the substances is fixed to $d_a = 0.00002$ and $d_b = 0.00001$. In addition to that, datasets are initialized with a Gaussian blob initializer. The initializer creates four Gaussian blobs at random positions and variances in the center 60% (20% for gs-kappa) of the domain, where the initialization of $c_a$ is the complement of $c_b$, i.e. $c_a = 1 - c_b$.

**Steady Configurations (`gs-delta, gs-theta, gs-iota, and gs-kappa`):**
- Dimensionality: $s = 10$, $t = 30$, $f = 2$, $x = 320$, $y = 320$, $z = 320$ (per configuration)
- Initial Conditions: random Gaussian blobs
- Boundary Conditions: periodic
- Time Step of Simulation: 1.0 (all configurations)
- Time Step of Stored Data:
    - gs-delta: 130.0
    - gs-theta: 200.0
    - gs-iota: 240.0
    - gs-kappa: 300.0
- Number of Warmup Steps (discarded, in time step of data storage):
    - gs-delta: 0
    - gs-theta: 0
    - gs-iota: 0
    - gs-kappa: 15
- Spatial Domain Size of Simulation: $[0, 2.5] \times [0, 2.5]$
- Fields: concentration $c_a$, concentration $c_b$
- Varied Parameters: feed rate and kill rate determined by configuration (i.e., initial conditions only within configuration)
    - gs-delta: feed rate: 0.028, kill rate: 0.056
    - gs-theta: feed rate: 0.040, kill rate: 0.060
    - gs-iota: feed rate: 0.050, kill rate: 0.0605
    - gs-kappa: feed rate: 0.052, kill rate: 0.063
- Validation Set: random 15% split of all sequences from $s \in [0, 8[$
- Test Set: all sequences from $s \in [8, 10[$

**Unsteady Configurations (`gs-alpha, gs-beta, gs-gamma, and gs-epsilon`):**
- Dimensionality: $s = 10$, $t = 30$, $f = 2$, $x = 320$, $y = 320$, $z = 320$ (per configuration)
- Initial Conditions: random Gaussian blobs
- Boundary Conditions: periodic
- Time Step of Simulation: 1.0 (all configurations)
- Time Step of Stored Data:
    - gs-alpha: 30.0
    - gs-beta: 30.0
    - gs-gamma: 75.0

- – `gs-epsilon`: 15.0
- Number of Warmup Steps (discarded, in time step of data storage):
  - – `gs-alpha`: 75
  - – `gs-beta`: 50
  - – `gs-gamma`: 70
  - – `gs-epsilon`: 300
- Spatial Domain Size of Simulation: $[0, 2.5] \times [0, 2.5]$
- Fields: concentration $c_a$, concentration $c_b$
- Varied Parameters: feed rate and kill rate determined by configuration (i.e., initial conditions only within configuration)
  - – `gs-alpha`: feed rate: 0.008, kill rate: 0.046
  - – `gs-beta`: feed rate: 0.020, kill rate: 0.046
  - – `gs-gamma`: feed rate: 0.024, kill rate: 0.056
  - – `gs-epsilon`: feed rate: 0.020, kill rate: 0.056
- Validation Set: random $15\%$ split of all sequences from $s \in [0, 10[$
- Test Set: separate simulations with $s = 30$, $t = 100$, $f = 2$, $x = 320$, $y = 320$, $z = 320$ (per configuration)

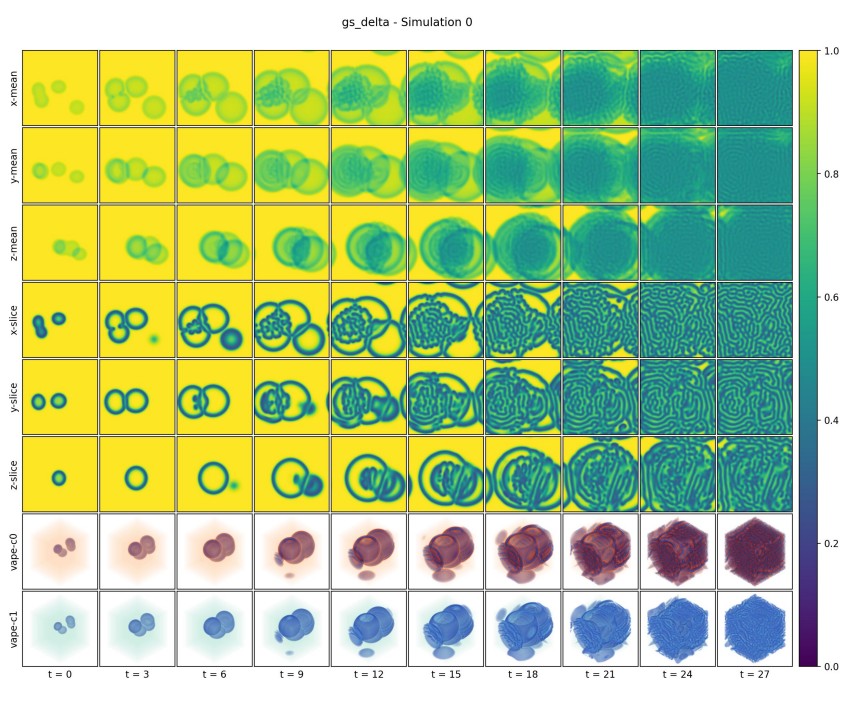

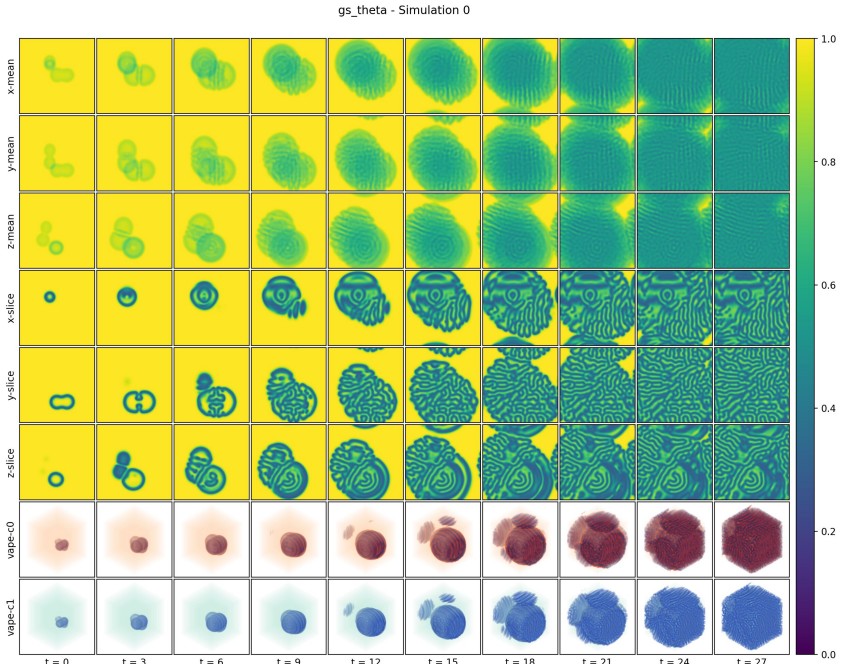

Figure B4: Random example simulations from steady configurations of the Gray-Scott model of a reaction-diffusion system: `gs-delta`, `gs-theta`.

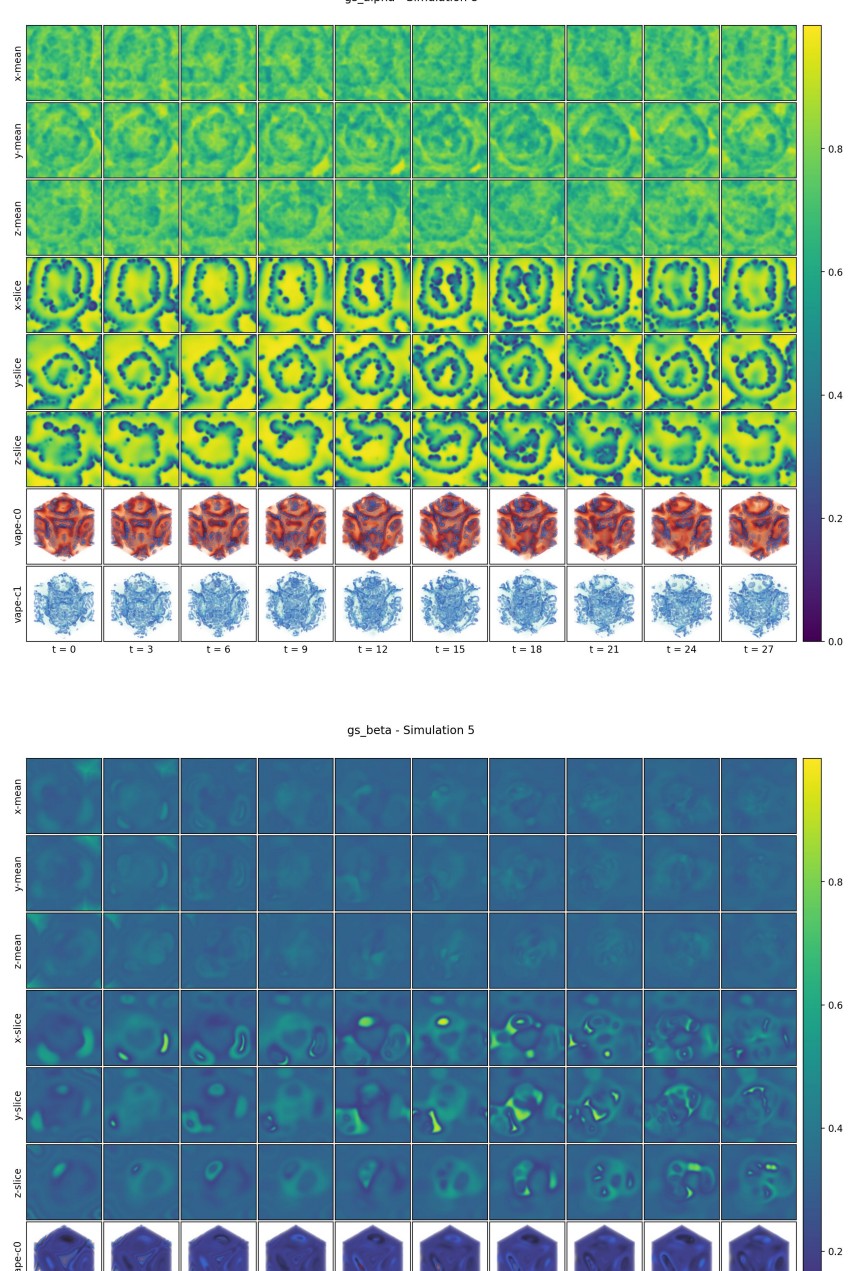

Figure B5: Random example simulations from unsteady configurations of the Gray-Scott model of a reaction-diffusion system: `gs-alpha`, `gs-beta`.

**Burgers (`burgers`)** bears resemblance to an advection-diffusion problem. Instead of modeling the transport of a scalar density, this equation describes how a flow field itself evolves due to the combined effects of advection and diffusion. This process can result in the formation of abrupt discontinuities, often referred to as shock waves, which present a significant challenge for accurate simulation. Burgers' equation also finds utility in fields such as nonlinear acoustics and the modeling of traffic flow. Figure B6 shows example visualizations from `burgers`.

- Dimensionality: $s = 60$, $t = 30$, $f = 3$, $x = 384$, $y = 384$, $z = 384$
- Initial Conditions: random truncated Fourier / Gaussian random field / diffused noise
- Boundary Conditions: periodic
- Time Step of Stored Data: 0.01 (with 50 substeps for the simulation)
- Spatial Domain Size of Simulation: $[0, 1] \times [0, 1]$
- Fields: velocity $(x, y)$
- Varied Parameters: viscosity $\nu \in [0.001, 0.005[$
- Validation Set: random 15% split of all sequences from $s \in [0, 50[$
- Test Set: all sequences from $s \in [50, 60[$

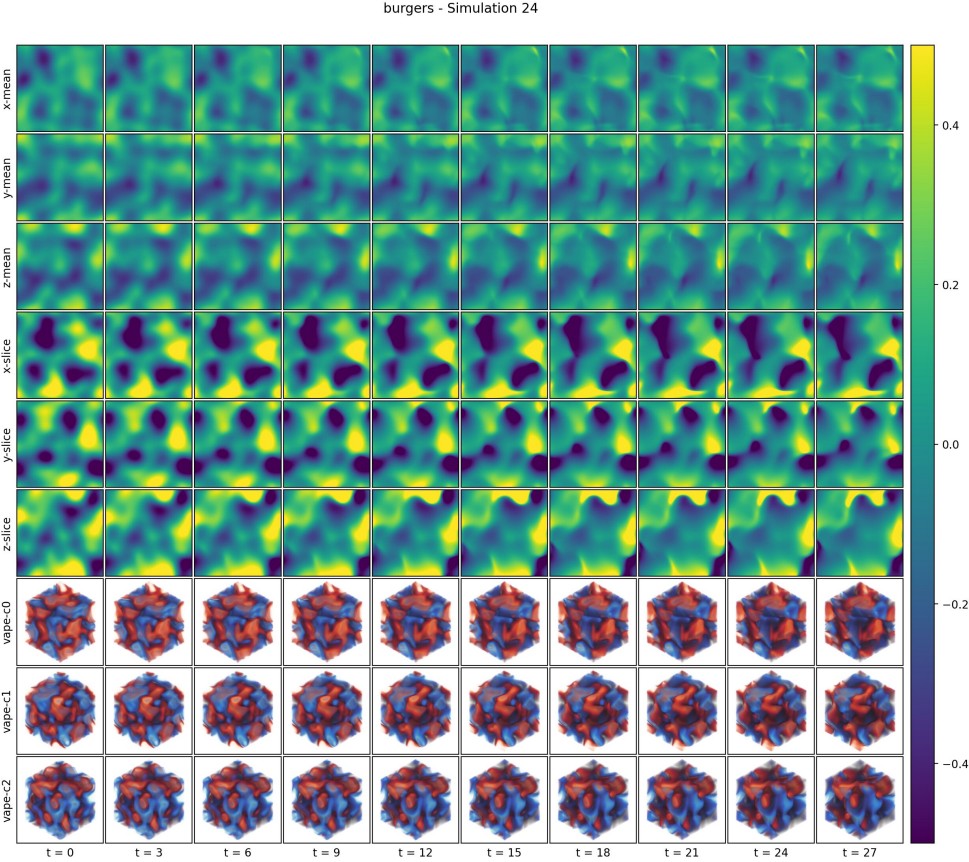

Figure B6: Random example simulation from `burgers`.

**Korteweg-de-Vries (`kdv`)**  presents simulations of the Korteweg-de-Vries equation within a periodic domain. This equation models dispersive, non-dissipative wave propagation and is a classic example of an integrable PDE. It poses a challenge because energy is transferred to high spatial frequencies, resulting in distinct, moving soliton waves that maintain their shape and propagation speed. Throughout these simulations, the convection coefficient remains constant at $-6$, and the dispersivity coefficient is consistently $1$. Figure B7 shows example visualizations from `kdv`.

- Dimensionality: $s = 60$, $t = 30$, $f = 2$, $x = 384$, $y = 384$, $z = 384$
- Initial Conditions: random truncated Fourier / Gaussian random field / diffused noise
- Boundary Conditions: periodic
- Time Step of Stored Data: 0.05 (with 10 substeps for the simulation)
- Spatial Domain Size of Simulation: varied per simulation
- Fields: velocity $(x, y)$
- Varied Parameters: domain extent $\in [30, 120[$ identically for $x, y, z$, i.e. a square domain, and viscosity $\nu \in [0.1, 0.25[$
- Validation Set: random 15% split of all sequences from $s \in [0, 50[$
- Test Set: all sequences from $s \in [50, 60[$

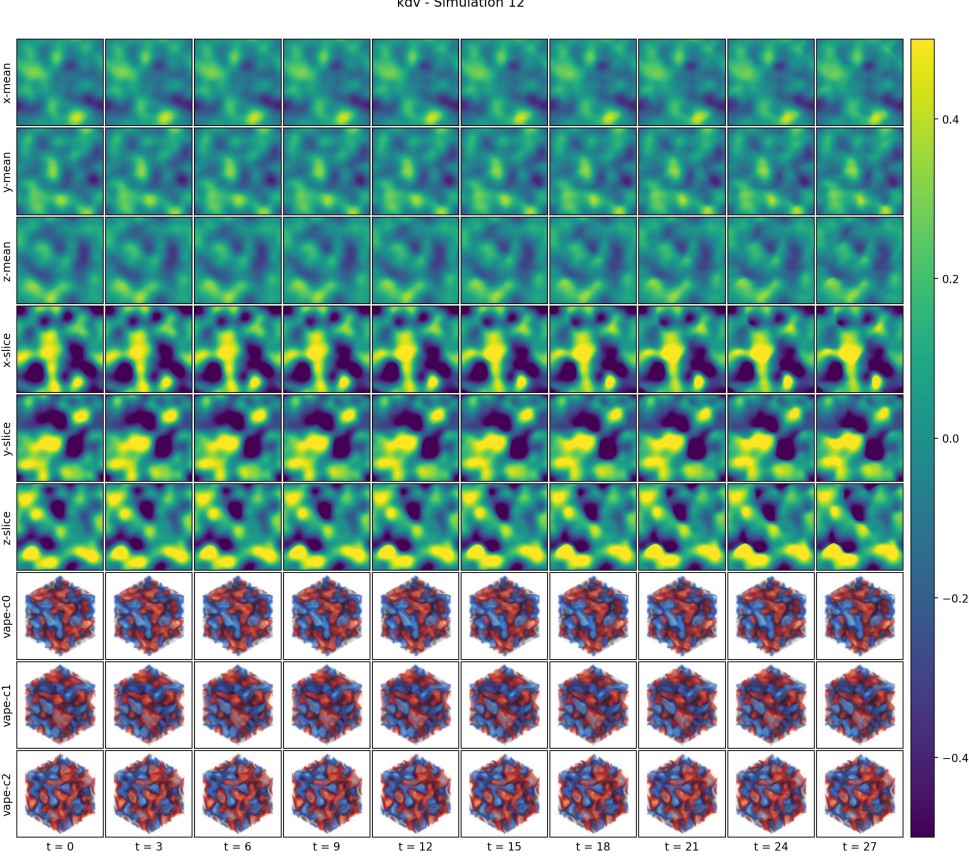

Figure B7: Random example simulation from `kdv`.

**Kuramoto-Sivashinsky (`ks`)** models thermo-diffusive flame instabilities in combustion and also finds use in reaction-diffusion systems on a periodic domain. It's notable for its chaotic behavior, where even slightly different initial conditions can lead to wildly divergent temporal trajectories over time. The initial transient phase of the simulations is not included in the dataset. Figure B8 shows example visualizations from `ks`.

- Dimensionality: $s = 60$, $t = 30$, $f = 1$, $x = 384$, $y = 384$, $z = 384$
- Initial Conditions: random truncated Fourier / Gaussian random field / diffused noise
- Boundary Conditions: periodic
- Time Step of Stored Data: 0.2 (with 2 substeps for the simulation)
- Number of Warmup Steps (discarded, in time step of data storage): 200
- Spatial Domain Size of Simulation: varied per simulation
- Fields: density
- Varied Parameters: domain extent $\in [10, 130[$ identically for $x, y$, i.e. a square domain
- Validation Set: random 15% split of all sequences from $s \in [0, 600[$
- Test Set: separate simulations with $s = 50$, $t = 200$, $f = 1$, $x = 384$, $y = 384$, $z = 384$

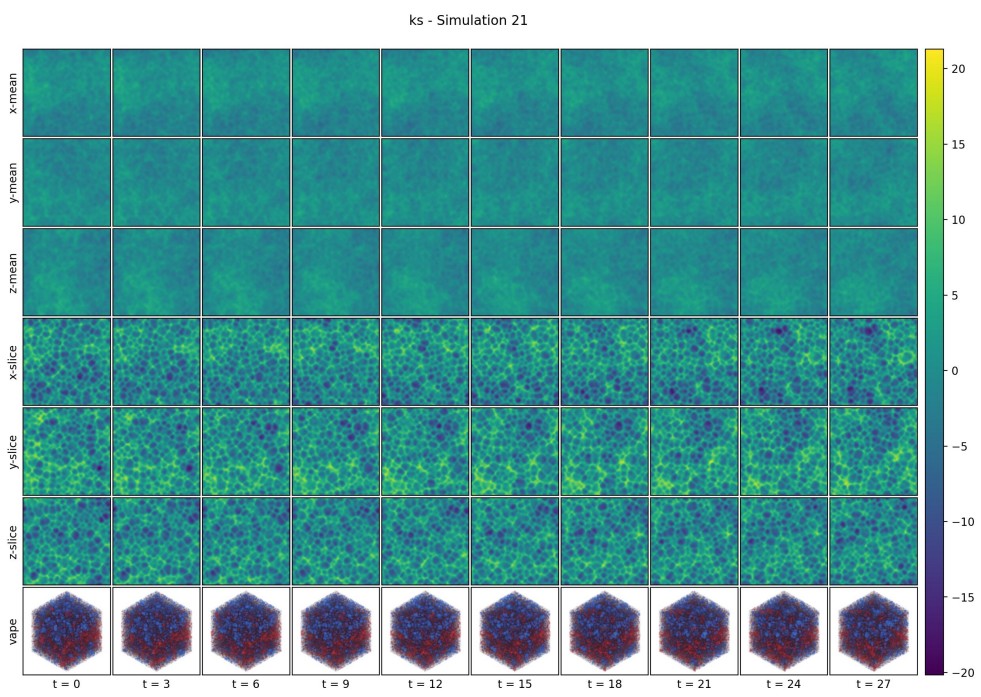

Figure B8: Random example simulation from `ks`.

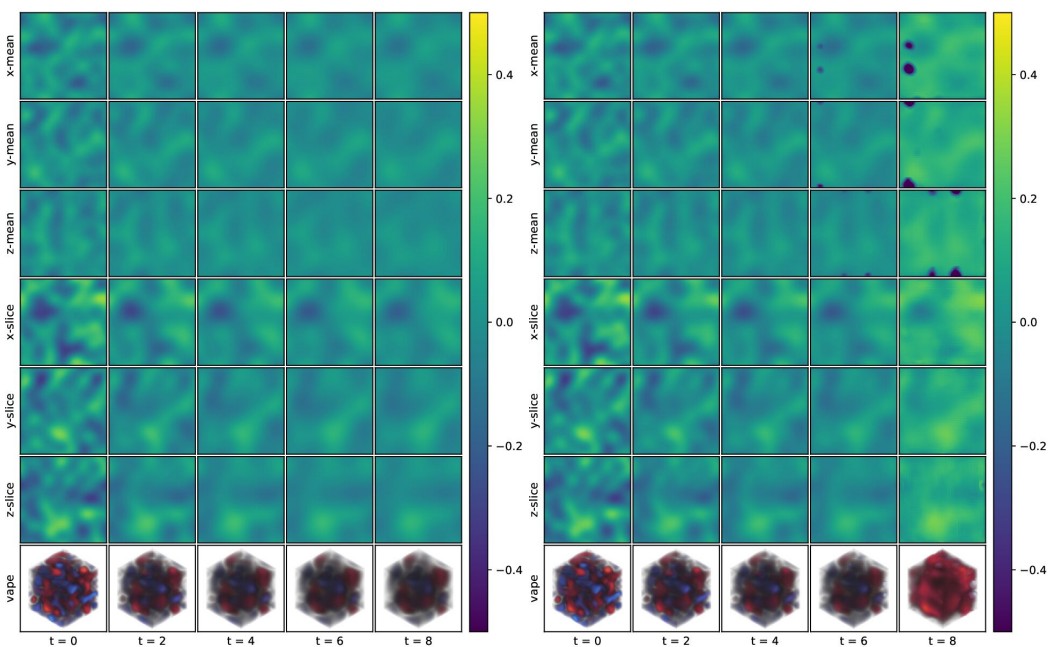

Figure B9: hyp. Reference (left) and autoregressive prediction for $t = 8$ steps with P3D-$L$ <128|384> (right) on the test set at resolution $384^3$.

## B.3 VISUALIZATIONS OF PREDICTIONS

Below, we visualize several example predictions from the P3D-$L$ network trained on crop size $128^3$ on the test datasets. See Figures B9 to B17. During inference, we apply the network to larger crops of domain size $320^3$ for the Gray-Scott PDEs and $384^3$ for all other PDEs. The resolution is significantly higher than what the network was originally trained on. In addition, since the full domain is periodic, we change the padding mode of convolutional layers within the convolutional encoder and decoder from "zeros" to "circular". We consider autoregressive rollouts of up to $t = 8$ steps.

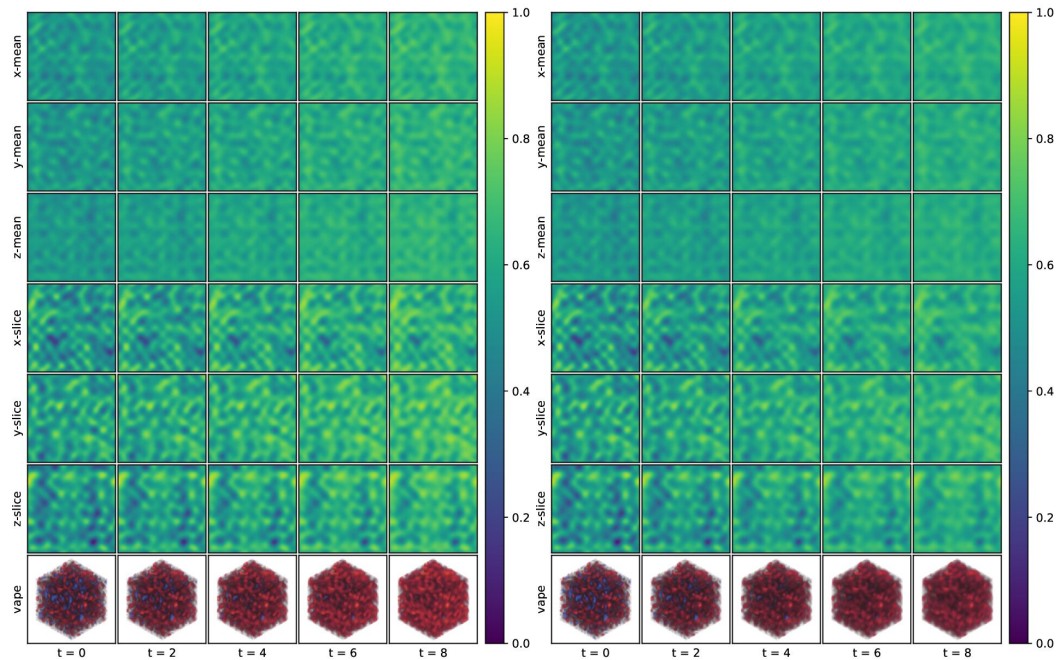

Figure B10: fisher. Reference (left) and autoregressive prediction for $t = 8$ steps with P3D-$L$ <128|384> (right) on the test set at resolution $384^3$.

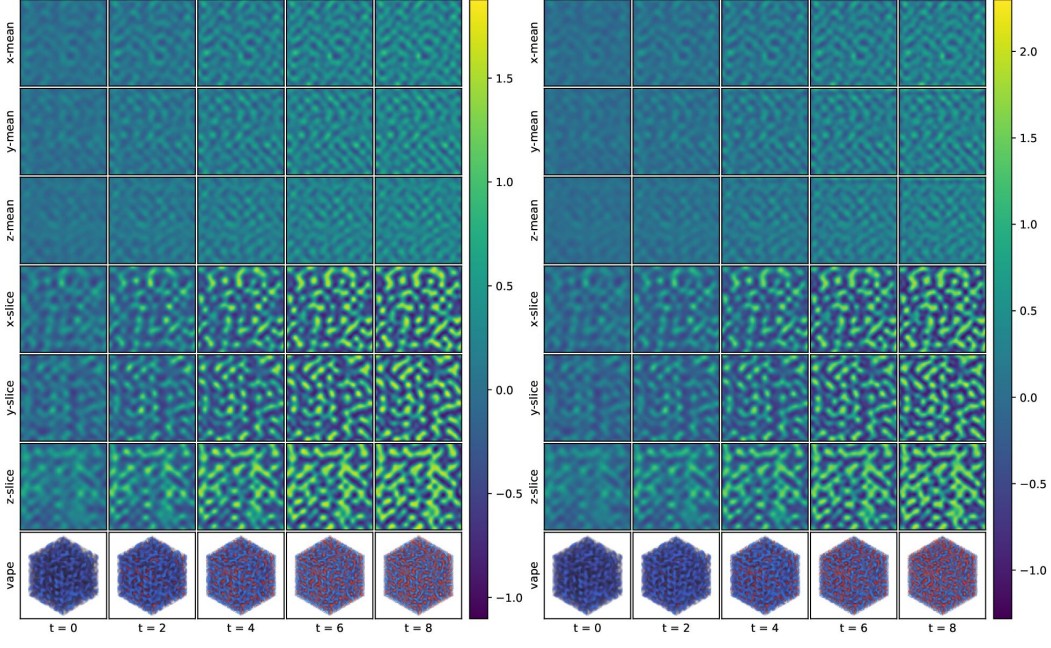

Figure B11: sh. Reference (left) and autoregressive prediction for $t = 8$ steps with P3D-$L$ <128|384> (right) on the test set at resolution $384^3$.

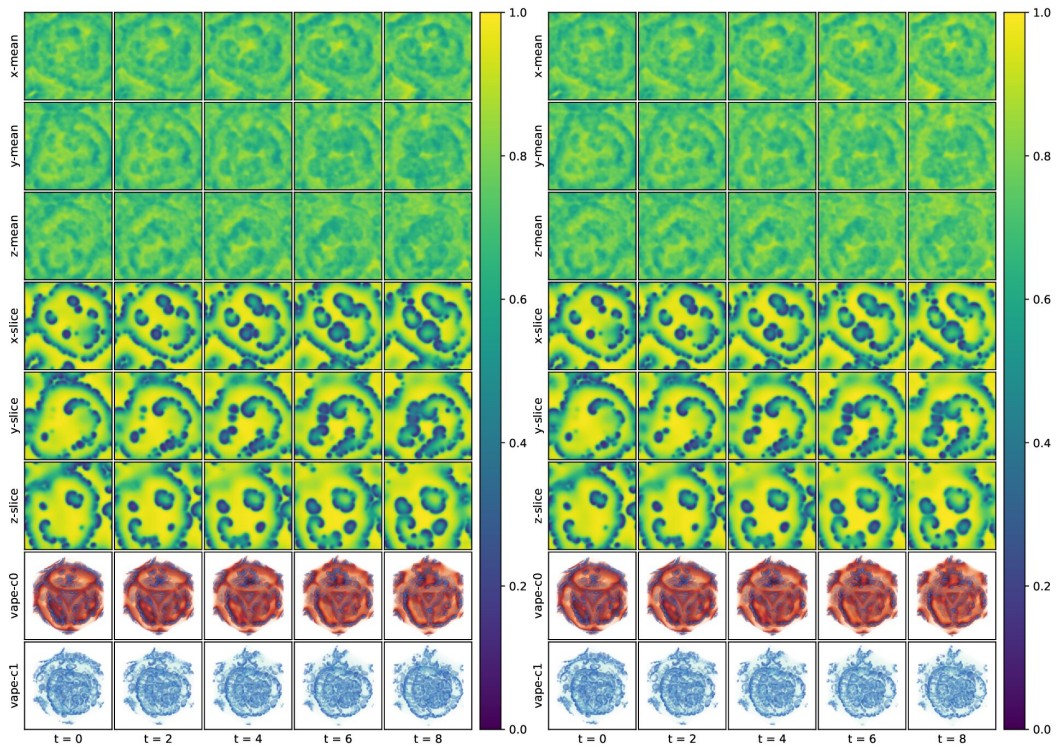

Figure B12: gs-alpha. Reference (left) and autoregressive prediction for $t = 8$ steps with P3D-$L$ <128|320> (right) on the test set at resolution $320^3$.

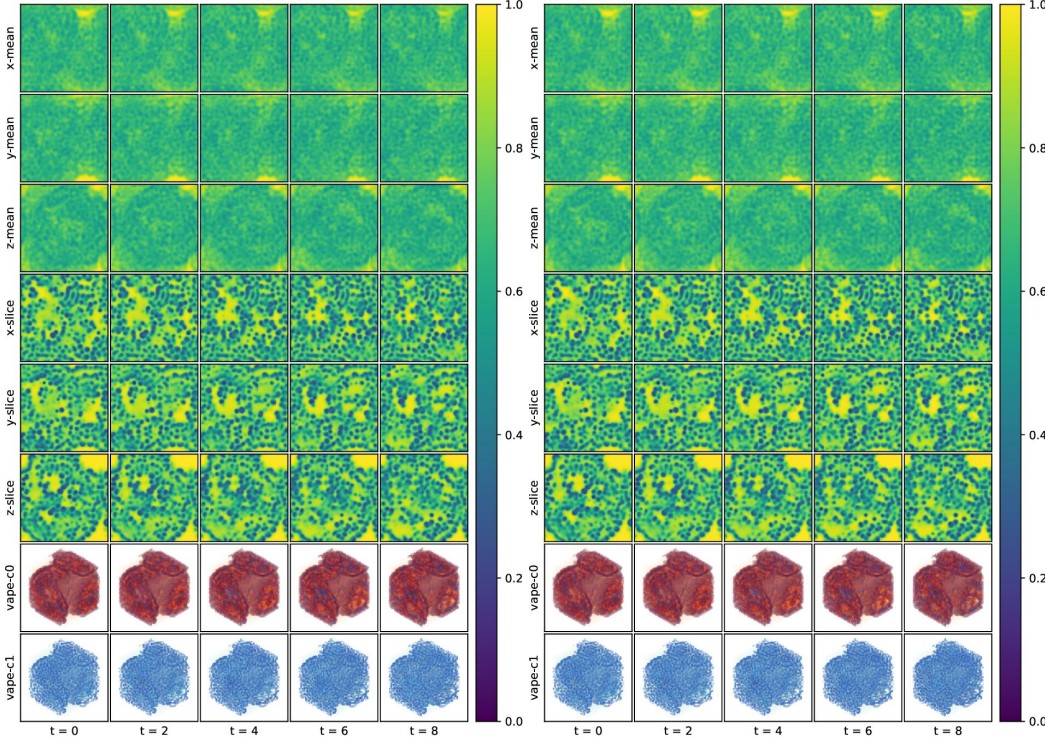

Figure B13: gs-epsilon. Reference (left) and autoregressive prediction for $t = 8$ steps with P3D-$L$ <128|320> (right) on the test set at resolution $320^3$.

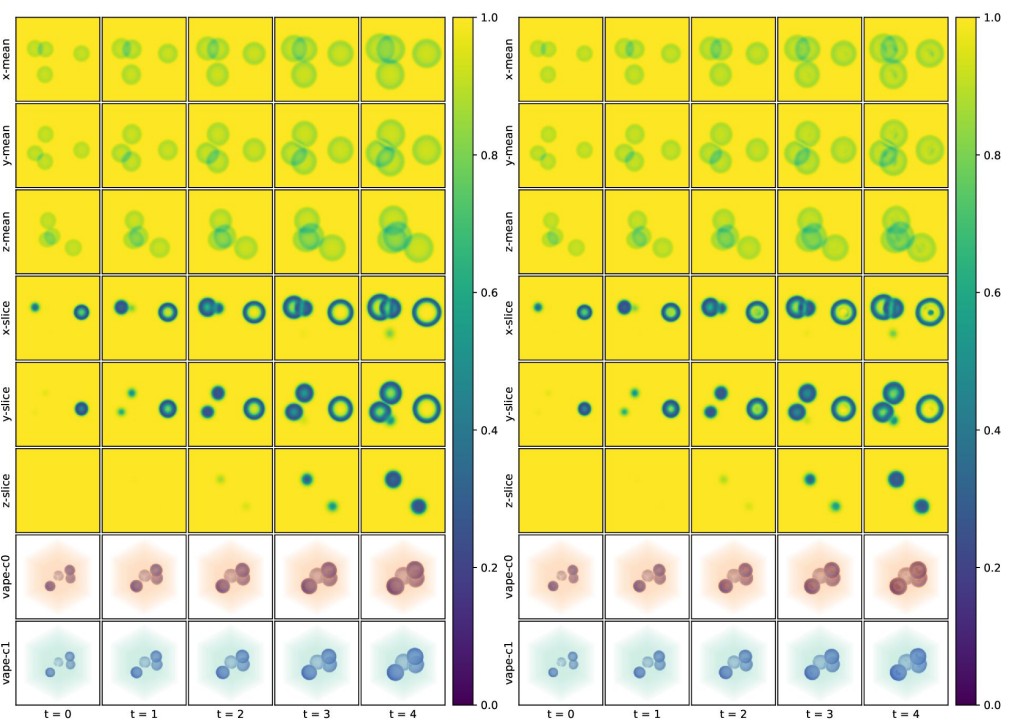

Figure B14: `gs-delta`. Reference (left) and autoregressive prediction for $t = 8$ steps with P3D-$L$ <128|320> (right) on the test set at resolution $320^3$.

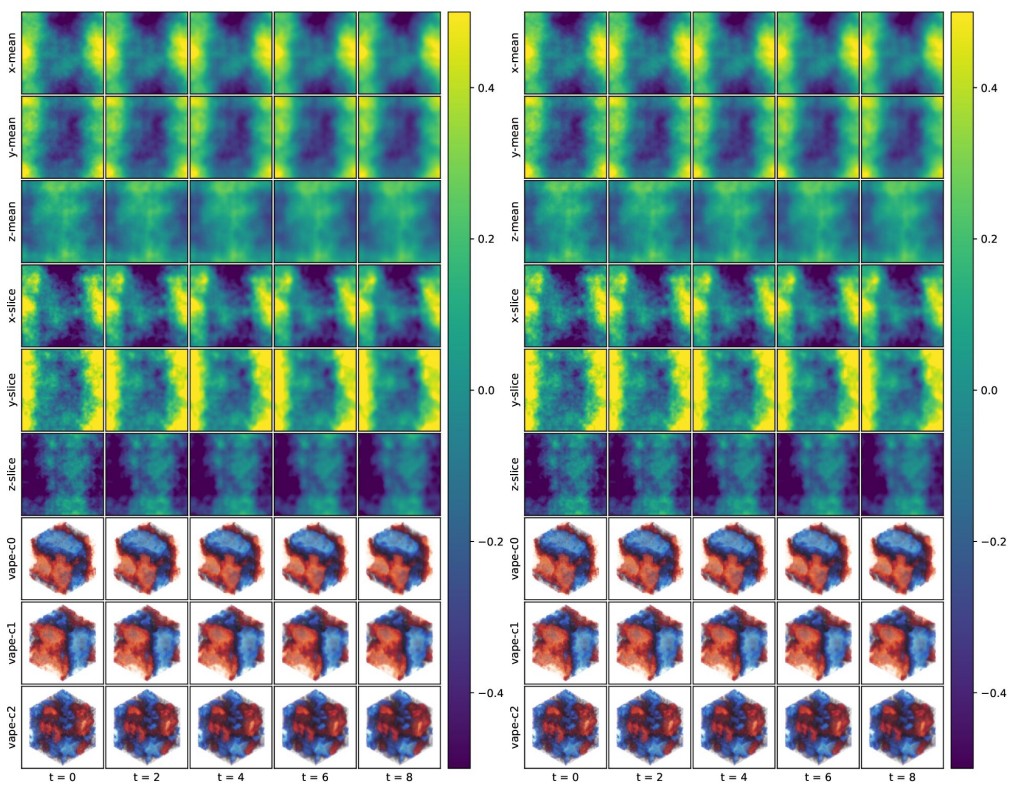

Figure B15: burgers. Reference (left) and autoregressive prediction for $t = 8$ steps with P3D-$L$ <128|384> (right) on the test set at resolution $384^3$.

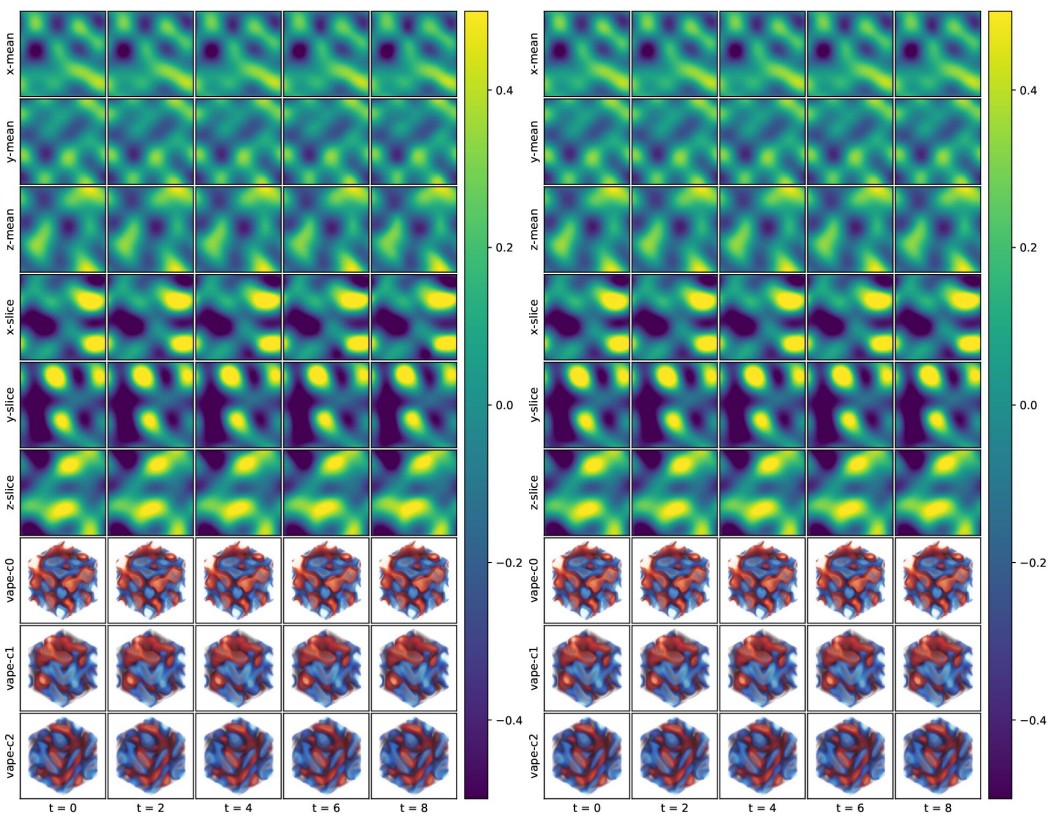

Figure B16: kdv. Reference (left) and autoregressive prediction for $t = 8$ steps with P3D-$L$ <128|384> (right) on the test set at resolution $384^3$.

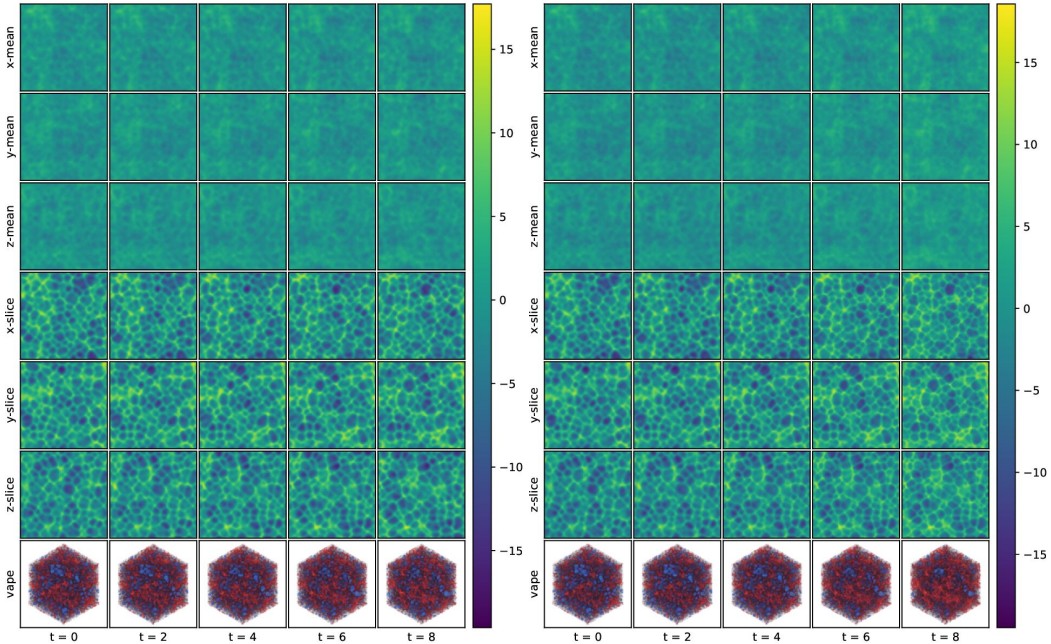

Figure B17: ks. Reference (left) and autoregressive prediction for $t = 8$ steps with P3D-$L$ <128|384> (right) on the test set at resolution $384^3$.

Table B3: Normalized RMSE ($\times 10^{-3}$) for crop size 128.

| Model Name | Fisher | GS $\alpha$ | GS $\beta$ | GS $\delta$ | GS $\epsilon$ | GS $\gamma$ | GS $\iota$ | GS $\kappa$ | GS $\theta$ | Hyp | KDV | KS | SH | Burgers | Average |
|---|---|---|---|---|---|---|---|---|---|---|---|---|---|---|---|
| P3D-$S$ | 8.3 | 15.7 | 29.3 | 14.1 | 13.4 | 47.4 | 12.9 | 16.0 | 6.3 | 73.4 | 92.0 | 18.9 | 91.4 | 27.2 | 33.3 |
| P3D-$B$ | 7.1 | 9.0 | 14.9 | 8.1 | 7.7 | 19.9 | 9.1 | 10.6 | 4.7 | 75.3 | 75.3 | 19.0 | 75.6 | 16.8 | 25.2 |
| P3D-$L$ | 6.9 | 5.7 | 8.9 | 5.7 | 5.4 | 16.0 | 6.0 | 7.1 | 2.3 | 69.1 | 65.8 | 18.4 | 62.7 | 11.1 | 20.8 |
| AFNO | 13.0 | 20.7 | 30.4 | 18.8 | 23.3 | 43.3 | 17.1 | 20.0 | 6.1 | 146.5 | 81.5 | 32.4 | 183.5 | 34.6 | 47.9 |
| AVIT | 124.6 | 49.7 | 60.8 | 51.7 | 62.7 | 101.0 | 47.6 | 48.1 | 56.7 | 619.2 | 191.2 | 65.1 | 470.7 | 177.0 | 151.9 |
| Swin3D | 11.7 | 24.3 | 30.7 | 22.0 | 23.0 | 50.0 | 22.8 | 17.5 | 5.2 | 153.6 | 90.1 | 27.0 | 190.8 | 36.7 | 50.4 |
| UNet$_{\mathrm{GenCFD}}$ | 17.3 | 35.8 | 19.5 | 6.4 | 24.9 | 50.7 | 6.4 | 6.4 | 6.4 | 214.2 | 107.7 | 43.2 | 573.1 | 45.8 | 82.7 |

Table B4: Normalized RMSE ($\times 10^{-3}$) for crop size 64.

| Model Name | Fisher | GS $\alpha$ | GS $\beta$ | GS $\delta$ | GS $\epsilon$ | GS $\gamma$ | GS $\iota$ | GS $\kappa$ | GS $\theta$ | Hyp | KDV | KS | SH | Burgers | Average |
|---|---|---|---|---|---|---|---|---|---|---|---|---|---|---|---|
| P3D-$S$ | 10.4 | 21.0 | 25.4 | 5.2 | 18.1 | 43.3 | 4.6 | 4.6 | 4.6 | 92.2 | 100.4 | 50.5 | 111.8 | 34.4 | 37.6 |
| P3D-$B$ | 8.3 | 11.7 | 10.9 | 4.2 | 10.1 | 26.1 | 3.8 | 3.8 | 3.8 | 83.6 | 92.2 | 42.1 | 98.7 | 26.2 | 30.4 |
| P3D-$L$ | 6.8 | 7.3 | 10.1 | 7.1 | 6.6 | 17.5 | 8.8 | 8.3 | 1.4 | 69.6 | 73.7 | 28.5 | 88.1 | 15.0 | 24.9 |
| AFNO | 19.7 | 20.0 | 22.7 | 3.1 | 21.0 | 38.0 | 2.3 | 2.3 | 2.3 | 189.4 | 90.8 | 39.4 | 210.7 | 36.3 | 49.9 |
| AVIT | 175.6 | 81.9 | 53.6 | 19.5 | 88.3 | 106.0 | 19.0 | 19.0 | 19.0 | 2004.7 | 206.6 | 92.7 | 450.8 | 213.9 | 253.6 |
| Swin3D | 20.7 | 67.4 | 56.9 | 5.0 | 59.7 | 98.4 | 4.5 | 4.5 | 4.5 | 155.6 | 115.9 | 61.2 | 285.3 | 46.2 | 70.4 |
| FactFormer | 19.06 | 18.01 | 13.94 | 3.75 | 16.10 | 30.76 | 3.75 | 3.75 | 3.75 | 182.78 | 95.13 | 31.53 | 184.20 | 40.48 | 46.2 |
| UNet$_{\mathrm{GenCFD}}$ | 36.6 | 35.1 | 72.7 | 14.0 | 42.5 | 65.5 | 13.7 | 13.6 | 13.7 | 272.4 | 128.4 | 37.0 | 324.4 | 56.2 | 80.4 |
| UNet$_{\mathrm{ConvNeXt}}$ | 18.6 | 40.3 | 49.8 | 6.8 | 36.2 | 66.8 | 6.5 | 6.5 | 6.5 | 190.8 | 108.8 | 42.1 | 364.5 | 49.0 | 70.9 |
| TFNO | 23.2 | 111.6 | 56.2 | 6.3 | 49.5 | 96.7 | 6.0 | 6.0 | 6.0 | 175.5 | 113.8 | 222.1 | 263.1 | 36.2 | 83.7 |

Table B5: Normalized RMSE ($\times 10^{-3}$) for crop size 32.

| Model Name | Fisher | GS $\alpha$ | GS $\beta$ | GS $\delta$ | GS $\epsilon$ | GS $\gamma$ | GS $\iota$ | GS $\kappa$ | GS $\theta$ | Hyp | KDV | KS | SH | Burgers | Average |
|---|---|---|---|---|---|---|---|---|---|---|---|---|---|---|---|
| P3D-$S$ | 9.2 | 49.7 | 56.3 | 6.3 | 33.7 | 57.2 | 6.3 | 6.3 | 6.3 | 146.5 | 127.3 | 158.4 | 176.2 | 38.8 | 62.7 |
| P3D-$B$ | 8.3 | 24.5 | 14.8 | 2.5 | 15.9 | 40.1 | 2.5 | 2.5 | 2.5 | 131.1 | 126.4 | 117.5 | 131.7 | 37.9 | 46.9 |
| P3D-$L$ | 7.9 | 18.5 | 11.4 | 1.9 | 14.2 | 32.4 | 1.9 | 1.9 | 1.9 | 109.3 | 110.8 | 110.9 | 129.4 | 27.3 | 41.4 |
| AFNO | 21.7 | 43.8 | 22.0 | 5.8 | 33.4 | 65.9 | 5.8 | 5.8 | 5.8 | 188.7 | 118.3 | 74.0 | 619.5 | 43.1 | 89.5 |
| AVIT | 237.1 | 126.7 | 50.7 | 13.6 | 107.6 | 122.7 | 13.6 | 13.6 | 13.6 | 1089.4 | 256.7 | 177.6 | 476.0 | 230.1 | 209.2 |
| Swin3D | 16.6 | 154.5 | 38.6 | 3.8 | 56.2 | 92.6 | 3.8 | 3.8 | 3.8 | 189.1 | 143.6 | 83.4 | 263.5 | 55.7 | 79.2 |
| FactFormer | 20.1 | 43.1 | 23.1 | 5.8 | 32.0 | 43.9 | 5.8 | 5.8 | 5.8 | 222.9 | 105.0 | 54.4 | 263.7 | 43.3 | 62.5 |
| UNet$_{\mathrm{GenCFD}}$ | 18.5 | 38.6 | 22.1 | 3.5 | 20.9 | 53.6 | 3.5 | 3.5 | 3.6 | 196.3 | 105.8 | 37.0 | 517.2 | 41.9 | 76.1 |
| UNet$_{\mathrm{ConvNeXt}}$ | 20.3 | 111.6 | 38.3 | 7.4 | 49.2 | 89.9 | 7.4 | 7.4 | 7.4 | 166.3 | 128.5 | 77.4 | 442.5 | 48.9 | 85.9 |
| TFNO | 21.9 | 176.8 | 41.2 | 5.6 | 46.4 | 84.4 | 5.6 | 5.6 | 5.6 | 175.3 | 121.8 | 204.1 | 252.4 | 38.3 | 84.7 |

## C EXPERIMENT 2: ISOTROPIC TURBULENCE

For the second experiment, we make use of the Johns Hopkins Turbulence Database (*JHTDB*). It contains data from various direct numerical simulations of homogeneous and wall-bounded turbulent flows (Perlman et al., 2007). The simulations are stored with space-time history and allow for arbitrary spatiotemporal query points.

**Isotropic Turbulence (`iso-turb`)**    a direct numerical simulation of the Navier-Stokes equations at Reynolds number around 433 simulated on a $1024^3$ periodic grid via a pseudo-spectral parallel code. It contains homogeneous isotropic turbulence, i.e., an idealized version of realistic turbulence with statistical properties that are invariant to translations and rotations of the coordinate axes. The following overview summarizes key characteristics of the dataset (for further details see Perlman et al., 2007):

- Dimensionality: $s = 1$, $t = 500$, $f = 4$, $x = 512$, $y = 512$, $z = 512$
- Boundary conditions: periodic
- Time step of stored data: 0.002
- Spatial domain size: $[0, 2\pi] \times [0, 2\pi] \times [0, 2\pi]$
- Fields: velocity X/Y/Z, pressure
- Validation set: random $15\%$ split of all timesteps from $t \in [0, 420]$
- Test set: all sequences from $t \in [420, 500]$

### C.1 EXTENDED EVALUATION

See Table C1 for an extended evaluation expanding upon Table 3. The $B$ config corresponds to the previous size of the baselines models for Swin3D, AViT and AFNO. The $S$ config decreases the hidden dimensionality of the model architecture, which we change from 768 ($B$) to 384 ($S$) for AViT and AFNO. For Swin3D, we decrease it from 96 ($B$) to 48 ($S$). For Swin3D and AViT, the hidden dimensionality corresponds to the token embedding dimension. We also trained an $L$ config for the AViT model with hidden dimensionality 1536. All baseline models were trained for 4000 epochs. Additionally, we kept training P3D-$S$ and P3D-$B$ for a total of 20000 epochs. The evaluation shows that the RMSE and spectral error keep improving, showing no indication of overfitting.

We also include a benchmark comparison regarding the number of parameters, GFLOPs, VRAM and throughput for inference of P3D and the different baseline architectures in Table C2.

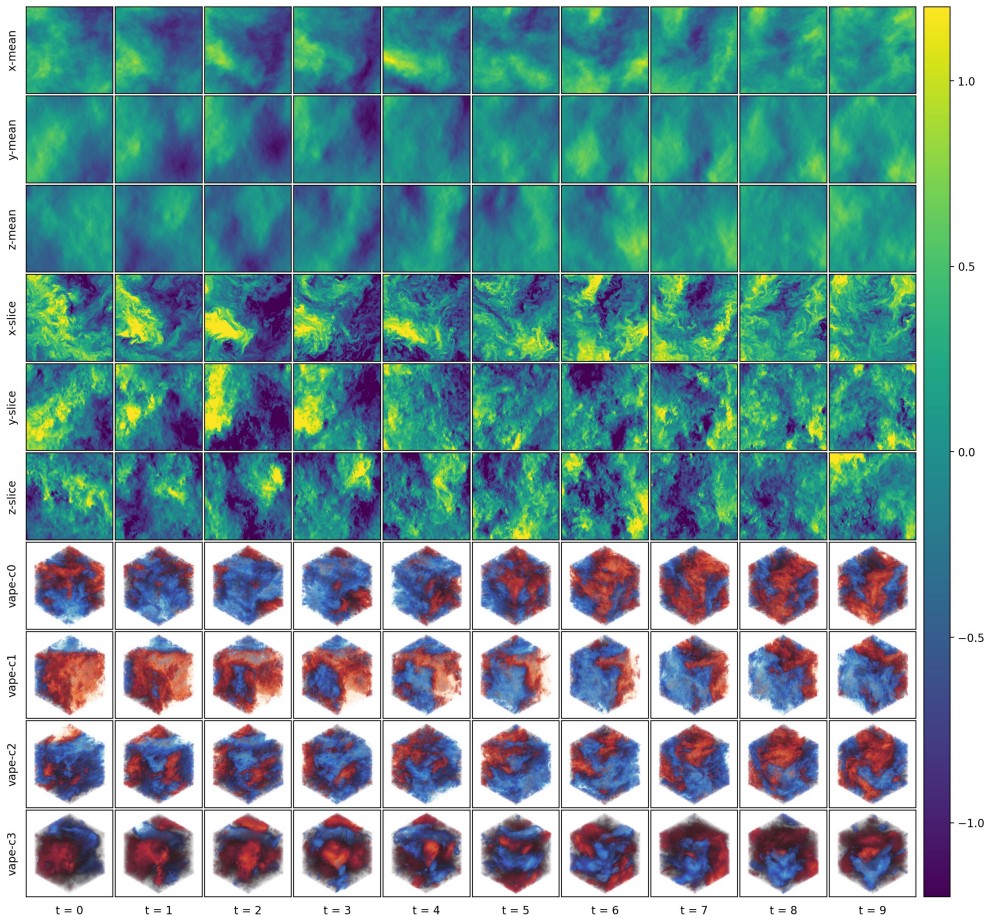

Figure C1: Isotropic Turbulence. Training dataset visualization at resolution $512^3$ showing the velocity X/Y/Z and pressure from $t = 0$ until $t = 420$. The first six rows are 2D mean projections and slices of the velocity in $x$-direction. The 3D renderings show velocity X/Y/Z and pressure.

Table C1: Performance comparison on the test set with crop size $128^3$ for RMSE ($\times 10^{-2}$) and the L2 enstrophy graph error ($\times 10^2$) at different autoregressive rollout steps.

| Model | RMSE | | | | L2 Enstrophy | | | | epochs |
|---|---|---|---|---|---|---|---|---|---|
| | 1 | 5 | 10 | 15 | 1 | 5 | 10 | 15 | |
| UNet$_{\mathrm{GenCFD}}$ | 5.48 | 25.42 | 48.60 | 67.72 | 4.25 | 14.5 | 22.7 | 140 | 4000 |
| Swin3D-$S$ | 4.04 | 11.53 | 18.66 | **24.30** | 6.28 | 59.7 | 120 | 158 | 4000 |
| Swin3D-$B$ | 3.22 | 10.45 | **18.15** | 24.36 | 4.06 | 52.1 | 112 | 156 | 4000 |
| AViT-$S$ | 10.09 | 21.47 | 32.60 | 40.36 | 31.60 | 44.1 | 99.2 | 144 | 4000 |
| AViT-$B$ | 9.45 | 19.57 | 30.00 | 37.77 | 26.70 | 49.1 | 84.6 | 112 | 4000 |
| AViT-$L$ | 9.18 | 18.80 | 29.76 | 39.20 | 24.20 | 54.6 | 96.8 | 129 | 4000 |
| AFNO-$S$ | 4.25 | 14.80 | 23.58 | 30.40 | 8.30 | 99.9 | 171 | 196 | 4000 |
| AFNO-$B$ | 3.69 | 13.33 | 23.52 | 29.80 | 7.69 | 88.7 | 158 | 190 | 4000 |
| P3D-$S$ | 2.81 | 9.87 | 20.50 | 28.25 | 2.15 | 8.23 | 21.6 | 31.9 | 4000 |
| | 2.17 | 8.99 | 19.40 | 27.40 | 1.29 | 6.68 | 16.0 | 25.3 | 20000 |
| P3D-$B$ | 2.04 | 8.79 | 20.23 | 31.52 | 0.72 | 1.39 | **3.38** | 19.2 | 4000 |
| | **1.54** | **8.11** | 21.09 | 44.92 | **0.21** | **0.71** | 3.49 | **14.7** | 20000 |

Table C2: Architecture benchmark comparison on crop size $128^3$ measured on a H100 GPU using CUDA 12.8. VRAM is measured at inference with batch size 1. Throughput with batch size 16.

| Model | Params | GFLOPs | VRAM | Throughput |
|---|---|---|---|---|
| Swin3D-$S$ | 18.9M | 79.6 | 2.2G | 51.2it/s |
| Swin3D-$B$ | 50.3M | 289.6 | 2.9G | 34.1it/s |
| AViT-$S$ | 15.1M | 48.8 | 0.16G | 859it/s |
| AViT-$B$ | 60.0M | 143.0 | 0.3G | 346it/s |
| AViT-$L$ | 239.2M | 467.0 | 1.1G | 128it/s |
| AFNO-$S$ | 17.0M | 942.2 | 0.2G | 167it/s |
| AFNO-$B$ | 64.1M | 2116.4 | 0.4G | 58.6it/s |
| P3D-$S$ | 11.2M | 238.7 | 0.9G | 88.9it/s |
| P3D-$B$ | 46.1M | 2330.6 | 2.1G | 30.9it/s |

## C.2   VORTICITY

The vorticity $\omega$, is a measure of the local rotation in a fluid flow. It is defined as the curl of the velocity vector field $\mathbf{u} = (u_x, u_y, u_z)$, where $u_x, u_y$ and $u_z$ are the velocity fields in $x$, $y$ and $z$ direction. The vorticity is defined as

$$\omega = \nabla \times \mathbf{u} = \left( \frac{\partial u_z}{\partial y} - \frac{\partial u_y}{\partial z} \right) e_1 + \left( \frac{\partial u_x}{\partial z} - \frac{\partial u_z}{\partial x} \right) e_2 + \left( \frac{\partial u_y}{\partial x} - \frac{\partial u_x}{\partial y} \right) e_3, \tag{9}$$

where $e_1, e_2$ and $e_3$ represent the unit vectors in $x$, $y$ and $z$ direction.

The vorticity is not part of the data. Only the velocity channels in X/Y/Z are available.

**Approximation by finite differences**   To compute the components of the vorticity numerically from the velocity, we use finite difference approximations for the partial derivatives in Equation (9).

For a grid point $(i, j, k)$ located in the interior of the discretized simulation domain, we approximate the partial derivates as

$$\frac{\partial u_x}{\partial x} \approx \frac{u_x^{i+1,j,k} - u_x^{i-1,j,k}}{2\Delta x}$$
$$\frac{\partial u_x}{\partial y} \approx \frac{u_x^{i,j+1,k} - u_x^{i,j-1,k}}{2\Delta y}$$
$$\frac{\partial u_x}{\partial z} \approx \frac{u_x^{i,j,k+1} - u_x^{i,j,k-1}}{2\Delta z}$$

and analogously for $u_y$ and $u_z$.

## C.3   ENSTROPHY GRAPH

We consider the magnitude $|\omega|$ of the vorticity $\omega$, which is a scalar field defined as

$$|\omega| = \sqrt{\sum_{i=1}^{n} |\omega_i|^2}. \tag{10}$$

The enstrophy of the magnitude of the vorticity field $|\omega|$ at wavenumber $k \in \mathbb{R}_+$ is defined as

$$\text{Enstrophy}(k) = \sum_{k \leq |m| \leq k+1} |\hat{\omega}(m)|^2, \tag{11}$$

where $\hat{w}$ with $m \in \mathbb{Z}^2$ are the Fourier coefficients of $|\omega|$, see Chen et al. (2024, C.2) for reference. The enstrophy spectrum is the graph of the function $k \mapsto \text{Enstrophy}(k)$. We average the enstrophy spectrum over different vorticity fields.

We compute the L2 enstrophy error between the reference enstrophy spectrum and the enstrophy spectrum of generated vorticities as

$$L^2_{\text{Enstrophy}} := \frac{1}{K+1} \sum_{0 \leq k \leq K} \left[ \text{Enstrophy}^{\text{avg}}_{\text{reference}}(k) - \text{Enstrophy}^{\text{avg}}_{\text{generated}}(k) \right]^2 \tag{12}$$

with $K = 16$.

**Hanning filter**   The data on the cropped domain of size $128^3$ is not periodic, thus there are artifacts at the boundary of the crop. This leads to problems when calculating the Fourier coefficients $\hat{\omega}$. We therefore smoothen $|\omega|$ towards the boundary by multiplying the data with the Hann window as defined in `torch.hann_window`.

In Figure C2 we show the magnitude of the vorticity $|\omega|$ generated by P3D-$B$ and the reference after applying the Hanning filter ($t = 1$ autoregressive steps).

See Figure C3 for the enstrophy spectrum $\text{Enstrophy}^{\text{avg}}_{\text{generated}}$ of the predicted vorticities by P3D-$B$ and the enstrophy spectrum $\text{Enstrophy}^{\text{avg}}_{\text{reference}}$ of the reference for autoregressive unrolling steps $t = 1$ and $t = 15$ on the test set.

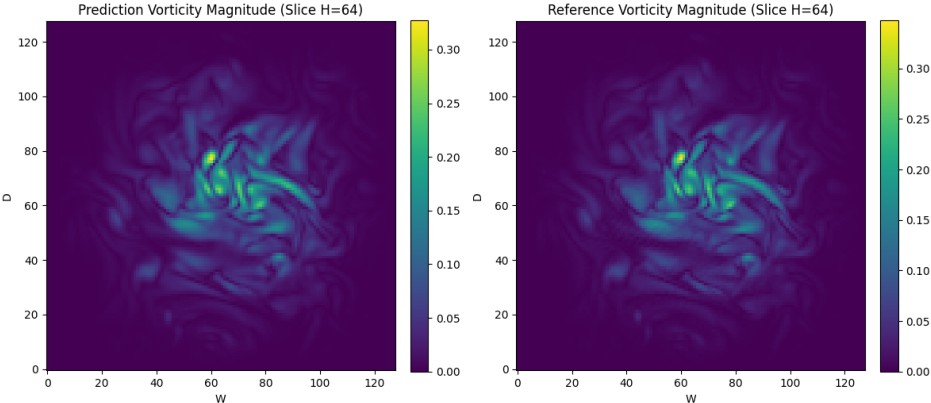

Figure C2: Generated vorticity magnitude $|\omega|$ by P3D-$B$ (left) and reference (right) after applying the Hanning filter.

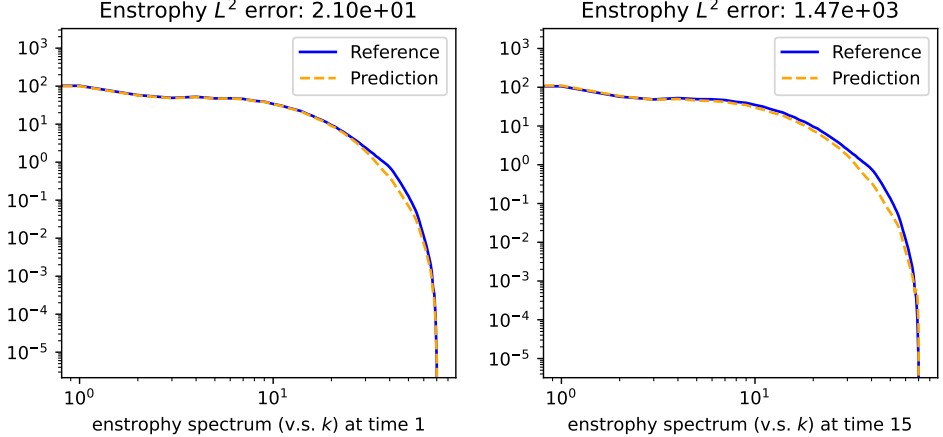

Figure C3: Enstrophy spectrum of P3D-$B$ and the reference simulation at resolution $128^3$ for $t = 1$ and $t = 15$ autoregressive prediction steps.

## C.4 TRAINING AND EVALUATION

The model weights for the evaluation in Table 4 are the EMA weights at epoch $2000$. We train both the $S$ and $B$ configurations of P3D on crop size $128^3$. Training P3D-$S$ for $4000$ epochs took 11h 48m and training P3D-$B$ took 20h 25m on four A100 GPUs. See Figure C4 for the validation loss curve.

In Figure C5 we show a comparison between the reference and P3D-$S$ <128|512> with an autoregressive rollout until $t = 16$ on the test set at resolution $512^3$.

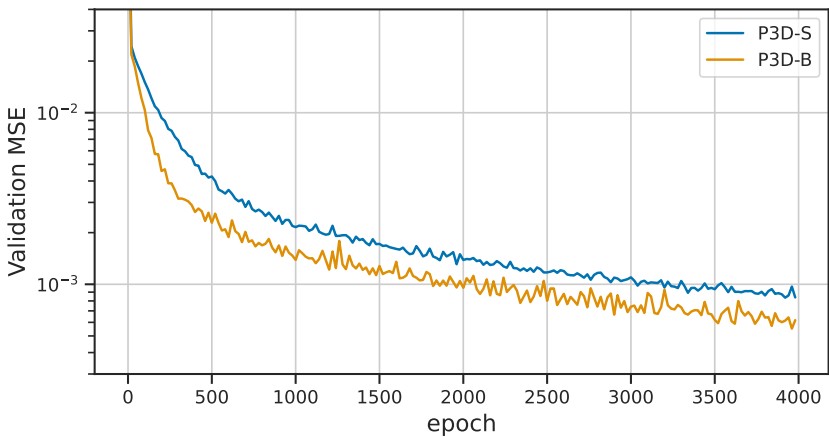

Figure C4: Isotropic Turbulence. Validation MSE for P3D-$S$ and P3D-$B$ during training.

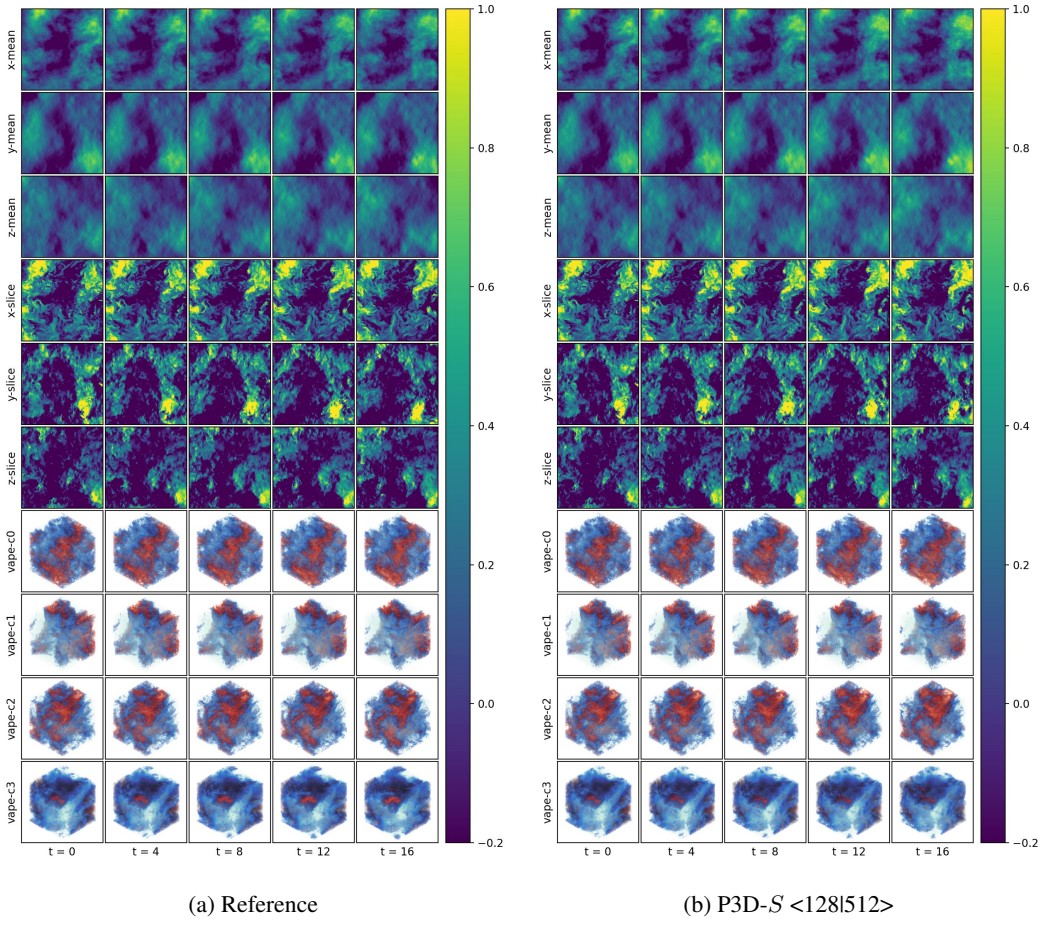

(a) Reference

(b) P3D-$S$ <128|512>

Figure C5: Isotropic Turbulence. Autoregressive prediction for $t = 16$ steps with P3D-$S$ <128|512> on the test set at resolution $512^3$. The first six rows are 2D mean projections and slices of the velocity in $x$-direction. The 3D renderings show velocity X/Y/Z and pressure.

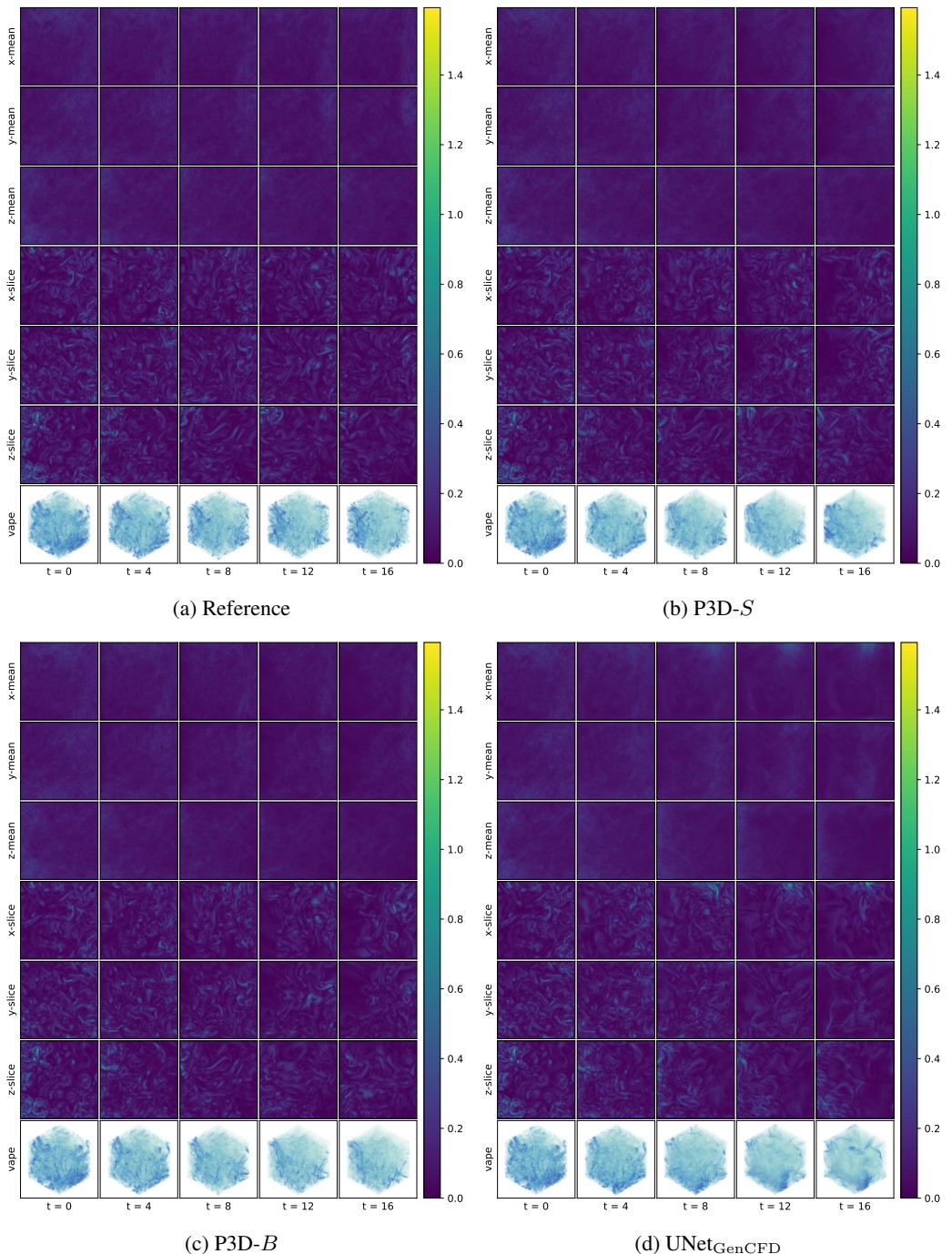

Figure C6: Vorticity magnitude $|\omega|$ calculated from the predicted velocity X/Y/Z for different models (1/2).

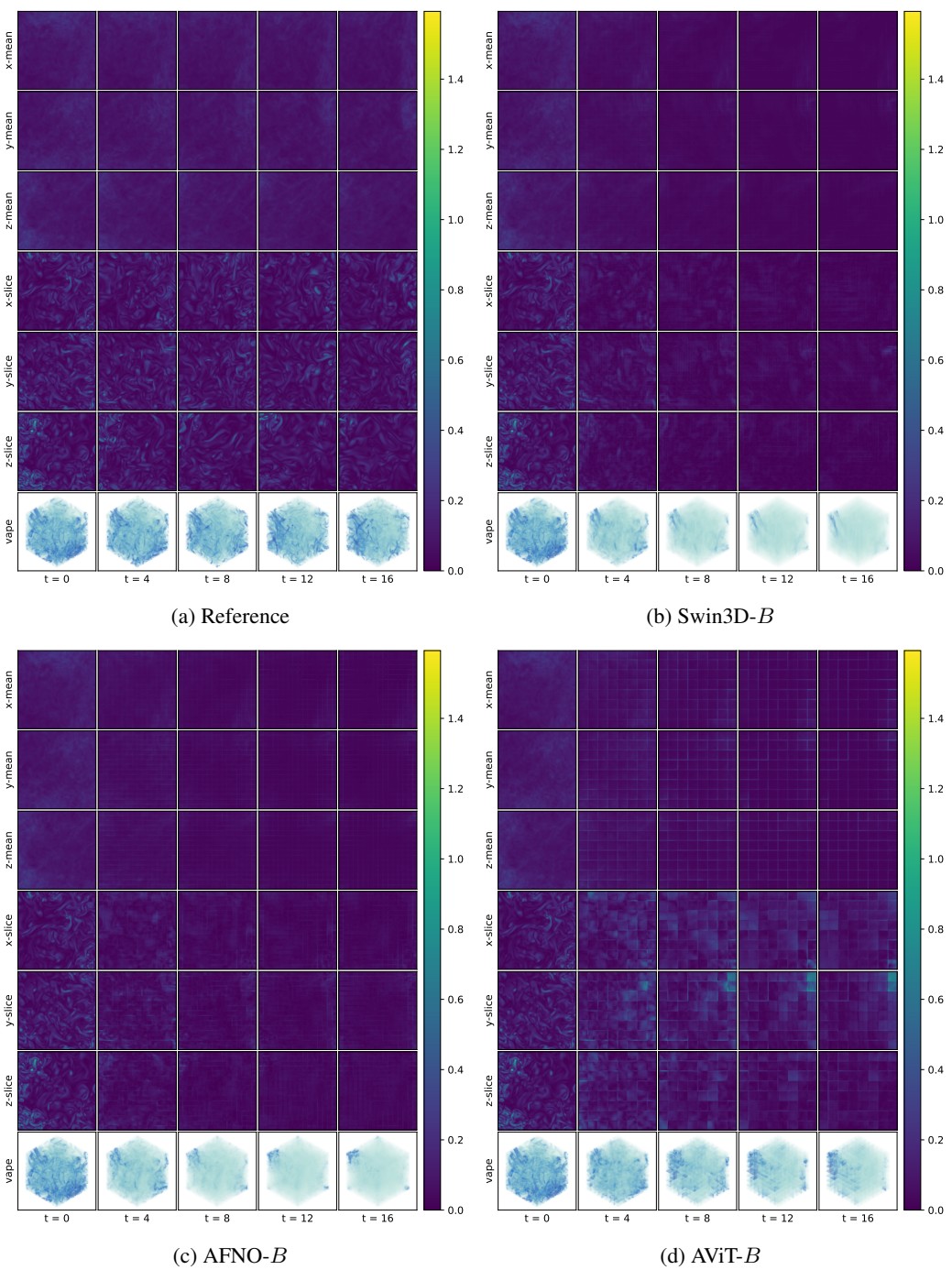

Figure C7: Vorticity magnitude $|\omega|$ calculated from the predicted velocity X/Y/Z for different models (2/2).

# D  EXPERIMENT 3: TURBULENT CHANNEL FLOW

The dataset for the turbulent channel flow (TCF) represents a periodic channel with no-slip boundaries at $\pm y$ that is driven by a dynamic forcing to re-inject energy lost due to wall friction, and prevent the flow from slowing down. This results in a continuous production of vortex structures at the walls, which have a very characteristic and well-studied, spatially-varying distribution (Hoyas & Jiménez, 2008). Due to the complexity of the flow, these flows require very long transient phases to develop the characteristic structures, We target this scenario by learning with a generative model, in the context of which the TCF problem represents a probabilistic learning problem to infer turbulent stats from the equilibrium phase, bypassing the costly transient warm-up phase.

## D.1  DATASET

We generate a dataset comprising 20 simulations with Reynolds numbers within the interval $[400, 800]$ spaced equidistantly. After the initial-warmup phase, we simulate $ETT = 20$ eddy turnover cycles, which we save in 200 snapshots with $\Delta t = 0.1$. The computational grid comprises $96 \times 96 \times 192$ spatially adaptive cells with a finer discretization near the wall. The data contains channels for the velocity in X/Y/Z direction as well as pressure. The computational grid is shown in Figure D1. In Figures D2 and D3 we show visualizations of the turbulent channel flow for Reynold numbers $Re = 400$ and $Re = 640$ respectively.

- Dimensionality: $s = 20$, $t = 200$, $f = 4$, $x = 96$, $y = 96$, $z = 192$
- Initial conditions: noise
- Boundary conditions: periodic (x), wall (y,z)
- Time step of stored data: 0.1
- Number of warmup steps (discarded, in time step of data storage): 200
- Spatial domain: $[-1, 1] \times [-1, 1] \times [-\pi, \pi]$
- Fields: velocity X/Y/Z, pressure
- Varied parameters: Reynolds number $\in [400, 800]$
- Validation set: random $15\%$ split of Reynolds number

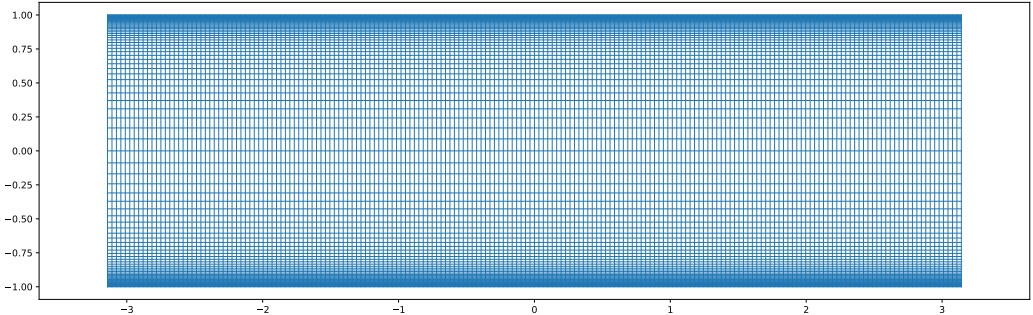

Figure D1: Computational grid of the turbulent channel flow simulation. The spatial discretization is refined in the near-wall region to resolve the boundary layer.

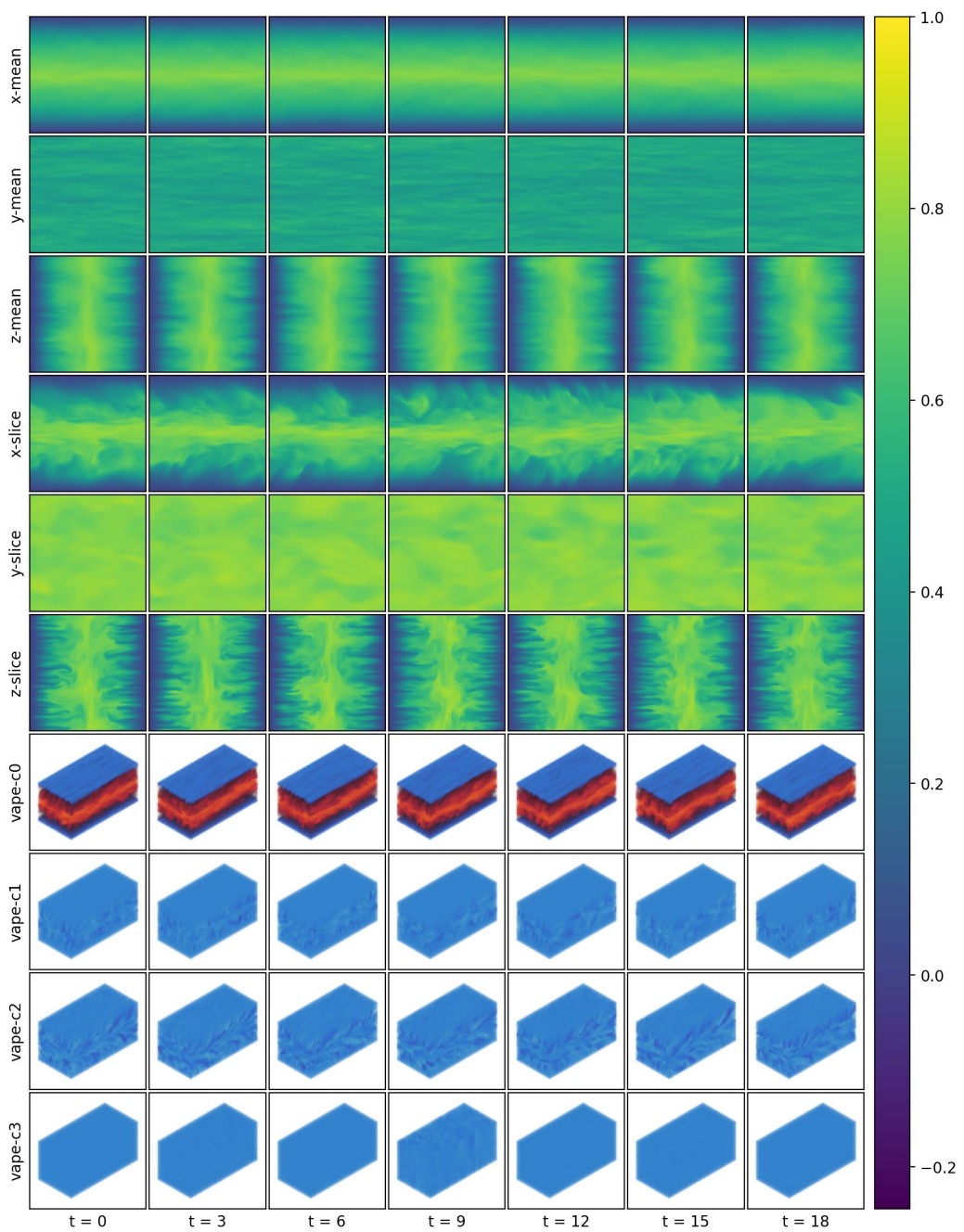

Figure D2: Turbulent channel flow with Reynolds number 400.

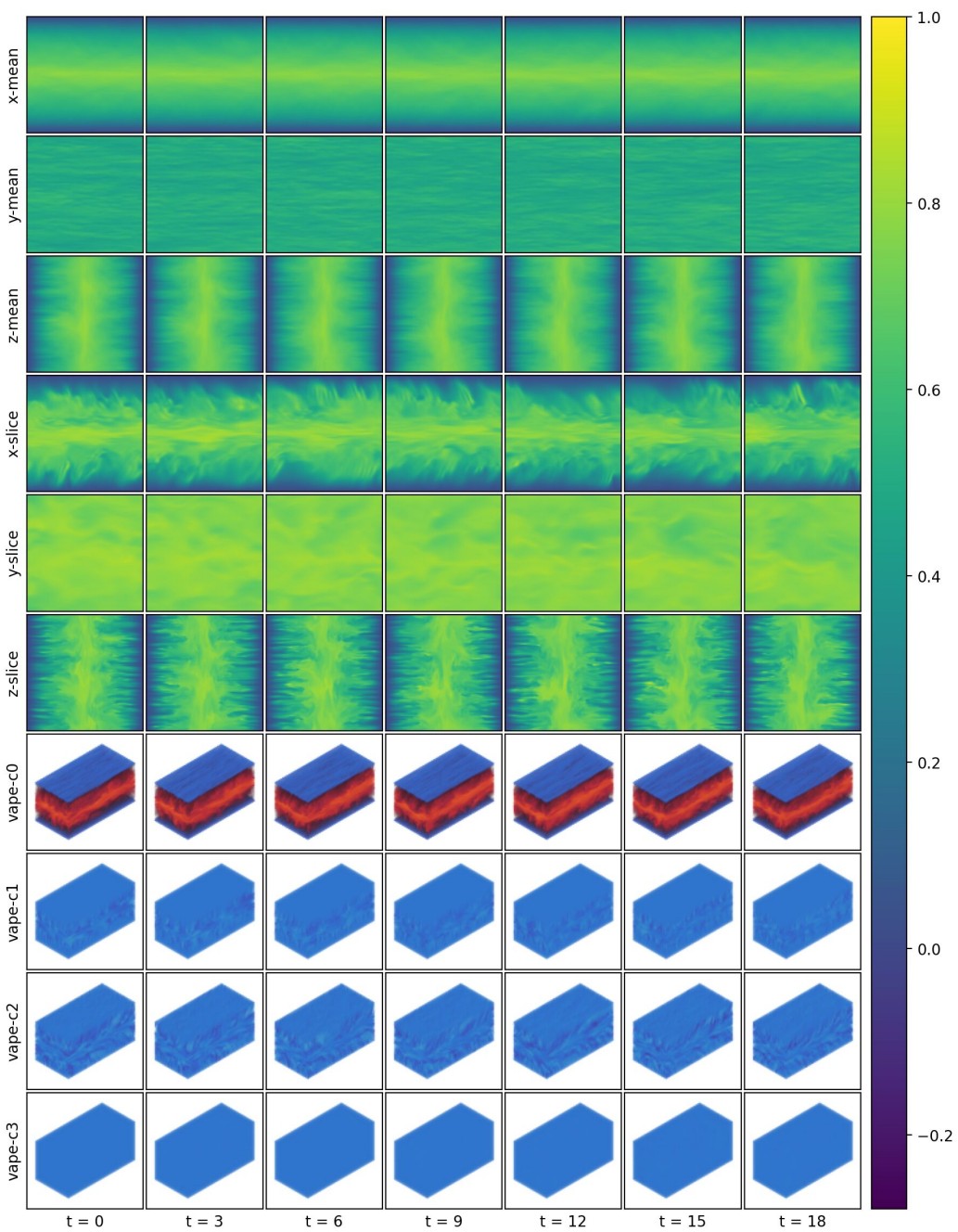

Figure D3: Turbulent channel flow with Reynolds number 640.

**Training S, B and L configs**   We train P3D with the configurations $S$, $B$ and $L$ for 400 epochs on the full domain of size $96^2 \times 192$. The training loss is shown in Figure D4. It is important to choose large architectures in our generative modeling setup based on flow matching. The network size, specifically the embedding dimension is critical for this task with the $L$ config reaching significantly lower loss values compared to the $S$ config trained with the same number of epochs. All models were trained on 4 A100 GPUs with 80GB VRAM. Training took 11h 4m, 14h 43m and 27h 55m for the $S$, $B$ and $L$ configs respectively. We train P3D-$L$ both with conditioning on the Reynolds number enabled and with conditioning disabled. We use the former for the visualization in Figure 9 and the latter in the comparison against baselines.

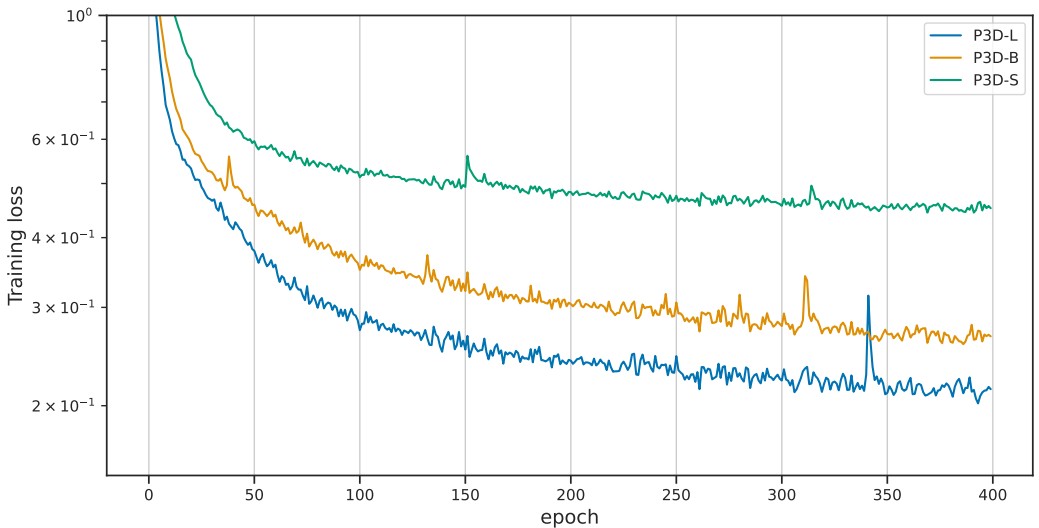

Figure D4: Turbulent channel flow. Training curve for different configurations of P3D.

## D.2   STATISTICAL EVALUATION

Let $\mathbf{x} = (x_1, x_2, x_3)$ denote the spatial coordinates and $u(\mathbf{x}, t, \mathrm{Re})$ denote the velocity of the flow direction X. The reference simulations reach an equilibrium phase after the initial transient phase of the warmup. Therefore, for the reference simulations, the moments $u_m(\mathbf{x}, t, \mathrm{Re})$ should be the same for all $t$. Additionally, the setup combining periodic boundary conditions and no-slip boundaries for the wall implies that the statistics *only* depend on the distance to the wall on the flow axis $x_1$, i.e., $u_m((x_1, x_2, x_3), t, \mathrm{Re})$ is the same for all $x_2$ and $x_3$ inside the domain. Thus it is reasonable to consider $u_m(x_1, \mathrm{Re})$ and calculate the moments by sampling over $x_2, x_3, t$. Since the baseline methods cannot be conditioned on the Reynolds number $\mathrm{Re}$, we also compute the velocity profile over $\mathrm{Re}$. We compute the L2 distance between mean $\mu$ $(m = 0)$ and variance $\sigma^2$ $(m = 1)$ of the velocity profile graph $x_1 \mapsto u_m(x_1)$ of the time-resolved DNS reference and generated samples. The resulting L2 distance for the mean $\mu$ and variance $\sigma^2$ is shown in Table 5, which provides a meaningful statistical metric to evaluate accuracy. We additionally train two baselines, AFNO and UNet$_{\mathrm{GenCFD}}$ with identical training setups as P3D-$L$ on the full domain.

**Moments of the flow field**   We included comparisons of the first three moments (mean, variance, skewness) $u_m$ of the flow direction (velocity in x-direction) averaging over $x_1$ as well. We report the standard deviation when estimating the moments of the reference based on randomly drawing 20 simulations states of the equilibrium phase per Reynolds number as done for the velocity profiles to properly assess how close the generated samples should match the reference. For the finetuned P3D-$L$ <X48|X48>, we picked the best model out of the five finetuned models. See Table D1. Overall, P3D-$L$ trained on the full domain and P3D-$L$ <X48|X48> achieve the best results.

**Velocity profile comparison**   See Figure D5 for a comparison between the mean channel flow of the reference simulation for different Reynolds number and the mean flow from the generative model P3D-$L$ trained on the full domain. We show additional comparisons between the different

Table D1: First three moments of the velocity field in the flow direction for the reference, P3D-$L$ and the two baseline methods.

| Moment | Mean | Variance | Skewness |
|---|---|---|---|
| Reference | 0.5034±0.0007 | 0.0511±0.0001 | -0.776±0.007 |
| UNet$_{\mathrm{GenCFD}}$ full domain | 0.5002 | 0.0532 | -0.723 |
| AFNO full domain | 0.5040 | 0.0930 | -0.361 |
| P3D-$L$ full domain | 0.5009 | 0.0513 | -0.789 |
| P3D-$L$ <X48\|X48> | 0.5044 | 0.0510 | -0.802 |

training and inference strategies in Figure D6. Scaling the P3D-$L$ network trained on the small crops of size $48^3$ to the full domain does not work well and results in incorrect velocity profiles. By finetuning with the context network, region crops can coordinate and obtain information about their relative position to each other as well as to the wall. As a results, the flow statistics improve significantly, more closely matching the reference and samples from P3D-$L$ trained on the full domain.

We show samples from P3D-L trained on the full domain in Figure D7, when applying P3D-L pretrained on the full domain without any finetuning in Figure D8 and with finetuning via the context network and learned region-dependent conditioning in Figure D9 respectively.

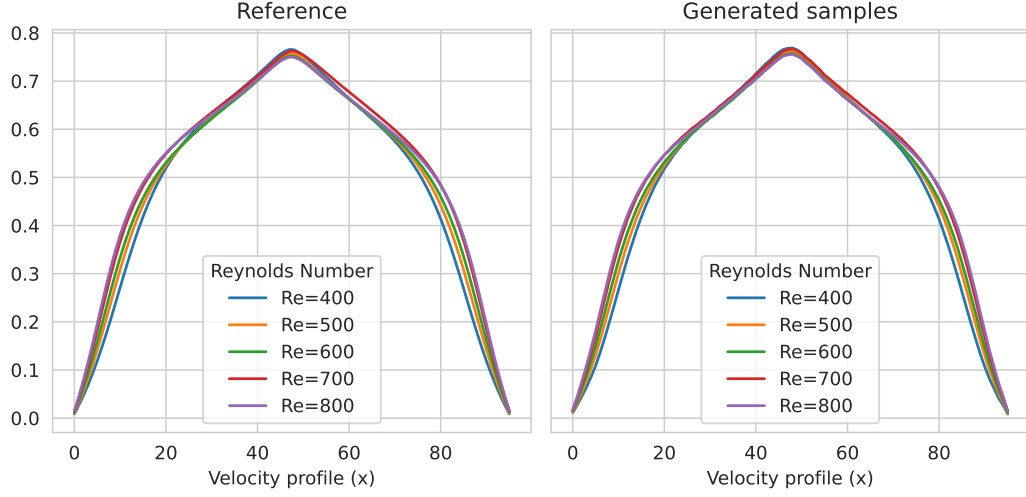

Figure D5: Comparison of the mean channel flow of the reference and of generated samples from P3D-$L$ trained on the full domain.

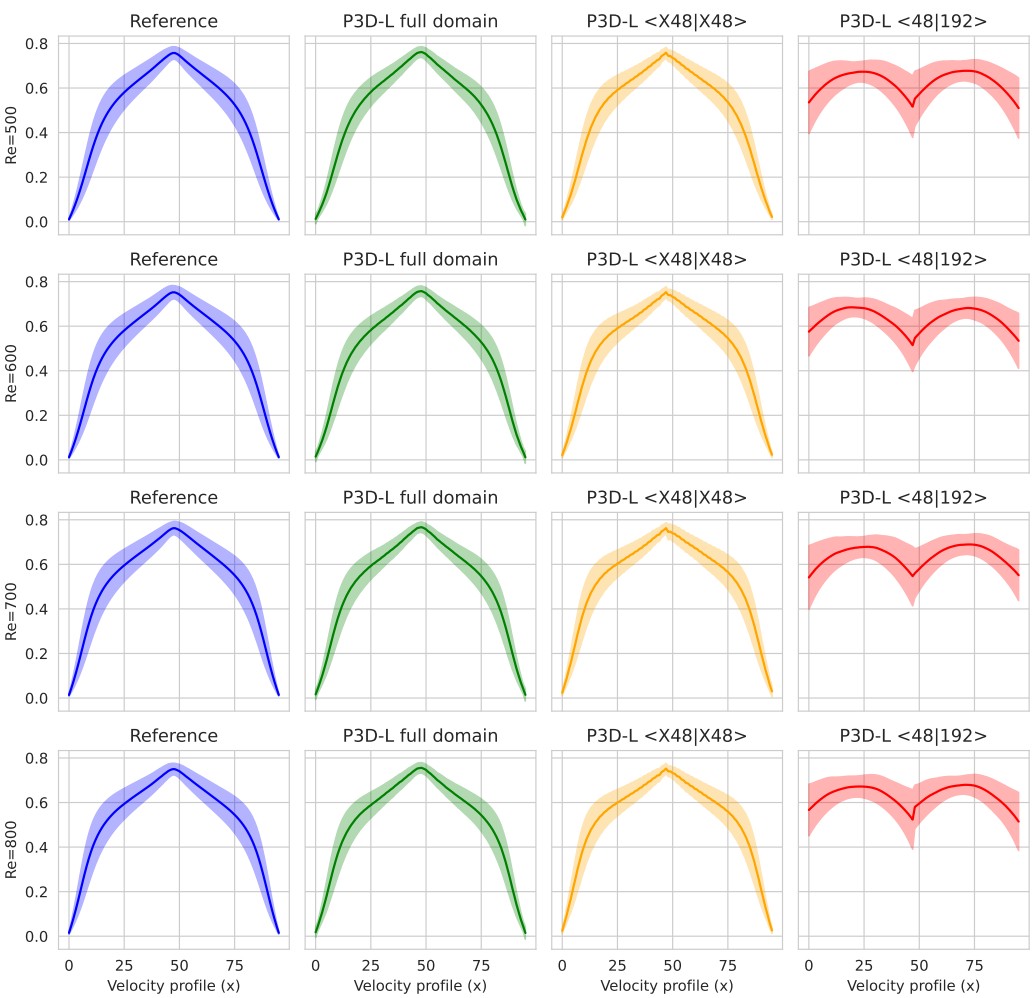

Figure D6: Comparison of the first two moments of the velocity profile between the reference DNS and generated samples from P3D-$L$ with different training and inference strategies.

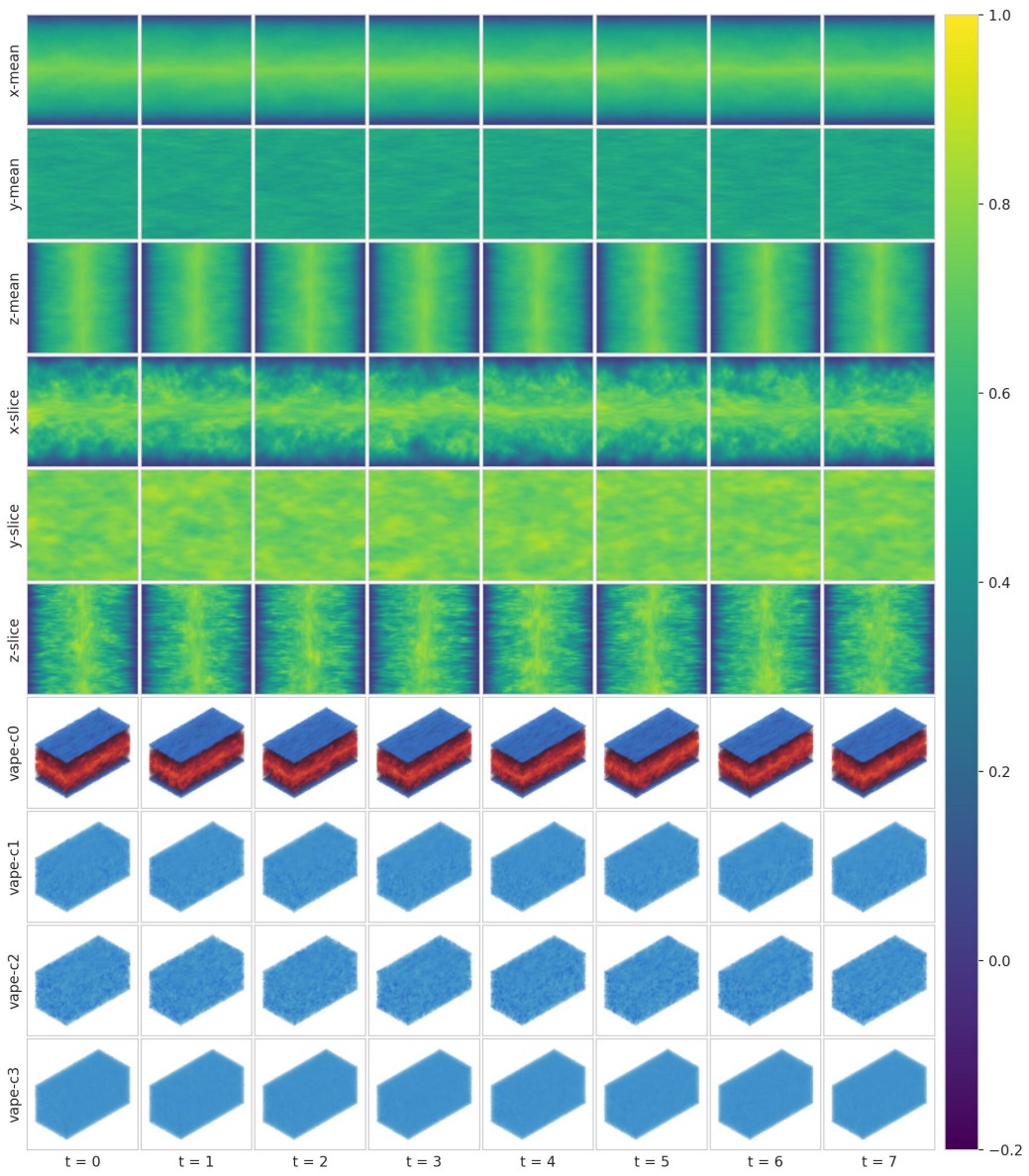

Figure D7: Samples from P3D-$L$ trained on the full domain at Re $= 800$.

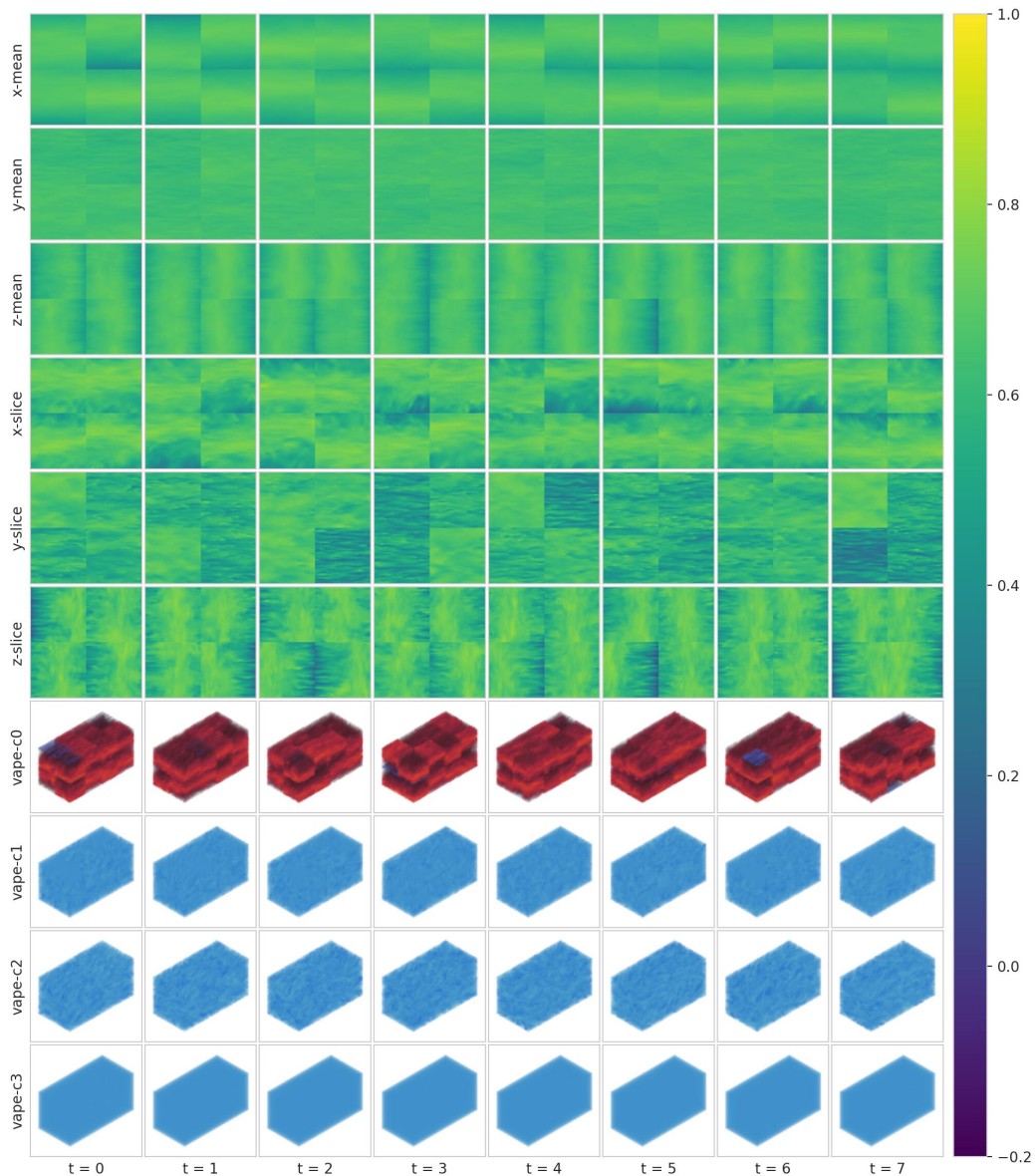

Figure D8: Samples from P3D-$L$ <48|192> pretrained on crops of size $48^3$. Inference on the full domain at Re $= 800$ produces incorrect samples, as information on the relative positions between region crops is not available.

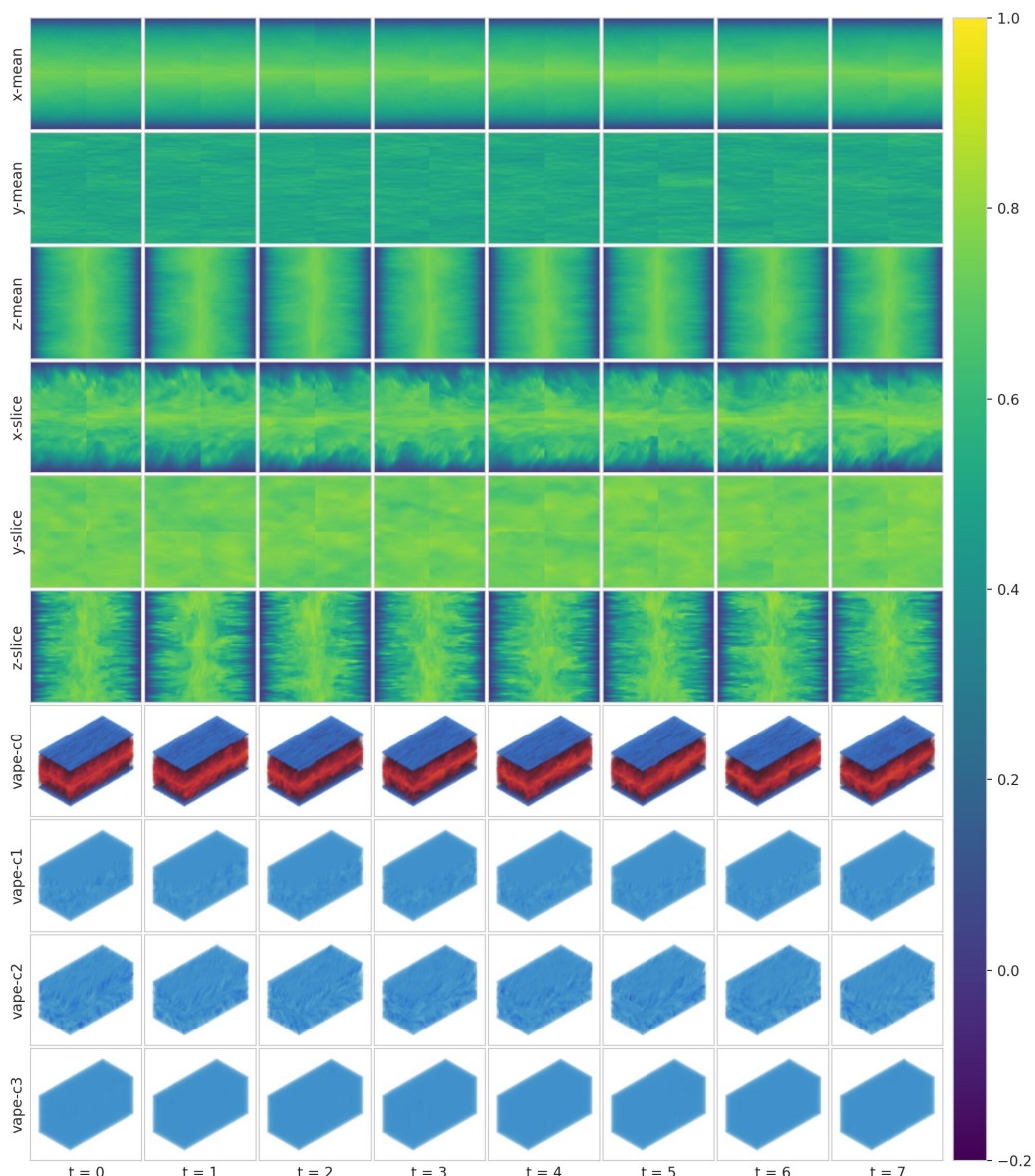

Figure D9: Samples from P3D-$L$ <X48|X48> pretrained on crops of size $48^3$ and finetuned with the context network. Inference on the full domain at Re $= 800$ produces samples that exhibit the correct flow statistics.

