# OpenReview forum: "P3D: Highly Scalable 3D Neural Surrogates for Physics Simulations with Global Context"
_ICLR.cc/2026/Conference — ICLR 2026 Poster_

### Official Review · Reviewer_HKuH · 2025-10-29

**Soundness:** 2
**Presentation:** 4
**Contribution:** 3
**Rating:** 6
**Confidence:** 4

**Summary:**

The paper proposes a network architecture and strategy for training on high resolution 3D data. The overall approach uses entirely translation equivariant blocks in the encoder and decoder to allow the model to take full advantage of cropped training. The authors then propose multiple approaches for scaling these models to high resolution 3D data. They experimentally evaluate their architecture on a multi-task learning problem, on a high resolution autoregressive task, and on a generation problem.

**Strengths:**

1. Overall this is a very well written paper. It makes the problem clear and proposes a solution along with varying approaches to implement that solution.
2. Apart from the one note below, the experiments are well designed and well suited to showing the scaling performance of the proposed method.
3. This is an enormous challenge in the field and this work is showing promising results on some extremely difficult tasks.

**Weaknesses:**

Overall, this is a strong paper that makes important progress on a challenging problem in the space. Right now there are a few issues that I'd consider to be large enough that I can't mark it as a clear accept, but with some minor changes, I'd be happy to increase my score. I've included comments on how these issues could be addressed experimentally which would lead to a larger increase, but to be clear, my primary asks to recommend acceptance are the inclusion of rollout metrics and several adjustments to claims and presentation.

Major:

1. 4.1 is definitely a weak spot in the paper.  The rest of the experiments are strong enough that I'm not marking down too far for this, but either experiments or claims could be adjusted to address several issues:
    1. The biggest issue is that that the claim of 14+ PDEs does not appear to be true. It's possible I'm missing something, but Appendix B lists 7 PDEs. It's not reasonable to consider Gray-Scott multiple PDEs due to the varying macroscale behavior.
   2. Second, there is limited evidence of performance as a "foundation model". While previous works like Poseidon have shown the conv+swin backbone is a strong basis for foundation models, the experimental evidence provided is insufficient to make that argument. These are largely fairly simple systems defined on periodic cubes, so this is a case where the model has a strong inductive bias and then is only evaluated for multi-task learning on problems that benefit from that inductive bias. To make this claim, I'd recommend including varying boundary conditions or geometric forcings or just generally more real-world inspired data. Otherwise, I'd recommend just being more explicit on what the limitations of the experiment are.
2. The difference in metrics between table 3 and 4 also makes it difficult to evaluate whether the results in table 4 are "good" beyond the accomplishment of running a model at all on such high resolution data. While it doesn't make sense to ask for ML baselines, I think simple baselines like persistence and predicting the field average would help put the results in context. Additionally, I think evaluating step 1 on the same distribution in both spaces would allow for some comparison between the models.
3. Generally speaking, the paper's claim of stable autoregressive rollouts does not seem well supported.  Metrics are provided for 1 and 15 steps in 4.2. Some evaluation of rollout on the scaled up model is necessary to make claims that the approach can make stable predictions on high resolution.
4. The paper could engage more with the weaknesses of the method. For instance, many of the architectural choices are not compatible with non-uniform grids so there isn't a natural extension to many realistic problems.

Minor:
1. For data generated for a specific project, there should really be more precise information about the generating conditions in the appendix, specifically precise mathematical descriptions for each task.
2. The context network is introduced, but it's not clear if its being used outside of 4.3. Is it being used? Would it have helped with the independent patch model <128|128> in 4.2? Similarly, there's not much comparison of the training approaches in Figure 4.
3. This might be a personal issue, so I'm not marking down for this either, but it's a bit weird to use llama library names for standard network components, particularly given this is not a NLP submission. It would be more informative to say the standard names for the model components, then mention you use the llama implementations.
4. Appendix C.1. links to table 4 - it seems to be an extended version of table 3 based on the caption and numbers.

**Questions:**

1. Is the "conv" model in the ablations in 4.1 replacing the attention layers with more convolutional blocks or is it just removing components?
2. Could the context network have eliminated the boundary artifacts in 4.2 for <128|128>?
3. How does this approach compare to baselines in the presence of boundary conditions or geometric influence?
4. How does the model perform over longer autoregressive rollouts?

---

> ### Author Response · Authors · 2025-11-26
>
> Thank you for your detailed and constructive feedback. We greatly appreciate your suggestions for improving the paper and address each point below.
>
> **Rollout metrics** We conducted additional evaluations for the rollout metrics of the scaled-up model at $512^3$ resolution. We used the same setup as in Table 3, where we compared P3D and baselines at the $512^3$ resolution. The two tables below present the enstrophy spectrum error and RMSE results.
>
> | Enstrophy | 1 | 5 | 10 | 15 | 20 | 30 | 40 | 50 | 60 |
> |---|---|---|---|---|---|---|---|---|---|
> | P3D-S <128\|128> | 2.15 | 8.23 | 21.60 | 31.90 | 43.20 | 90.15 | 122.83 | 124.51 | 207.14 |
> | P3D-S <128\|512> | 1.03 | 1.35 | 1.23 | 1.28 | 2.18 | 4.65 | 2.81 | 8.14 | 164.15 |
> | Mean-field | 222.9 | 223.5 | 225.9 | 229.3 | 231.7 | 241.3 | 255.9 | 229.5 | 212.4 |
>
> | RMSE | 1 | 5 | 10 | 15 | 20 | 30 | 40 | 50 | 60 |
> |---|---|---|---|---|---|---|---|---|---|
> | P3D-S <128\|128> | 2.81 | 9.87 | 18.81 | 26.80 | 33.62 | 45.48 | 54.70 | 61.67 | 67.70 |
> | P3D-S <128\|512> | 2.36 | 7.12 | 13.95 | 21.00 | 27.47 | 38.75 | 48.34 | 56.9 | 66.6 |
> | Mean-field | 53.74 | 53.56 | 53.37 | 53.11 | 52.81 | 52.19 | 51.57 | 50.71 | 49.94 |
>
> We performed inference with the scaled-up model P3D-S <128|512> on the full domain, but for evaluation, we computed the spectral error and RMSE on the $128^3$ crops. This ensures the metrics are directly comparable to those in Table 3.
>
> P3D-S <128|512>, which jointly infers the whole domain (cf. strategy (1) in section 3.3), maintains an enstrophy spectrum error below $10 \times 10^2$ for up to 50 autoregressive rollout steps, whereas P3D-S <128|128>, which processes individual regions, achieves this for only 10 steps. We note that the $512$ domain itself is cropped from an even larger $1024$ resolution, introducing boundary uncertainties that complicate autoregressive unrolling, even for the scaled-up model. Not being able to reliably compare small scale structures, the RMSE values are still slightly lower for the scaled-up <128|512> model. This P3D-S version outperforms the mean-field prediction for up to 40 to 50 steps.
>
> Overall, especially the enstrophy comparison highlights the advantages of applying P3D models pre-trained on 128^3 crops to larger domains (via the P3D-S <128|512> model) for autoregressive rollouts. We will include additional visualizations in the updated manuscript.
>
> - **Number of PDEs** in Section 4.1: Thank you for highlighting this point. We currently list "14 PDEs, 8 of which are distinct versions of the Gray-Scott reaction-diffusion model". Since each version of Gray-Scott exhibits a distinct behavior, we listed them as "PDEs", placing our definition between two extremes: a fully general PDE with configurable operators (broadest category) and a separate PDE for every unique coefficient set (finest taxonomy). Our approach considered different dynamics as distinct PDEs. However, we agree that this can be misunderstood, and we will clarify it in the text by referring "14 types of PDE dynamics" stemming from "8 PDEs" in the future.
>
> - **P3D Foundation Model**: We agree that definitively claiming P3D is a foundation model for general PDEs in 3D requires additional experiments demonstrating positive transfer between pretraining and unseen downstream datasets. Furthermore, even more diverse high-resolution simulations with varying boundary conditions and geometries are necessary. This was our reason for taking care to never refer to trained P3D models as "foundation models" in our submission.
>
> As we believe that foundation models for PDEs represent a very promising future direction, especially for 3D, and that P3D is a highly promising architecture for future efforst in this direction, we refer to this concept in the related work and the conclusion, stating it: "[...] establishes the foundation for scaling scientific foundation models to very high resolutions [...]". We will improve the wording in these sections to make this point clearer.
>
> - **Discussing Limitations**: We are aware of the limitations regarding generalization and extension to non-uniform grids/meshes. We will add a dedicated Limitations section to the updated manuscript to discuss these aspects, focusing particularly on the reliance on grids and boundary artifacts.
>
> - **Additional Changes for the Updated Manuscript** We will incorporate the following changes:
>   -  Provide more precise information about the data generating conditions for each dataset in the Appendix.
>   - Update the naming of the components/layers of the context network to use more general terms instead of the llama library names.
>   - Fix the incorrect reference in Appendix C.1: as you noted, it should be Table 3, not Table 4.

---

> > ### Author Response · Authors · 2025-11-26
> >
> > Regarding your questions:
> >
> > > Is the "conv" model in the ablations in 4.1 replacing the attention layers with more convolutional blocks or is it just removing components?
> >
> > The P3D-S-conv model is the result of simply removing components and consists only of the convolutional encoder and decoder.
> >
> > > Could the context network have eliminated the boundary artifacts in 4.2 for <128|128>?
> >
> > The experiment in Section 4.2 did not include fine-tuning with context. We explored using the context model in earlier stages, and while it does help with boundary artifacts, they are not completely eliminated (as seen in Figure D9 for the channel flow experiment). However, we found that in the absence of global long-range dependencies, the <128|512> scaling strategy is very efficient: it does not require fine-tuning and handles boundary artifacts effectively.
> >
> > > How does this approach compare to baselines in the presence of boundary conditions or geometric influence?
> >
> > Note that both training and inference on the cropped data impose a specific time-dependent boundary condition. Additionally, the channel flow experiment inherently includes both no-slip and periodic boundaries. Beyond this, we have not yet considered further experiments focused on the influence of boundary conditions or geometry, and we reserve this for future work.
> >
> > > How does the model perform over longer autoregressive rollouts?
> >
> > We included additional results with up to 60 autoregressive rollout steps for the isotropic turbulence simulation at $512^3$ resolution. These demonstrate that autoregressive rollouts with P3D-S maintain stability and good spectral errors for up to 50 steps.

---

### Official Review · Reviewer_rjE6 · 2025-10-30

**Soundness:** 3
**Presentation:** 4
**Contribution:** 3
**Rating:** 8
**Confidence:** 4

**Summary:**

The authors introduce a new architecture that uses convolution and attention modules in an encoder-decoder UNet structure and combine it with a latent global attention module that is used to exchange global information and inject region specific information into the decoder.
They evaluate various training and scaling strategies focusing on three setups: jointly learning multiple PDEs, pretraining on smaller patches and scaling to larger patches at inference, and a probabilistic training scenario.
Across all scenarios P3D generally outperforms the compared baselines, showing superior accuracy and efficiency.

**Strengths:**

## Originality
- The paper introduces a novel combination of architectural components essential for scaling to larger grids and systematically analyzes their contributions through ablations.
- The learned region tokens introduced in the bottleneck and the specific conditioning of the decoder regions is, to the best of my knowledge, a novel architectural design that, according to the presented results, helps to propagate global information.

## Quality
- Experiments are well designed and a rigorous description of the PDE setups as well as the architectural design and training hyperparameters are provided in the Appendix.
- The authors benchmark against multiple strong baselines and report consistent improvements in accuracy as well as efficiency.

## Clarity
- The paper’s organization is clear and logical, moving from background and motivation to contributions, ablations and results, which makes the narrative easy to follow.
- Main concepts are well described and most additional information that is crucial for understanding is provided in the Appendix.

## Significance
- Very interesting paper for the neural surrogate community, where scaling is essential to move towards real world problems.
- The authors outline design choices and findings (e.g.: domain-crop pretraining or region token based conditioning) that can be integrated into other methofs. Elements of this paper can therefore can serve as building blocks for future work.

**Weaknesses:**

1. Lack of uncertainty/variance reporting for the first two experiment setups. While I understand this is computationally expensive, I think it would be an important addition if the authors could at least include std across a few seeds in the main tables where they compare against other architectures.

2. The paper only reports 1-step errors for the mulit-PDE setting in Section 4.1: Including rollout errors or spectral metrics (as done in Section 4.2) would provide deeper insights.

3. Minor inconsistencies:
    - In Appendix A.5, the latent token dimensions notation is slightly confusing: (B, C, H, W, D) -> (N, (H × W × D), C). Why do you use B for batch and then N?
    - Section 4.1 mentions a maximum number of 3 channels. However Table B1 lists a maximum of two fields per PDE. Clarification would be helpful.

**Questions:**

1. How is data normalization handled? I think this can be a bit nuanced in the mulit-PDE setting: could be normalised per channel per PDE, or per channel across all PDEs with the same channels, or different strategy?

2. How does the P3D-shift ablation perform without the region tokens? It would be interesting to see if the information flow across windows is enough or this global information injection in the decoder is essential.

3. I think an important downstream task of such scalable architectures is finetuning on an unseen PDE: It would be interesting to have experiments that show what advantage pretraining can bring if we e.g. have a new PDE that is extremely expensive to simulate and we can only afford a few samples. Is there positive transfer?

4. Could you provide a breakdown of how the presented P3D model ablations (e.g. Table 2) compare with respect to their efficiency (Params, GFLOPs, Memory, Throughput)?

---

> ### Author Response · Authors · 2025-11-26
>
> Thank you for your detailed and constructive feedback. We greatly appreciate your suggestions for improving the paper and address each point below.
>
> **Uncertainty Metrics:** Below we provide mean/std for 3 different seeds for the P3D S, B and L configurations for the experiments in Section 4.1 for a crop size $32^3$.
>
> | nRMSE | Mean | Standard deviation | Min | Max |
> | ----- | ---- | ------------------ | --- | --- |
> | P3D-S | 0.057 | 0.0045 | 0.054 | 0.062 |
> | P3D-B | 0.043 | 0.0032 | 0.040 | 0.046 |
> | P3D-L | 0.039 | 0.0035 | 0.037 | 0.043 |
>
> Thus, the results (and our observed trends) are quite stable over repeated runs and varying network initializations.
>
> **Hyperparameters:** Training models with different seeds is indeed very expensive. However, for the models P3D-S, AFNO, UNet and Swin3D we performed an additional hyperparameter sweep over the learning rate
>
> | Learning rate | $10^{-3}$ | $5 \times 10^{-4}$ | $2 \times 10^{-4}$ | $10^{-4}$ | $5 \times 10^{-5}$ |
> | ------------- | --------- | ------------------ | ------------------ | ------| ----------- |
> | AFNO | 1.554 | 0.096 | 0.095 | 0.086 | 0.083 |
> | Swin3D | 0.084 | 0.080 | 0.079 | 0.083 | 0.088 |
> | UNet$_\text{GenCFD}$ | 0.966 | 0.085 | 0.076 | 0.079 | 0.079 |
> | P3D-S | 0.066 | 0.054 | 0.062 | 0.057 | 0.064 |
>
> These additional runs show that the results presented in our paper for a learning rate $10^{-4}$ were representative.
>
> **Rollout metrics for Section 4.1:** This is a good point. In the multi-PDE learning setup on cropped data, models that perform well on a single step, don't necessarily perform well on longer autoregressive rollouts. The reason for this is that the predicted solutions $s^*$ for $s_{t+\Delta t}^\text{crop}$ given $s_{t}^\text{crop}$, see Eq. (2) in the paper, typically differ in terms of their distribution from the training data (especially at the boundaries). This data shift is a classic challenge in auto-regressive rollouts, and is especially severe for small crops, where the boundaries have a strong influence.
>
> Below, we give an evaluation of the nRMSE up to 10 autoregressive rollout steps at crop size $128^3$. We removed the Swift-Hohenberg equation from the evaluation below, since we noticed it becomes unstable very fast and skews results.
>
> | Model | 1 | 5 | 10 |
> | :--- | :---: | :---: | :---: |
> | AFNO | 0.037 | 0.180 | 0.360 |
> | AVIT | 0.127 | 0.381 | 0.972 |
> | P3D-S | 0.028 | 0.166 | 0.396 |
> | P3D-B | 0.021 | 0.129 | 0.339 |
> | P3D-L | 0.017 | 0.156 | 0.639 |
> | Swin3D | 0.039 | 0.167 | 0.328 |
> | TFNO | 0.096 | 0.278 | 0.405 |
> | UNet$_\text{GenCFD}$ | 0.044 | 0.232 | 0.359 |
>
> P3D still has an advantage over baselines up to 5 autoregressive steps. For 10+ steps, predicting the dynamics becomes very difficult and all models have a similar performance with high nRMSE. Note that we show visualizations of P3D-L throughout the appendix for up to 8 autoregressive steps.
>
> We also analyzed **longer autoregressive rollouts in Section 4.2**, where the experiment setup makes more sense. See the table below for an evaluation:
>
> | Enstrophy | 1 | 5 | 10 | 15 | 20 | 30 | 40 | 50 | 60 |
> |---|---|---|---|---|---|---|---|---|---|
> | P3D-S <128\|128> | 2.15 | 8.23 | 21.60 | 31.90 | 43.20 | 90.15 | 122.83 | 124.51 | 207.14 |
> | P3D-S <128\|512> | 1.03 | 1.35 | 1.23 | 1.28 | 2.18 | 4.65 | 2.81 | 8.14 | 164.15 |
> | Mean-field | 222.9 | 223.5 | 225.9 | 229.3 | 231.7 | 241.3 | 255.9 | 229.5 | 212.4 |
>
> | RMSE | 1 | 5 | 10 | 15 | 20 | 30 | 40 | 50 | 60 |
> |---|---|---|---|---|---|---|---|---|---|
> | P3D-S <128\|128> | 2.81 | 9.87 | 18.81 | 26.80 | 33.62 | 45.48 | 54.70 | 61.67 | 67.70 |
> | P3D-S <128\|512> | 2.36 | 7.12 | 13.95 | 21.00 | 27.47 | 38.75 | 48.34 | 56.9 | 66.6 |
> | Mean-field | 53.74 | 53.56 | 53.37 | 53.11 | 52.81 | 52.19 | 51.57 | 50.71 | 49.94 |
>
> We performed inference with the scaled-up model P3D-S <128|512> on the full domain, but for evaluation, we computed the spectral error and RMSE on the $128^3$ crops. This ensures the metrics are directly comparable to those in Table 3.
>
> P3D-S <128|512>, which jointly infers the whole domain (cf. strategy (1) in section 3.3), maintains an enstrophy spectrum error below $10 \times 10^2$ for up to 50 autoregressive rollout steps, whereas P3D-S <128|128>, which processes individual regions, achieves this for only 10 steps. We note that the $512$ domain itself is cropped from an even larger $1024$ resolution, introducing boundary uncertainties that complicate autoregressive unrolling, even for the scaled-up model. Not being able to reliably compare small scale structures, the RMSE values are still slightly lower for the scaled-up <128|512> model. This P3D-S version outperforms the mean-field prediction for up to 40 to 50 steps.

---

> > ### Author Response · Authors · 2025-11-26
> >
> > > In Appendix A.5, the latent token dimensions notation is slightly confusing: (B, C, H, W, D) -> (N, (H × W × D), C). Why do you use B for batch and then N?
> >
> > Thank you for noticing this inconsistency. We will clarify and update the notation in the updated manuscript.
> >
> > > Section 4.1 mentions a maximum number of 3 channels. However Table B1 lists a maximum of two fields per PDE. Clarification would be helpful.
> >
> > Thank you for pointing this out: the velocity fields for the KdV and Burgers PDE scenarios have three components, the velocity in X, Y and Z direction, i.e. 3 channels. We will correct this in the updated manuscript.
> >
> > > How is data normalization handled? I think this can be a bit nuanced in the mulit-PDE setting: could be normalised per channel per PDE, or per channel across all PDEs with the same channels, or different strategy?
> >
> > This is a good question, however, for our data generation (inspired by the "APEBench" dataset), the PDEs are parametrized to stay close to the [-1, 1] range in terms of their outputs. We will explain this in the updated appendix.
> >
> > > How does the P3D-shift ablation perform without the region tokens? It would be interesting to see if the information flow across windows is enough or this global information injection in the decoder is essential.
> >
> > The region tokens are only active during finetuning in Section 4.3. Therefore, for the P3D-shift ablation, the information flow through the upsampling/downsampling is sufficient for achieving good performance. Note that there is a 16x downsampling in the convolutional and transformer encoders. For window size 4, each window in the bottleneck layer has a receptive field of at least 64^3.
> >
> > > I think an important downstream task of such scalable architectures is finetuning on an unseen PDE: It would be interesting to have experiments that show what advantage pretraining can bring if we e.g. have a new PDE that is extremely expensive to simulate and we can only afford a few samples. Is there positive transfer?
> >
> > That's a very good point. Positive transfer for unseen PDEs was shown in 2D experiments, and we expect it to transfer similarly well to 3D. We did not perform new experiments to verify claim this for 3D and, due to the scope of the training runs, leave this to future work.
> > It is definitely an important direction for large-scale architectures, and, rather than providing overly preliminary results, we plan to evaluate proper pretraining together with transfer capabilities to downstream tasks in future experiments.
> >
> >
> > > Could you provide a breakdown of how the presented P3D model ablations (e.g. Table 2) compare with respect to their efficiency (Params, GFLOPs, Memory, Throughput)?
> >
> > Below, we provide detailed information on each architecture in Table 2. Throughput is measured on a RTX 5000 GPU. Note that we identified a mistake in the calculation of FLOPs. In the paper, we actually report the MACs. One MAC is two FLOPs. Therefore, we will update the manuscript with the corrected number, which is the previous one scaled by 2.
> >
> > |   | Params | GFLOPs | Memory | Throughput |
> > | - | ------ | ------ | ------ | ---------- |
> > | P3D-S-conv | 0.34M  | 110.56 | 0.96GB | 66.67 it/s |
> > | P3D-S-patch | 10.90M | 77.64 | 0.31GB | 100.14 it/s |
> > | P3D-S-no-c | 11.23M | 238.76 | 1.0 GB | 35.44 it/s |
> > | P3D-S-shift | 11.23M | 238.76 | 1.0 GB | 35.94 it/s |
> > | P3D-S | 11.23M | 238.76 | 1.0 GB | 35.51 it/s |
> >
> > And for different window sizes:
> >
> > |   | Params | GFLOPs | Memory | Throughput |
> > | - | ------ | ------ | ------ | ---------- |
> > | P3D-S $w=2$ | 11.23M | 235.33 | 1.0 GB | 36.50 it/s |
> > | P3D-S $w=4$ | 11.23M | 238.76 | 1.0 GB | 35.51 it/s |
> > | P3D-S $w=8$ | 11.23M | 266.31 | 1.1 GB | 21.70 it/s |
> >
> > These numbers confirm that P3D has excellent scaling and throughput across architectural variations.

---

### Official Review · Reviewer_EzRu · 2025-10-31

**Soundness:** 3
**Presentation:** 2
**Contribution:** 3
**Rating:** 6
**Confidence:** 4

**Summary:**

The paper proposes a hybrid 3D CNN–Transformer surrogate that learns on small crops and fuses them with a lightweight global context sequence model (via adaln) to capture long-range dependencies while keeping memory/computation manageable. It scales to high-resolution 3D flows, and can also be trained probabilistically (using flow-matching) to generate channel-flow fields with faithful statistics across Reynolds numbers.

**Strengths:**

1. An innovative domain-decomposition neural surrogates with an effective and efficient module to fuse in global context, which is a meaningful step towards large-scale PDE modelling

2. The empirical performance of the model is quite strong. After training on local crop it can be deployed to larger domain.

3. Clear ablations isolate the global-context module as the source of gains over crop-only baselines.

**Weaknesses:**

The paper’s design and "why it works” intuition are not fully clear to me

1. Decoder/attention choice. The text says a Transformer decoder with “LLMAttention.” Does this mean causal attention over latent/region tokens? If so, why prefer causal over bi-directional for spatial/latent context aggregation (is it solely because of saving up computeduring fine-tuning)? A more in-depth discussion (e.g., compute/memory in backprop, training stability) would clarify the design choice.

2. Region token vs. simpler conditioning. For the region token, did the authors try a simpler variant: feeding a region index/ID embedding (analogous to a timestep/position embedding) directly into the diffusion model via adaln instead of introducing a learnable token?

**Questions:**

On patch artifacts in Fig. D9. I still notice some patch artifacts. Could this stem from padding choices in the conv layer (e.g., zero vs. reflect vs. circular)?

---

> ### Author Response · Authors · 2025-11-26
>
> Thank you for your detailed and constructive feedback. We greatly appreciate your suggestions for improving the paper and address each point below.
>
> > Decoder/attention choice. The text says a Transformer decoder with “LLM Attention.” Does this mean causal attention over latent/region tokens? If so, why prefer causal over bi-directional for spatial/latent context aggregation (is it solely because of saving up computed during fine-tuning)? A more in-depth discussion (e.g., compute/memory in backprop, training stability) would clarify the design choice.
>
> Our intention with the name "LLMAttention" was to emphasize that we can use efficient approximations and highly hardware-optimized implementations that are adopted from the LLM community. This does not encompass variants with causal attention mechanisms. In our experiments, we use self-attention based on all input tokens without any causal masking via HyperAttention [1].  We will clarify these aspects. Further experiments regarding different attention mechanisms, or alternatives such as state space models, are very interesting directions for future work.
>
> > Region token vs. simpler conditioning. For the region token, did the authors try a simpler variant: feeding a region index/ID embedding (analogous to a timestep/position embedding) directly into the diffusion model via adaln instead of introducing a learnable token?
>
>
> We agree that this is an interesting experiment. We investigated this specific alternative: replacing the learnable region token with a fixed positional embedding fed directly into the adaIN layers.
> Our preliminary evaluation (5 seeds, 20 epochs) using a fixed position embedding for each region token instead of learnable tokens achieves improvements over no finetuning, but does not perform as good as the learnable token variant, cf. the table below.
>
> | Validation Loss | Mean |
> | - | -- |
> | No finetuning | 0.195 |
> | Learnable Regions Tokens | 0.176 $\pm$ 0.017 |
> | Position Region Tokens | 0.187 $\pm$ 0.011 |
>
>
> > On patch artifacts in Fig. D9. I still notice some patch artifacts. Could this stem from padding choices in the conv layer (e.g., zero vs. reflect vs. circular)?
>
> These patch artifacts are due to the scaling strategy <X48|X48> which independently decodes regions. This strategy allows in principle to distribute the encoding and decoding of regions across different GPUs and all inter GPU communication is via the context network. Because patches are decoded independently, there are small discontinuities at the boundary between different regions, which is a limitation of the <X48|X48> scaling strategy. Mitigating these discontinuities could potentially be handled via additional (convolutional) post-processing layers.
>
> [1] Han, Insu, et al. "Hyperattention: Long-context attention in near-linear time." arXiv:2310.05869 (2023).

---

### Official Review · Reviewer_1qCk · 2025-11-02

**Soundness:** 2
**Presentation:** 2
**Contribution:** 2
**Rating:** 4
**Confidence:** 4

**Summary:**

This is a solid engineering effort with many moving parts: a UNet encoder–decoder, local attention, a small global context stack, a region‑token MLP conditioning, and a probabilistic training scheme. I am slightly concerned that the paper is an incremental 3D variant of the earlier "PDE-Transformer" approach which the authors cite as a predecessor. I think this is an interesting paper and the insights can prove useful to the community once disentangled. However, I think many of the architectural components are not well-justified by the experiments (particularly those choices which have fallen out of practice in the field), and the current evaluation has a few asymmetries that make the improvements over baselines harder to trust. I appreciate the amount of work that went in, but I would like to see some improvements to these points before acceptance, so that the paper and experiments can present insights in a more effective way to the community. I have compiled some notes below to help you strengthen the justification for your claims and make the evaluation fairer.

**Strengths:**

* Comprehensive set of experimental datasets and baselines across different physical regimes.
* Reasonable scaling argument via crop training and stitched inference.
* Dataset visualisations look quite polished.
* The context finetuning idea for channel flow is interesting when wall information is missing.

**Weaknesses:**

Major:

1. Novelty
   1. The paper reads as a 3D variant of prior "PDE-Transformer" work. It is a bit challenging to identify which architectural modification is the major contribution to the performance improvements, as this is something I would be quite interested in reading about. I recommend you isolate which component it is, and demonstrate effect sizes. Right now there are a lot of moving parts with insufficient ablation support.
   2. LLMLayer and "hyper attention" are mentioned but seemingly never defined despite being part of the "core architectural blocks". The only text I found on "hyper attention" in the paper is "We use hyper attention , this is a specialized attention variant designed for efficient long-sequence processing". Please spell out exactly what these are (I'm assuming these are from another paper you could simply cite), including complexity and memory scaling, or simply use a standard transformer block.

2. Baselines
   1. The crop axis of the evaluation feels hard to trust. P3D‑L at 32^3 beats Swin3D at 128^3 on the multi‑PDE suite. To me, this suggests under‑tuned baselines or other asymmetries in how the evaluation is being performed. It is perhaps also an issue with this type of evaluation, done by masking the data in a way that is suitable for the presented model. In addition, there is a fixed learning rate and weight regularization across all models, rather than using the reported training settings presented in each baseline's paper - which likely vary based on architecture settings. I would recommend at the very least using the same training settings as presented in baseline papers, or, even better, doing modest hyperparameter sweeps for the strongest baselines. Otherwise, baseline comparisons may lose their scientific value.
   2. Comparing speed against an accurate DNS is misleading. Comparisons need to be done either (1) on accuracy, with fixed compute budget for all models (likely resulting in using an LES, not a DNS), or, (2) on speed, at a fixed accuracy target, which is likely to result in using a coarser LES at comparable accuracy to your model. See panel A of Figure 1 of https://www.pnas.org/doi/10.1073/pnas.2101784118 for an example of what this should look like. At the moment, you are comparing diagonally in this accuracy-runtime plot, so it is impossible to draw meaningful conclusions. (Nevertheless, I commend the authors for running the DNS on the GPU, as doing a CPU comparison is the other common mistake.)
   3. For channel flow, from what I can tell, your model is conditioned on Re, but the baselines are not. Either condition baselines or train per‑Re baselines. In general I do not consider providing a model with an additional scalar context to be a novel contribution worthy of highlighting in a comparison. Also, the grid is non‑uniform in the wall‑normal direction; please explain how convolutions and positional information handle this and ensure baselines get the same treatment (assuming this is something you are controlling for).
   4. In general, I would like to see some spectra for the tasks. It is hard to know if your model is doing well by staring at thumbnail images of the density fields.
   5. The EMA reporting was a bit confusing to me. Please state which tables use EMA weights.

3. Ablations
   1. In general, there are many moving parts in this model. Any additional complexity needs to be justified with quantitative evidence, which I do not see as strong enough at the moment.
   2. Region‑token conditioning: please cite the original AdaIN/FiLM papers (I think AdaIN is mentioned by name, but not cited); please also clarify novelty. It would also be important to quantify effect size on at least two datasets.
   3. Context token density: each 16^3 crop becomes one latent token in the context model, so it would be useful to show sensitivity to token density (8^3/16^3/32^3) and where accuracy breaks.
   4. This is a point I am not confident about so please correct me if I am wrong. But I am confused about the positional encoding; the paper states "P3D does not include any absolute positional embeddings," yet the context model injects 3D sine–cosine PEs. But, since sine-cosine PEs are absolute, the global context actually seems to have full positional information. For relative positional encodings you would need to use RoPE.
   5. Regarding this "Inactive gradients" idea, this seems like another complex idea, which is also not completely necessary for the model itself, introduced in the paper without sufficient quantitative justification. I think this would not only result in a biased gradient estimator to save some VRAM (which does not seem to be theoretically explored), but would also introduce extra stochasticity into the training process. Why not use more standard approaches like gradient checkpointing or, e.g., LayerDrop?
   6. PixelShuffle3D: would be helpful to compare to more standard approaches to see how much this helps. It might be interesting for other contexts but at the moment it is hard to know whether this actually helps.
   7. The splits vary by time and by simulation index s which is a bit hard to follow. It might be nice to document the splits in a small table.


Minor:

1. In general, I felt like there was too much emphasis on the datasets and not enough emphasis on the model. These datasets are indeed visually nice to look at, but this paper is about the model. Only if the figures are part of the story should they be in the main text of the paper. For example, Figure 7, despite the nice 3D renders of this fluid, is completely unnecessary. I would include some spectra there instead so I can actually see how well the model is doing. Or other analysis on the model itself. Not to mention there is no reference flow for this figure, because it is crammed in to include the 3D render.
2. I felt like some of the math notation was a bit performative. For example, there was this $u(x, t)$ function defined with set theory notation, but then it was immediately dropped, even in the context of defining the $\mathbf{u}$ fields (it is not concrete how these two are related). If you need this to define the fields, then do it, but otherwise I would drop it.
3. The appendix feels too long and low-density in terms of information, it is a lot of the same figure generated for different datasets, many of which there is no discernible pattern at all that the reader should be following. The PDF is 40MB as a result of all of the figures. The dataset does not need to be a literal part of the PDF itself. The reader does not need to see all 9 timesteps, sliced through every axis, for a dataset that was (I think) not even introduced first by this paper.
4. For filtering choices, a Hanning window is applied before some vorticity visuals. Please state clearly whether spectra are computed on filtered or raw fields and show both.
5. "weight regularization" -> "weight decay".

**Questions:**

[Included in Weaknesses]

---

> ### Author Response · Authors · 2025-11-26
>
> Thank you for your detailed and constructive feedback. We greatly appreciate your suggestions for improving the paper and address each point below.
>
> 1. **Novelty**
>     - **PDE-Transformer Ablation** A key difference between the 3D extension of PDE-Transformer and the P3D backbone is the replacement of the patchification and projection layers with a convolutional encoder and decoder connected via skip-connections. This choice is theoretically motivated by the scaling of the number of tokens. In 2D, an input of size $np \times np$ with patch size $p$ yields $n^2$ tokens. In 3D, an input of $np \times np \times np$ yields $n^3$ tokens, while each token simultaneously encodes $p^3$ cells instead of $p^2$. This exacerbates the trade-off between computational efficiency and accuracy when moving from 2D to 3D. For this reason, we employ a convolutional encoder/decoder pair with skip connections to retain high-resolution features and achieve better computational efficiency.
>
>        As shown in Table 2 (original manuscript), P3D significantly improves over the 3D PDE-Transformer, showing improvements of 6.0% ($32^3$), 26.7% ($64^3$), and 37.5% ($128^3$) in terms of MSE. Table 2 also presents several ablations of the P3D backbone, detailing the performance of individual components like the convolutional encoder/decoder pair and the pure transformer backbone.
>
>     - Additional novel and major components beyond the P3D backbone are the crop training strategy, the domain scaling analysis, and the context network and region tokens used for efficient fine-tuning, which are thoroughly evaluated in Sections 4.2 and 4.3.
>
>     > LLMLayer and "hyper attention" are mentioned but seemingly never defined despite being part of the "core architectural blocks"... Please spell out exactly what these are...
>
>     Thank you for pointing out this omission, we will update the manuscript to clarify this point. The LLMLayer is simply a standard Transformer block (as described in Appendix A.4, "Core architectural blocks"). For the self-attention of latent tokens within this block, we use the Hyperattention algorithm [1]. We will update the text to clearly state that we use the standard Transformer block with the Hyperattention algorithm [1].
>
>
> 2. **Baselines**
> Running extensive experiments in 3D is computationally demanding (e.g., training a single model like $\text{UNet}_\text{GenCFD}$ at $128^3$ takes approximately 7 days on a 4-GPU node). We have verified proper training conditions for all models, but this is why we did not include broad a hyperparameter search in our paper. We have now performed a sweep over the learning rate for several baselines at crop size $32^3$ and report the results below (nRMSE):
>
>    | Learning rate | $10^{-3}$ | $5 \times 10^{-4}$ | $2 \times 10^{-4}$ | $10^{-4}$ | $5 \times 10^{-5}$ |
>    | ------------- | --------- | ------------------ | ------------------ | ------| ----------- |
>    | AFNO | 1.554 | 0.096 | 0.095 | 0.086 | 0.083 |
>    | Swin3D | 0.084 | 0.080 | 0.079 | 0.083 | 0.088 |
>    | UNet$_\text{GenCFD}$ | 0.966 | 0.085 | 0.076 | 0.079 | 0.079 |
>    | P3D-S | 0.066 | 0.054 | 0.062 | 0.057 | 0.064 |
>
>    Overall, the default learning rate ($2 \times 10^{-4}$) was well-chosen for most baselines, and we believe that the models are compared under fair conditions.

---

> > ### Author Response · Authors · 2025-11-26
> >
> > > The crop axis of the evaluation feels hard to trust. P3D‑L at 32^3 beats Swin3D at 128^3 on the multi‑PDE suite
> >
> > We understand your concern. Since the (test) nRMSE is averaged over 14 distinct PDE dynamics, poor performance on any single PDE can significantly skew the total result. This is exacerbated by the normalization of the nRMSE, making the metric generally less reliable for multi-PDE comparisons. While nRMSE is commonly used for multi-PDE tasks, looking at the (validation) MSE, which corresponds directly to the loss function, shows models are less spread out while the overall ranking is preserved.
> >
> > | Crop size | $32^3$ | $64^3$ | $128^3$ |
> > | ------ | --- | --- | --- |
> > | FactFormer | 0.0053 | 0.0022 | -
> > | UNet$_\text{GenCFD}$ | 0.0033 | 0.0045|0.00405
> > | AFNO   | 0.0054 | 0.0026 | 0.00277
> > | Swin3D | 0.0047 | 0.0027 | 0.00170
> > | P3D-S  | 0.0034 | 0.0016 |0.00077
> > | P3D-B  | 0.0033 | 0.0016 |0.00064
> > | P3D-L  | 0.0030 | 0.0010 | 0.00058
> >
> > For crop size $32^3$, we used the best learning rate for each model as determined by the hyperparameter sweep reported in the previous table.
> >
> > It is worth noting that UNet and AFNO in Table 1 of the main paper could potentially benefit from even smaller learning rates. While we observe training loss spikes for UNet at $64^3$ and $128^3$ and for AFNO at $128^3$, we do not see similar instabilities for experiments from Sections 4.2 and 4.3. We will retrain both models with a lower learning rate and report the updated results. We note that training UNet at $128^3$ takes approximately 7 days already, which places the final results after the discussion period. However, we do not believe these issues are critical to the analysis and conclusions of the experiment.
> >
> > > Comparing speed against an accurate DNS is misleading. Comparisons need to be done either (1) on accuracy, with fixed compute budget... or, (2) on speed, at a fixed accuracy target.
> >
> > We fully agree that the comparison should reflect the accuracy/speed trade-off. We recomputed the channel flow dataset using a solver with a coarser resolution ($48 \times 48 \times 96$ compared to the original $96 \times 96 \times 192$) using an otherwise identical configuration. The original high-resolution solver produces $0.0022$ samples/s, while the coarser solver produces $0.007$ samples/s. Note that the speedup is only moderate (3.18x) despite the considerably lower resolution, as it requires smaller timesteps in the adaptive time stepper module to preserve stability. We evaluated the Mean L2 and Variance L2 using the same methodology as the generative models (cf. Table 5). For the coarser solver, we obtained a Mean L2 of $1531.1 \times 10^{-5}$ and a Variance L2 of $59.5 \times 10^{-5}$.
> >
> > Below is the comparison of Mean L2 vs. Speedup (relative to the full-resolution DNS solver):
> >
> > | | Mean L2 | Speedup |
> > | - | ----- | ------- |
> > | AFNO | 28.73 | 1246x |
> > | UNet$_\text{GenCFD}$ | 132.38 | 81x |
> > | P3D-L | 3.02 | 144x
> > | Solver [$48^2 \times 92$] | 1531.1 | 3.18x |
> >
> > Thus, our trained model still clearly outperforms less accurate numerical simulations: the coarse simulation is still 45x slower, while yielding a larger error.
> >
> > Yes, you are correct; P3D-L was conditioned on the Reynolds (Re) number, which represents an evaluation asymmetry. We have rerun the training of P3D-L without Re number conditioning.
> >
> > |  | Mean L2 | Variance L2 |
> > | - | ------ | ----------- |
> > | with Re conditioning | 3.02 | 13.20 |
> > | without Re conditioning | 2.70 | 9.32 |
> >
> > Removing the Re conditioning did not have a detrimental effect on the evaluation. In fact, the metrics are slightly better, which we attribute to the natural variance of the training/evaluation process.

---

> > > ### Author Response · Authors · 2025-11-26
> > >
> > > > In general, I would like to see some spectra for the tasks. It is hard to know if your model is doing well by staring at thumbnail images of the density fields.
> > >
> > > We'd like to highlight the isotropic turbulence case from Section 4.2, for which we show spectra in Figure C3 in the Appendix, but only for the P3D-B model so far. The spectra match very well at t=1 and t=15, and we will update this plot in the revised manuscript to include other models as well.
> > >
> > > For Section 4.1, spectra vary widely across the different PDEs, complicating direct comparison. Also, due to training on cropped data, predictions with low errors tend to be smoother than the ground truth because of boundary uncertainty, which makes assessing performance via spectra more difficult. We therefore rely on MSE/nRMSE metrics, which more directly reflect the training objective.
> > >
> > > In Section 4.3, the channel flow is not periodic, again complicating the evaluation of spectra. For the channel flow task, visualizations of the mean velocity along the flow axis, cf. Figure D6 in the appendix, are used more commonly. We will include additional plots in the updated manuscript to also show the standard deviation of the velocity along the flow axis (this is already shown as a shaded area around the mean velocity in Figure D6). These are directly related to the Mean L2 and Variance L2 metrics in Table 5. We will expand Figure 7 to show these for P3D-L <48|192> and P3D-L <X48|X48>.
> > >
> > > > The EMA reporting was a bit confusing to me. Please state which tables use EMA weights.
> > >
> > > All model evaluations use EMA weights with a decay factor $\beta=0.999$. In cases where metrics are computed every epoch, we report the EMA over epochs with $\beta=0.99$. If this is the case, it is clearly noted in the table caption.
> > >
> > > 3. **Ablations**
> > >
> > > > Region‑token conditioning: please cite the original AdaIN/FiLM papers... please also clarify novelty. It would also be important to quantify effect size on at least two datasets.
> > >
> > > We will update the manuscript to explicitly cite the AdaIN/FiLM papers. While the use of AdaIN is standard across many architectures, the combination with region tokens for faster information flow during fine-tuning is, to the best of our knowledge, novel. We quantify the effect of region tokens in Section 4.3 using the turbulent channel flow. Although additional experiments would be beneficial, the channel flow experiment already clearly demonstrates the importance of information flow through region tokens for fine-tuning ($4541 \times 10^{-5}$ Mean L2 without region tokens vs. $23.6 \times 10^{-5}$ with region tokens).
> > >
> > > > Context token density: each 16^3 crop becomes one latent token in the context model, so it would be useful to show sensitivity to token density (8^3 / 16^3 / 32^3) and where accuracy breaks.
> > >
> > > That's an interesting experiment that we will consider for future work. The context token density is currently coupled to the architecture of the P3D encoder and decoder. Changing the density would therefore require altering the P3D backbone architecture itself, which would introduce an additional layer of complexity and new design hyperparameters.
> > >
> > > > This is a point I am not confident about so please correct me if I am wrong. But I am confused about the positional encoding; the paper states "P3D does not include any absolute positional embeddings," yet the context model injects 3D sine–cosine PEs...
> > >
> > > You are correct, and we apologize for the confusion. The statement that P3D lacks absolute positional embeddings refers only to the P3D backbone (the convolutional and transformer encoder/decoder pairs). The absolute positional encoding is only used for region tokens/latent tokens during fine-tuning via the context network. We will update the wording to clearly distinguish between the P3D backbone architecture and the extension using the context network and region tokens.
> > >
> > > > Regarding this "Inactive gradients" idea... Why not use more standard approaches like gradient checkpointing or, e.g., LayerDrop?
> > >
> > > We agree that gradient checkpointing and fully sharded training setups are viable alternatives. We empirically verified that fine-tuning with inactive gradients for a percentage of regions works, but we agree that checkpointing can achieve a similar benefit with potentially lower stochasticity. In our experience, methods like LayerDrop generally do not perform as well for scientific machine learning tasks as they do for computer vision, often resulting in lower accuracy, though we have not formally tested them for P3D. We note that inactive gradients can be combined with gradient checkpointing for further memory reduction. Furthermore, the region-based processing independently benefits inference speed, as encoding and decoding of different regions can be distributed across GPUs before global aggregation by the context network.

---

> > > > ### Author Response · Authors · 2025-11-26
> > > >
> > > > > PixelShuffle3D: would be helpful to compare to more standard approaches to see how much this helps.
> > > >
> > > > Could you clarify which alternatives or standard upsampling approaches you would like us to compare against? PixelShuffle2D is already used for upscaling in PDE-Transformer and architectures like U-DiT [2]. It is a very natural and efficient upsampling method that does not require any learnable weights and is easily combined with convolutional layers.
> > > >
> > > > > The splits vary by time and by simulation index s which is a bit hard to follow. It might be nice to document the splits in a small table.
> > > >
> > > > Thank you for the suggestion. We will add a corresponding table in the updated manuscript to clearly document the dataset splits.
> > > >
> > > > Regarding your additional points (minor):
> > > >
> > > > > In general, I felt like there was too much emphasis on the datasets and not enough emphasis on the model... For example, Figure 7, despite the nice 3D renders of this fluid, is completely unnecessary. I would include some spectra there instead...
> > > >
> > > > In Section 4.3, the channel flow is not periodic, which complicates the analysis/evaluation of spectra. For the channel flow task, visualizations of the mean velocity along the flow axis, cf. Figure D6 in the appendix, are used more commonly. We will include additional plots in the updated manuscript to also show the standard deviation of the velocity along the flow axis (this is already shown as a shaded area around the mean velocity in Figure D6). These are directly related to the Mean L2 and Variance L2 metrics in Table 5. We will expand Figure 7 to show these for P3D-L <48|192> and P3D-L <X48|X48>.
> > > >
> > > > > I felt like some of the math notation was a bit performative. For example, there was this $u(x,t)$ function defined with set theory notation...
> > > >
> > > > Are you referring to the notation $u(\mathbf{x}, t, \text{Re})$ in Appendix D.2? We used this definition to precisely define the values used in Table D1, which provides an additional comparison of velocity field moments. Our intention was to ensures clarity and precision in defining the data used for that specific comparison for readers.
> > > >
> > > > > The appendix feels too long and low-density in terms of information... The PDF is 40MB as a result of all of the figures...
> > > >
> > > > Thank you for the suggestion; we agree the appendix is quite long. The PDEs in Section 4.1 and the solver are based on [3]. However, we modified the hyperparameters and initial conditions to obtain more diverse and interesting 3D behavior, necessitating detailed descriptions for better reproducibility. Since Reviewer HKuH requested additional mathematical descriptions of each PDE, we are currently balancing these competing needs for information density. We will reduce file size of the PDF by changing the compression of images in the appendix.
> > > >
> > > > > For filtering choices, a Hanning window is applied before some vorticity visuals. Please state clearly whether spectra are computed on filtered or raw fields and show both.
> > > >
> > > > We compute spectra on the filtered fields. In the updated manuscript, we will include additional plots showing the energy spectra on the raw fields as well.
> > > >
> > > > > "weight regularization" -> "weight decay".
> > > >
> > > > We will update the manuscript to consistently use the term "weight decay".

---

> > > > > ### Author Response · Authors · 2025-11-26
> > > > >
> > > > > To summarize, we have performed additional experiments and provided clarifications to address the weaknesses you raised. These include:
> > > > > - Additional hyperparameter sweeps for baselines in Section 4.1.
> > > > > - A more in-depth discussion of nRMSE balancing and a comparison of models using MSE.
> > > > > - Fixing the Reynolds number conditioning in P3D in Section 4.3 for a fairer comparison.
> > > > > - Introducing a coarse solver for the channel flow experiment to compare models/solvers based on the speed vs. accuracy trade-off.
> > > > > - Additional clarifications on architectural details.
> > > > >
> > > > > We believe the ablation of P3D is strong, based on the ablation in Table 2 in Section 4.1, where different variants of P3D are compared. Table 2 also includes a comparison with the 3D PDE-Transformer, clearly demonstrating the accuracy improvement from the convolutional encoder/decoder pair. Region tokens are evaluated and analyzed in Table 5 in Section 4.3.
> > > > >
> > > > > We agree that several design choices could be further ablated and that additional experiments could provide interesting and useful insights. However, we believe the current experiments and evaluations are thorough, comprehensive and already provide a consistent picture. We kindly ask the reviewer to reconsider his/her initial assessment in light of these results and clarifications.
> > > > >
> > > > >
> > > > > [1] Han, Insu, et al. "Hyperattention: Long-context attention in near-linear time." arXiv:2310.05869 (2023).
> > > > >
> > > > > [2] Tian, Yuchuan, et al. "U-DiTs: Downsample Tokens in U-shaped Diffusion Transformers." Advances in Neural Information Processing Systems 37 (2024): 51994-52013.
> > > > >
> > > > > [3] Koehler, Felix, Simon Niedermayr, and Nils Thuerey. "APEBench: A benchmark for autoregressive neural emulators of PDEs." Advances in Neural Information Processing Systems 37 (2024): 120252-120310.

---

### Meta-Review · Area_Chair_H5f8 · 2026-01-04

**Summary:**

The initial reviews gave mixed scores and raised several major concerns to be addressed by the rebuttal:

1) Certain technical claims, e.g., the descriptions of the PDE problems and the statement about autoregressive rollouts, should be carefully revisited and toned down to better reflect the nature of the technical method.

2) Multiple reviewers suggested reporting rollout metrics to better understand the method’s performance in making stable predictions.

3) Several reviewers noted that the paper’s technical method contains many moving parts and suggested a more thorough ablation study to pinpoint its source of efficacy.

4) The intuition behind some design decisions in the technical method was unclear. Reviewers requested further clarifications or comparisons with alternative variants.

5) There were some critical questions about the comparisons with the baselines, e.g., whether the baselines have been properly tuned and whether the metrics have been evaluated in a fair setup for all methods.

There are also individual questions requesting more elaboration on the paper’s experimental setups and training details, a more upfront discussion of its limitations, clarification of its technical novelty relative to related network models, and so on.

**Reviewer Concerns:**

I think the rebuttal has addressed most concerns, though some minor issues may persist and affect the reviewers’ final evaluation. Below is my summary of the rebuttal’s response to each major concerns listed in the previous question:

1) The rebuttal fixed several claims and clarified its technical details. I think the response has properly addressed the concern.

2) The rebuttal added the requested rollout metrics at 512^3 resolution to evaluate the model’s performance, concluding that the proposed model remains competitive for up to 50 steps. I think this additional evaluation has properly addressed the concern.

3) In response to this question, the rebuttal has incorporated additional statistics (e.g., time, memory cost, and throughout) to enhance the ablation study. The rebuttal has also introduced new experiments to evaluate different variants of the proposed model. Overall, I think the current rebuttal has largely addressed this concern.

4) The rebuttal clarified the term “LLMAttention” and conducted an additional experiment to test the simple variant suggested by the reviewer. I think the clarification and additional experiments can partially address the reviewer’s question about “why it works”, but some in-depth discussion would be more helpful, which the rebuttal leaves as future work.

5) I think this is a critical concern that every research paper should treat carefully. The rebuttal provided additional statistics to address questions about crop-axis evaluation and how to conduct a fair speed-accuracy comparison. The rebuttal concluded that, based on the additional experiments, their proposed model remains competitive after incorporating the reviewer’s feedback. I think this response has partially addressed the concern. One remaining issue I have is that it is generally tricky to conduct a fair wall-clock time evaluation for numerical simulations and network methods, as it largely depends on their implementations on hardware platforms. This seems to be a common issue in the community, and I believe this paper has made a good-faith effort to conduct a fair and comprehensive comparison.

**Reviewer Scores:**

Most reviewers were already positive about the paper, and I think the rebuttal has addressed most remaining concerns. I expect that three positive reviewers (6/6/8) would maintain their support for this paper, and I think the negative reviewer (4) might be persuaded to become more supportive, given that the rebuttal has addressed many of the concerns about the baselines and the ablation study. Overall, I think the four reviewers would probably reach a consensus of accepting this work.

---

### Decision · Program_Chairs · 2026-01-26

Accept (Poster)